# Gaussian Approximation and Multiplier Bootstrap for Polyak-Ruppert Averaged Linear Stochastic Approximation with Applications to TD Learning

**Sergey Samsonov**
HSE University
svsamsonov@hse.ru

**Eric Moulines**
Ecole Polytechnique,
MBUZAI

**Qi-Man Shao**
Department of Statistics and Data Science,
Shenzhen International Center of Mathematics,
Southern University of Science and Technology

**Zhuo-Song Zhang**
Department of Statistics and Data Science,
Shenzhen International Center of Mathematics,
Southern University of Science and Technology

**Alexey Naumov**
HSE University,
Steklov Mathematical Institute
of Russian Academy of Sciences

## Abstract

In this paper, we obtain the Berry-Esseen bound for multivariate normal approximation for the Polyak-Ruppert averaged iterates of the linear stochastic approximation (LSA) algorithm with decreasing step size. Moreover, we prove the non-asymptotic validity of the confidence intervals for parameter estimation with LSA based on multiplier bootstrap. This procedure updates the LSA estimate together with a set of randomly perturbed LSA estimates upon the arrival of subsequent observations. We illustrate our findings in the setting of temporal difference learning with linear function approximation.

## 1 Introduction

Stochastic approximation (SA) methods are a central component for solving various optimization problems that arise in machine learning [32, 26], empirical risk minimization [72] and reinforcement learning [42, 67]. There is a vast number of contributions in the literature, which cover both asymptotic [48, 53] and non-asymptotic [45, 15, 36] properties of the SA estimates. The primarily important property among the asymptotic ones of the SA estimates is their asymptotic normality [53], which is important due to its role in constructing (asymptotic) confidence intervals and hypothesis testing [71]. However, a natural question of the rate of convergence in the appropriate central limit theorems (CLT) is not well addressed in literature even in the relatively simple setting of the linear stochastic approximation (LSA) [21], [34], [9].

Alternatively, confidence sets for SA algorithms can be constructed in a non-asymptotic manner based on concentration inequalities [4]. These bounds are often regarded as loose [60], yielding suboptimal performance of the statistical procedures based on the latter estimates [28]. In contrast, for statistical inference procedures based on independent and identically distributed (i.i.d.) observations, such as $M$-estimators [71], there is a machinery of non-parametric methods for constructing confidence sets with the bootstrap [19, 58]. This approach is accompanied with theoretical guarantees, showing the non-asymptotic validity of the bootstrap-based confidence intervals for parameters in linear regression [64] and statistical tests [12]. Extending theoretical guarantees to a non-classical situation with online learning algorithms encounters serious problems, essentially related to the problem of obtaining rate of convergence in the corresponding CLTs. At the same time, many phenomena arising in the analysis of nonlinear SA algorithms already appear in the analysis of LSA problems.

38th Conference on Neural Information Processing Systems (NeurIPS 2024).

The LSA procedure aims to find an approximate solution for the linear system $\bar{\mathbf{A}}\theta^\star = \bar{\mathbf{b}}$ with a unique solution $\theta^\star$ based on a sequence of observations $\{(\mathbf{A}(Z_k), \mathbf{b}(Z_k))\}_{k\in\mathbb{N}}$. Here $\mathbf{A} : \mathsf{Z} \to \mathbb{R}^{d\times d}$ and $\mathbf{b} : \mathsf{Z} \to \mathbb{R}^d$ are measurable functions and $(Z_k)_{k\in\mathbb{N}}$ is a sequence of noise variables taking values in some measurable space $(\mathsf{Z}, \mathcal{Z})$ with a distribution $\pi$ satisfying $\mathbb{E}[\mathbf{A}(Z_k)] = \bar{\mathbf{A}}$ and $\mathbb{E}[\mathbf{b}(Z_k)] = \bar{\mathbf{b}}$. We focus on the setting of independent and identically distributed (i.i.d.) observations $\{Z_k\}_{k\in\mathbb{N}}$. With a sequence of decreasing step sizes $(\alpha_k)_{k\in\mathbb{N}}$ and the starting point $\theta_0 \in \mathbb{R}^d$, we consider the estimates $\{\bar{\theta}_n\}_{n\in\mathbb{N}}$ given by

$$\theta_k = \theta_{k-1} - \alpha_k\{\mathbf{A}(Z_k)\theta_{k-1} - \mathbf{b}(Z_k)\}, \ \ k \geq 1, \quad \bar{\theta}_n = n^{-1}\sum_{k=n}^{2n-1}\theta_k, \ \ n \geq 1. \tag{1}$$

Here, we have fixed the size of the *burn-in* period (see, e.g., [16, 44]) to $n_0 = n$. Provided that $n$ is large enough, the burn-in size affects only a constant factor in the subsequent bounds. The sequence $\{\theta_k\}_{k\in\mathbb{N}}$ corresponds to the standard LSA iterates, while $\{\bar{\theta}_n\}_{n\in\mathbb{N}}$ corresponds to the Polyak-Ruppert (PR) averaged iterates [59, 53]. It is known that $\bar{\theta}_n$ is asymptotically normal with a minimax-optimal covariance matrix (see [53] and [23] for discussion). Specifically, under appropriate technical conditions on the step sizes $\{\alpha_k\}$ and noisy observations $\{\mathbf{A}(Z_k)\}$, it holds that

$$\sqrt{n}(\bar{\theta}_n - \theta^\star) \xrightarrow{d} \mathcal{N}(0, \Sigma_\infty),$$

where $\Sigma_\infty$ is the asymptotic covariance matrix defined later in Section 3.1. There is a long list of contributions to the non-asymptotic analysis of $\bar{\theta}_n$, particularly [43] and [16], which study moment and Bernstein-type concentration bounds for $\sqrt{n}(\bar{\theta}_n - \theta^\star)$. Unfortunately, such bounds do not imply Berry-Esseen type inequalities for $\sqrt{n}(\bar{\theta}_n - \theta^\star)$, that is, they do not allow us to control the quantity

$$\rho_n^{\mathrm{Conv}} = \sup_{B\in\mathrm{Conv}(\mathbb{R}^d)}\left|\mathbb{P}\big(\sqrt{n}(\bar{\theta}_n - \theta^\star) \in B\big) - \mathbb{P}(\Sigma_\infty^{1/2}\eta \in B)\right|, \tag{2}$$

where $\mathrm{Conv}(\mathbb{R}^d)$ refers to the set of convex sets in $\mathbb{R}^d$. While the Berry-Esseen bounds are a popular subject of study in probability theory, starting from the classical work [20], most results are obtained for sums of random variables or martingale difference sequences [52, 8]. We can only mention a few results for SA algorithms, see Section 2 for more details. This paper aims to provide the latter bounds for the specific setting of the LSA procedure. Our primary contribution is twofold:

- We establish a BerryEsseen bound for accuracy of normal approximation of the distribution of Polyak-Ruppert averaged LSA iterates with a polynomially decreasing step size. Our results suggest that the best rate of normal approximation, in the sense of (2), is of order $n^{-1/4}$ up to logarithmic factors in $n$, where $n$ denotes the number of samples. Interestingly, this rate is achieved with an aggressive step size, $\alpha_k = c_0/\sqrt{k}$. Our proof technique follows the Berry-Esseen bounds for nonlinear statistics provided in [63].
- We provide non-asymptotic confidence bounds for the distribution of the PR-averaged statistic $\sqrt{n}(\bar{\theta}_n - \theta^\star)$ using the multiplier bootstrap procedure. In particular, our bounds imply that the quantiles of the exact distribution of $\sqrt{n}(\bar{\theta}_n - \theta^\star)$ can be approximated at a rate of $n^{-1/4}$, where $n$ is the number of samples used in the procedure, provided that $n$ is sufficiently large (see A4 for exact conditions). To the best of our knowledge, this is the first non-asymptotic bound on the accuracy of bootstrap approximation in SA algorithms. We apply the proposed methodology to the temporal difference learning (TD) algorithm for policy evaluation in reinforcement learning.

The rest of the paper is organized as follows. In Section 2, we provide a literature review on the non-asymptotic analysis of the LSA algorithm and bootstrap methods. Next, in Section 3, we analyze the convergence rate of Polyak-Ruppert averaged LSA iterates to the normal distribution. In Section 4, we discuss the multiplier bootstrap approach for LSA and establish bounds on the accuracy of approximating the quantiles of the true distribution. Finally, we apply our findings to TD learning and present numerical illustrations in Section 5.

**Notations.** For matrix $A \in \mathbb{R}^{d\times d}$ we denote by $\|A\|$ its operator norm. For symmetric matrix $Q = Q^\top \succ 0$, $Q \in \mathbb{R}^{d\times d}$ and $x \in \mathbb{R}^d$ we define the corresponding norm $\|x\|_Q = \sqrt{x^\top Q x}$, and define the respective matrix $Q$-norm of the matrix $B \in \mathbb{R}^{d\times d}$ by $\|B\|_Q = \sup_{x\neq 0}\|Bx\|_Q/\|x\|_Q$. For sequences $a_n$ and $b_n$, we write $a_n \lesssim b_n$ if there exist a constant $c > 0$ such that $a_n \leq cb_n$ for $c > 0$. For simplicity we state the main results of the paper up to constant factors.

## 2 Related works

Among contributions to the analysis of the LSA algorithm, we should mention the papers [53, 34, 9, 6]. These works investigate the asymptotic properties of the LSA estimates (such as asymptotic normality and almost sure convergence) under i.i.d. and Markov noise. Non-asymptotic results for the LSA and PR-averaged LSA estimates were obtained in [55, 47, 7, 35, 44], where MSE bounds were established, and in [43, 17, 16], which provided high-probability error bounds. The latter results enable the construction of Bernstein-type confidence intervals for the error $\bar{\theta}_n - \theta^\star$. Unfortunately, the corresponding bounds typically depend on unknown problem properties of (1), related to the design matrix $\bar{\mathbf{A}}$ and the noise variables $\mathbf{A}(Z_k)$, $\mathbf{b}(Z_k)$. For this reason, applying these error bounds in practice is complicated. Furthermore, concentration bounds for the LSA error [43, 17, 16] do not imply convergence rates of the rescaled error $\sqrt{n}(\bar{\theta}_n - \theta^\star)$ to the normal distribution in Wasserstein or Kolmogorov distance. Non-asymptotic convergence rates were previously studied in [2] using the Stein method, but the resulting rate corresponds to a smoothed Wasserstein distance. Recent work [65] investigates convergence rates to the normal distribution in Wasserstein distance for LSA with Markovian observations. Both papers yield bounds that are less tight with respect to their dependence on trajectory length $n$ than those presented in the present work, see a detailed comparison after Theorem 2.

A popular method for constructing confidence intervals in the context of parametric estimation is based on the bootstrap approach ([19]). Its analysis has attracted many contributions, in particular a series of papers [12] and [13] that validate a bootstrap procedure for a test based on the maximum of a large number of statistics. Their study shows a close relationship between bootstrap validity results, Gaussian comparison and anticoncentration bounds for rectangular sets. The papers [64] and [27] investigate the applicability of likelihood-based statistics for finite samples and large parameter dimensions under possible model misspecification. The important step in proving bootstrap validity is again based on Gaussian comparison and anticoncentration bounds, but now for spherical sets. The bootstrap procedure for spectral projectors of covariance matrices is discussed in [46] and [31]. The authors follow the same steps to prove the validity of the bootstrap.

Extending the classical bootstrap approach to online learning algorithms is a challenge. For example, the iterates $\{\theta_k\}_{k\in\mathbb{N}}$ determined by (1) are not necessarily stored in memory, which makes the classical bootstrap inapplicable. This problem can be solved by performing randomly perturbed updates of the online procedure, as proposed in [22] for the iterates of the Stochastic Gradient Descent (SGD) algorithm. The authors in [56] used the same procedure for the case of Markov noise and policy evaluation algorithms in reinforcement learning, but in both papers the authors only consider the asymptotic validity. In our paper we use the same multiplier bootstrap approach (see Section 4), but we provide an explicit error bound for the bootstrap approximation of the distribution of the statistics $\sqrt{n}(\bar{\theta}_n - \theta^\star)$.

In addition to the bootstrap approach, one can also use the pivotal statistics [37, 40, 41] or various estimates of the asymptotic covariance matrix [73] to construct the confidence intervals for $\theta^\star$. The latter approach can be based on the plug-in estimators [39], batch mean estimators [11] or in combination with the multiplier bootstrap approach [74]. However, the theoretical guarantees for mentioned methods remain purely asymptotic.

## 3 Accuracy of normal approximation for LSA

We first study the rate of normal approximation for the tail-averaged LSA procedure. When there is no risk of ambiguity, we use simply the notations $\mathbf{A}_k = \mathbf{A}(Z_k)$ and $\mathbf{b}_k = \mathbf{b}(Z_k)$. Starting from the definition (1), we get with elementary transformations that

$$\theta_k - \theta^\star = (\mathrm{I} - \alpha_k \mathbf{A}_k)(\theta_{k-1} - \theta^\star) - \alpha_k \varepsilon_k , \tag{3}$$

where we have set $\varepsilon_k = \varepsilon(Z_k)$ with

$$\varepsilon(z) = \tilde{\mathbf{A}}(z)\theta^\star - \tilde{\mathbf{b}}(z) , \quad \tilde{\mathbf{A}}(z) = \mathbf{A}(z) - \bar{\mathbf{A}} , \quad \tilde{\mathbf{b}}(z) = \mathbf{b}(z) - \bar{\mathbf{b}} .$$

Here the random variable $\varepsilon(Z_k)$ can be viewed as a noise, measured at the optimal point $\theta^\star$. We now assume the following technical conditions:

**A1.** *Sequence $\{Z_k\}_{k\in\mathbb{N}}$ is a sequence of i.i.d. random variables defined on a probability space $(\Omega, \mathcal{F}, \mathbb{P})$ with distribution $\pi$.*

**A2.** $\int_Z \mathbf{A}(z)\mathrm{d}\pi(z) = \bar{\mathbf{A}}$ *and* $\int_Z \mathbf{b}(z)\mathrm{d}\pi(z) = \bar{\mathbf{b}}$, *with the matrix* $-\bar{\mathbf{A}}$ *being Hurwitz. Moreover,* $\|\varepsilon\|_\infty = \sup_{z \in Z}\|\varepsilon(z)\| < +\infty$, *and the mapping* $z \to \mathbf{A}(z)$ *is bounded, that is,*

$$C_{\mathbf{A}} = \sup_{z \in Z}\|\mathbf{A}(z)\| \vee \sup_{z \in Z}\|\tilde{\mathbf{A}}(z)\| < \infty . \tag{4}$$

*Moreover, for the noise covariance matrix*

$$\Sigma_\varepsilon = \int_Z \varepsilon(z)\varepsilon(z)^\top \mathrm{d}\pi(z) \tag{5}$$

*it holds that its smallest eigenvalue is bounded away from* $0$, *that is,*

$$\lambda_{\min} := \lambda_{\min}(\Sigma_\varepsilon) > 0 . \tag{6}$$

It is possible to change (4) to the moment-type bound as it was previously considered in [43] and [16], see the detailed discussion after Theorem 2. The fact that the matrix $-\bar{\mathbf{A}}$ is Hurwitz implies that the linear system $\bar{\mathbf{A}}\theta = \bar{\mathbf{b}}$ has a unique solution $\theta^\star$. Moreover, this fact is sufficient to show that the matrix $\mathrm{I} - \alpha\bar{\mathbf{A}}$ is a contraction in an appropriate matrix $Q$-norm for small enough $\alpha > 0$. Precisely, the following result holds:

**Proposition 1.** *Let* $-\bar{\mathbf{A}}$ *be a Hurwitz matrix. Then for any* $P = P^\top \succ \mathrm{I}$, *there exists a unique matrix* $Q = Q^\top \succ \mathrm{I}$, *satisfying the Lyapunov equation* $\bar{\mathbf{A}}^\top Q + Q\bar{\mathbf{A}} = P$. *Moreover, setting*

$$a = \frac{\lambda_{\min}(P)}{2\|Q\|} , \quad \text{and} \quad \alpha_\infty = \frac{\lambda_{\min}(P)}{2\kappa_Q\|\bar{\mathbf{A}}\|_Q^2} \wedge \frac{\|Q\|}{\lambda_{\min}(P)} , \tag{7}$$

*where* $\kappa_Q = \lambda_{\max}(Q)/\lambda_{\min}(Q)$, *it holds for any* $\alpha \in [0, \alpha_\infty]$ *that* $\alpha a \leq 1/2$, *and*

$$\|\mathrm{I} - \alpha\bar{\mathbf{A}}\|_Q^2 \leq 1 - \alpha a . \tag{8}$$

The proof of Proposition 1 is provided in Appendix D.1. Note that it is possible to set $P = \mathrm{I}$ as in [18], yet it is possible that other choices of $P$ could be more beneficial for particular applications. Now consider an assumption on the step sizes $\alpha_k$ and number of observations $n$:

**A3.** *The step sizes* $\{\alpha_k\}_{k \in \mathbb{N}}$ *has a form* $\alpha_k = c_0/k^\gamma$, *where* $\gamma \in [1/2; 1)$ *and* $c_0 \in (0; \alpha_\infty \wedge a \wedge (1 - \gamma)]$. *Moreover, we assume that* $n \geq d$, *and*

$$\begin{cases} \frac{\sqrt{n}}{(1+\log n)\log n} \geq \frac{c_0\kappa_Q C_{\mathbf{A}}^2}{a(1-\sqrt{2}/2)} \vee \frac{4}{ac_0(1-\sqrt{2}/2)} , & \text{if } \gamma = 1/2 , \\ \frac{n^{1-\gamma}}{\log n} \geq \frac{2c_0\kappa_Q C_{\mathbf{A}}^2}{a(2\gamma-1)(1-(1/2)^{1-\gamma})} \vee \frac{8\gamma(1-\gamma)}{ac_0(1-(1/2)^{1-\gamma})} , & \text{if } 1/2 < \gamma < 1 . \end{cases} \tag{9}$$

The main aim of lower bounding $n$ is to ensure that the number of observations is large enough in order that the LSA error related to the choice of initial condition $\theta_0 - \theta^\star$ becomes small.

## 3.1 Central limit theorem for Polyak-Ruppert averaged LSA iterates.

It is known that the assumptions A1-A3 guarantee that the CLT applies to the iterates of $\bar{\theta}_n$, namely,

$$\sqrt{n}(\bar{\theta}_n - \theta^\star) \xrightarrow{d} \mathcal{N}(0, \Sigma_\infty) , \tag{10}$$

where the asymptotic covariance matrix $\Sigma_\infty$ has a form

$$\Sigma_\infty = \bar{\mathbf{A}}^{-1}\Sigma_\varepsilon\bar{\mathbf{A}}^{-\top}, \tag{11}$$

and $\Sigma_\varepsilon$ is defined in (5). This result can be found for example in [53] and [23]. We are interested in the Berry-Esseen type bound for the rate of convergence in (10), that is, we aim to bound $\rho_n^{\text{Conv}}$ defined in (2) w.r.t. the available sample size $n$. We control $\rho_n^{\text{Conv}}$ using a method from [63] based on randomized multivariate concentration inequality. Below we briefly state its setting and required definitions. Let $X_1, \ldots, X_n$ be independent random variables taking values in $\mathcal{X}$ and $T = T(X_1, \ldots, X_n)$ be a general $d$-dimensional statistics such that $T = W + D$, where

$$W = \sum_{\ell=1}^n \xi_\ell, \quad D := D(X_1, \ldots, X_n) = T - W, \tag{12}$$

$\xi_\ell = h_\ell(X_\ell)$ and $h_\ell : \mathcal{X} \to \mathbb{R}^d$ is a Borel measurable function. Here the statistics $D$ can be non-linear and is treated as an error term, which is "small" compared to $W$ in an appropriate sense. Assume that $\mathbb{E}[\xi_\ell] = 0$ and $\sum_{\ell=1}^n \mathbb{E}[\xi_\ell \xi_\ell^\top] = I_d$. Let $\Upsilon = \Upsilon_n = \sum_{\ell=1}^n \mathbb{E}[\|\xi_\ell\|^3]$. Then, with $\eta \sim \mathcal{N}(0, I_d)$,

$$\sup_{B \in \mathrm{Conv}(\mathbb{R}^d)} |\mathbb{P}(T \in A) - \mathbb{P}(\eta \in A)| \leq 259 d^{1/2} \Upsilon + 2\mathbb{E}[\|W\|\|D\|] + 2\sum_{\ell=1}^n \mathbb{E}[\|\xi_\ell\|\|D - D^{(\ell)}\|], \quad (13)$$

where $D^{(\ell)} = D(X_1, \ldots, X_{\ell-1}, X'_\ell, X_{\ell+1}, \ldots, X_n)$ and $X'_\ell$ is an independent copy of $X_\ell$. This result is due to [63, Theorem 2.1]. One can modify the bound (13) for the setting when $\sum_{\ell=1}^n \mathbb{E}[\xi_\ell \xi_\ell^\top] = \Sigma \succ 0$. This result due to [63, Corollary 2.3]. Following the construction (12), we set $T = \sqrt{n}\bar{\mathbf{A}}(\bar{\theta}_n - \theta^\star)$ and consider it as a nonlinear statistic of i.i.d. random variables $Z_1, \ldots, Z_{2n}$, which drive the LSA dynamics (1). We can exactly represent $T$ as a sum of linear ($W$) and non-linear parts ($D$), where

$$W = -\frac{1}{\sqrt{n}} \sum_{k=n}^{2n-1} \varepsilon_{k+1}, \quad D = \frac{1}{\sqrt{n}} \frac{\theta_n - \theta^\star}{\alpha_n} - \frac{1}{\sqrt{n}} \frac{\theta_{2n} - \theta^\star}{\alpha_{2n}} - \frac{1}{\sqrt{n}} \sum_{k=n+1}^{2n} (\mathbf{A}_k - \bar{\mathbf{A}})(\theta_{k-1} - \theta^\star)$$

$$+ \frac{1}{\sqrt{n}} \sum_{k=n+1}^{2n} (\theta_{k-1} - \theta^\star) \left( \frac{1}{\alpha_k} - \frac{1}{\alpha_{k-1}} \right).$$

The proof of this result can be bound in Proposition 3. To obtain a bound for the approximation accuracy in (2) using the bound (13), we need to upper bound $\mathbb{E}^{1/2}[\|D(Z_1, \ldots, Z_{2n})\|^2]$ and $\mathbb{E}[\|D - D^{(i)}\|]$. The first result below provides a second moment bound on $D$:

**Theorem 1.** *Assume A1, A2, and A3. Then we obtain the following error bound:*

$$\mathbb{E}^{1/2}\left[\|D(Z_1, \ldots, Z_{2n})\|^2\right] \lesssim \frac{\sqrt{\kappa_Q}\|\varepsilon\|_\infty}{\sqrt{ac_0}} \left( \frac{1}{n^{(1-\gamma)/2}} + \frac{c_0 C_{\mathbf{A}}}{\sqrt{1-\gamma}n^{\gamma/2}} \right)$$

$$+ \sqrt{\kappa_Q}\Delta_1 \exp\left\{ -\frac{c_0 a n^{1-\gamma}}{2(1-\gamma)} \right\} \|\theta_0 - \theta^\star\|,$$

*where $\lesssim$ stands for inequality up to an absolute constant, and $\Delta_1 = \Delta_1(n, a, C_{\mathbf{A}}, c_0)$ is a polynomial function defined in Appendix A.3, eq. (29).*

The proof of Theorem 1 is provided in Appendix A.3. Now it remains to upper bound the term $\mathbb{E}[\|D - D^{(i)}\|]$, which is done in Appendix B.1 using the synchronous coupling methods [10]. Combining these bounds, we obtain the following theorem:

**Theorem 2.** *Assume A1, A2, and A3. Then the following bound holds:*

$$\rho_n^{\mathrm{Conv}} \lesssim \frac{d^{1/2}\|\varepsilon\|_\infty^3}{\lambda_{\min}^{3/2}\sqrt{n}} + \frac{1}{\lambda_{\min}} \left( \frac{C_1}{n^{(1-\gamma)/2}} + \frac{C_2}{n^{\gamma/2}} \right) + \frac{\Delta_2}{\lambda_{\min}} \exp\left\{ -\frac{c_0 a n^{1-\gamma}}{2(1-\gamma)} \right\} \|\theta_0 - \theta^\star\|, \quad (14)$$

*where $\Delta_2 = \Delta_2(n, a, C_{\mathbf{A}}, \mathrm{Tr}\,\Sigma_\varepsilon, c_0)$ is a polynomial function defined in (35), and constants $C_1, C_2$, depending upon $a, C_{\mathbf{A}}, \kappa_Q, \mathrm{Tr}\,\Sigma_\varepsilon, c_0$, are defined in Appendix B, eq. (36).*

The proof of Theorem 2 is provided in Appendix B. Note that the assumption A2 requires that $\varepsilon(Z_1)$ is almost sure bounded. It is a strong assumption, but it can be partially relaxed. Following the stability of matrix products technique, used in [17, Proposition 3], it is possible to consider the setting when the random variable $\|\tilde{\mathbf{A}}(Z_1)\|$ has only finite number of moments. In particular, we expect that assuming finite third moment of $\|\tilde{\mathbf{A}}(Z_1)\|$ and $\|\varepsilon(Z_1)\|$ is sufficient to obtain a counterpart to Theorem 1. However, this generalization requires non-trivial technical work on generalizing the stability of matrix products result (see Corollary 4 in Appendix D ).

Note that the bound of Theorem 2 predicts the optimal error of normal approximation for Polyak-Ruppert averaged estimates of order $n^{-1/4}$, which is achieved with the aggressive step size $\alpha_k = c_0/\sqrt{k}$, that is, when setting $\gamma = 1/2$ in (14). In this case we obtain the optimized bound

$$\rho_n^{\mathrm{Conv}} \lesssim \frac{C_3}{\lambda_{\min} n^{1/4}} + \frac{d^{1/2}\|\varepsilon\|_\infty^3}{\lambda_{\min}^{3/2}\sqrt{n}} + \frac{\Delta_1 \exp\{-c_0 a\sqrt{n}\}}{\lambda_{\min}} \|\theta_0 - \theta^\star\|, \quad (15)$$

where $C_3 = C_3(a, C_{\mathbf{A}}, \kappa_Q, \mathrm{Tr}\,\Sigma_\varepsilon, \|\varepsilon\|_\infty)$ is provided in (36).

**Discussion.** Our proof technique of Theorem 2 reveals an interesting feature: fastest rate of convergence in the convex distance $\rho_n^{\mathrm{Conv}}$ corresponds to the learning rate schedule that admits the fastest decay of the second-order term in the MSE bound for remainder statistics $D$ (see Theorem 1). Results similar to the one of Theorem 2 have been recently obtained in the literature in [65] and [2]. The author in [65] considers the LSA problem specified to the temporal-difference learning (see Section 5) with Markov noise and obtains convergence rate in Wasserstein distance of order $n^{-1/4}$, which corresponds to the "optimal" step size schedule $\alpha_k = c_0/k^{3/4}$. Using the bound of [49, eq. (3)] (see also section 2 in [57]), this result yield a suboptimal bound of order $n^{-1/8}$ for the convex distance $\rho_n^{\mathrm{Conv}}$. Such an upper bound may be loose for some classes of distributions, but it is not clear if in particular setting of LSA the bound of [65] could imply scaling of order $n^{-1/4}$ for $\rho_n^{\mathrm{Conv}}$. At the same time, in case of $X_1, \ldots, X_n$ forming a Markov chain in (12) there is no available counterpart of the bound (13). Generalizing (13) is an interesting research direction that would allow to obtain a counterpart of Theorem 2 in case of Markovian dynamics. Similarly, the result of [2] holds for much stronger metrics, which controls the convergence of moments of twice differentiable functions. We provide additional details about connections between this metric and $\rho_n^{\mathrm{Conv}}$ in Appendix B.2. At the same time, the authors in [2] cover the non-linear setting of PR-averaged iterates of stochastic gradient descent algorithm under strong convexity.

**Remark 1.** *The leading (with respect to $n$) terms of the bound from Theorem 1 have an implicit dependence on the problem dimension $d$ due to the presence of $\lambda_{\min}$. Yet the result of Theorem 1 can be improved in a sense of dependence in dimension if one is interested not in the rates of convergence for $\sqrt{n}(\bar\theta_n - \theta^\star)$, but in the projected iterated $\sqrt{n}\Pi^\top(\bar\theta_n - \theta^\star)$ for some $\Pi \in \mathbb{R}^{d \times m}$, $m \le d$. If this is the case, one may apply (13) for the class $\mathrm{Conv}_m = \mathrm{Conv}(\mathbb{R}^m)$ of convex sets in $\mathbb{R}^m$ and obtain, setting step size $\alpha_k = c_0/\sqrt{k}$, and $\Sigma_\varepsilon^{(\Pi)} = \Pi\Sigma_\varepsilon\Pi^\top$, that*

$$\rho_n^{\mathrm{Conv}} \lesssim \frac{\mathsf{C}_4}{\lambda_{\min} n^{1/4}} + \frac{m^{1/2}\|\varepsilon\|_\infty^3}{\lambda_{\min}^{3/2}\sqrt{n}} + \frac{\Delta_2 e^{-c_0 a\sqrt{n}}}{\lambda_{\min}}\|\theta_0 - \theta^\star\| \;,$$

*and the constant $\mathsf{C}_4 = \mathsf{C}_4(a, \mathsf{C}_{\mathbf{A}}, \kappa_Q, \mathrm{Tr}\,\Sigma_\varepsilon^{(\Pi)}, \|\varepsilon\|_\infty)$ is provided in (36).*

**Remark 2.** *Results similar to Theorem 1 can be obtained not only for the Polyak-Ruppert averaged estimator $\bar\theta_n$, but also for the last iterate $\theta_n$. In particular, it is known (see e.g. [23]), that the last iterate error $\theta_n - \theta^\star$ is also asymptotically normal:*

$$\frac{\theta_n - \theta^*}{\sqrt{\alpha_n}} \to \mathcal{N}(0, \Sigma_{\mathrm{last}}) \;,$$

*where the covariance matrix $\Sigma_{\mathrm{last}}$ is different from $\Sigma_\infty$. In such a case $\Sigma_{\mathrm{last}}$ can be found as a solution to appropriate Lyapunov equation, see [23]. Then, we expect that it is possible to use the perturbation-expansion approach from [1] together with randomized concentration inequalities [63] (see (13)), in order to obtain the Berry-Esseen bound*

$$\sup\nolimits_{B \in \mathrm{Conv}(\mathbb{R}^d)} \left| \mathbb{P}\big(\tfrac{\theta_n - \theta^*}{\sqrt{\alpha_n}} \in B\big) - \mathbb{P}(\Sigma_{\mathrm{last}}^{1/2}\eta \in B) \right| \lesssim \sqrt{\alpha_n} \;.$$

*We leave the detailed derivation for future work.*

## 4 Multiplier bootstrap for LSA

In order to perform statistical inference with the Polyak-Ruppert estimator $\bar\theta_n$, we propose an online bootstrap resampling procedure, which recursively updates the LSA estimate as well as a large number of randomly perturbed LSA estimates, upon the arrival of each data point. The suggested procedure follows the one outlined in [22]. It has the following advantages: it does not rely on the asymptotic distribution of the error $\sqrt{n}(\bar\theta_n - \theta^\star)$, does not require to know the moments of $\sqrt{n}(\bar\theta_n - \theta^\star)$ or its asymptotic covariance matrix $\Sigma_\infty$, and does not involve any data splitting.

We state the suggested procedure as follows. Let $\mathcal{W}^{2n} = \{W_\ell\}_{1 \le \ell \le 2n}$ be a set of i.i.d. random variables, independent of $\mathcal{Z}^{2n} = \{Z_\ell\}_{1 \le \ell \le 2n}$, with $\mathbb{E}[W_1] = 1$ and $\mathrm{Var}[W_1] = 1$. We write, respectively, $\mathbb{P}^{\mathsf{b}} = \mathbb{P}(\cdot|\mathcal{Z}^{2n})$ and $\mathbb{E}^{\mathsf{b}} = \mathbb{E}(\cdot|\mathcal{Z}^{2n})$ for the corresponding conditional probability and expectation. In parallel with procedure (1) that generates $\{\theta_k\}_{1 \le k \le 2n}$ and $\bar\theta_n$, we generate $M$ independent samples $(w_n^\ell, \ldots, w_{2n}^\ell)$, $1 \le \ell \le M$ distributed as $\mathcal{W}^{2n}$, and recursively update $M$ randomly perturbed LSA estimates, that is,

$$\begin{aligned} \theta_k^{\mathsf{b},\ell} &= \theta_{k-1}^{\mathsf{b},\ell} - \alpha_k w_k^\ell \{\mathbf{A}(Z_k)\theta_{k-1}^{\mathsf{b},\ell} - \mathbf{b}(Z_k)\} \;, \quad k \ge n+1 \;, \quad \theta_n^{\mathsf{b},\ell} = \theta_n \;, \\ \bar\theta_n^{\mathsf{b},\ell} &= n^{-1}\sum\nolimits_{k=n}^{2n-1} \theta_k^{\mathsf{b},\ell} \;, \quad n \ge 1 \;. \end{aligned} \tag{16}$$

We use a short notation $\bar{\theta}_n^{\mathsf{b}}$ for $\bar{\theta}_n^{\mathsf{b},1}$. The key idea of the procedure (16) is that the "Bootstrap-world" distribution (that is, the one conditional on $\mathcal{Z}^{2n}$) of the perturbed samples $\sqrt{n}(\bar{\theta}_n^{\mathsf{b}} - \bar{\theta}_n)$ is close to the distribution of the quantity of interest, that is, $\sqrt{n}(\bar{\theta}_n - \theta^\star)$. Precisely, the main result of this section will show that the quantity

$$\sup_{B \in \mathrm{Conv}(\mathbb{R}^d)} |\mathbb{P}^{\mathsf{b}}(\sqrt{n}(\bar{\theta}_n^{\mathsf{b}} - \bar{\theta}_n) \in B) - \mathbb{P}(\sqrt{n}(\bar{\theta}_n - \theta^\star) \in B)| \tag{17}$$

is small. Although an analytic expression for $\mathbb{P}^{\mathsf{b}}(\sqrt{n}(\bar{\theta}_n^{\mathsf{b}} - \bar{\theta}_n) \in B)$ is not available, one can approximate it from numerical simulations according to (16) by generating sufficiently large number $M$ of perturbed trajectories. Standard arguments, see e.g. [62, Section 5.1] suggest that the accuracy of Monte-Carlo approximation is of order $M^{-1/2}$. To analyze the suggested procedure, we shall impose an additional assumption on the trajectory length $n$:

**A4.** *Assumption A3 holds with $\gamma = 1/2$, and $c_0 \leq 1/(\mathrm{C}_{\mathbf{A}}^2 \kappa_Q \mathrm{e})$. Moreover, setting*

$$h(n) = \left\lceil \left(\frac{4\,\mathrm{C}_{\mathbf{A}}\,\kappa_Q^{1/2}}{(\sqrt{2}-1)a}\right)^2 (1 + 2\log{(2n^4)})^2 \right\rceil, \tag{18}$$

*it holds that*

$$\frac{\sqrt{n}}{h(n)} \geq \frac{2}{a(\sqrt{2}-1)} \vee \frac{c_0}{\alpha_\infty}, \text{ and } \frac{\sqrt{n}}{\log^2 n} \geq \frac{c_0(1 \vee \mathrm{C}_{\mathbf{A}}^2)}{a} \vee c_0 a\, \mathrm{C}_{\mathbf{A}}^2 \vee \frac{4}{ac_0} \tag{19}$$

*Moreover, we assume that for $\lambda_{\min}$ defined in (6) it holds that*

$$\lambda_{\min} \geq 8\|\varepsilon\|_\infty \sqrt{\frac{\|\Sigma_\varepsilon\| \log n}{n}} + \frac{8(\|\Sigma_\varepsilon\| + \|\varepsilon\|_\infty^2) \log n}{n} \tag{20}$$

Note that the new bound (19) simply states that $\sqrt{n}/\log^2(n)$ is sufficiently large, since $h(n)$ scales as $\log^2 n$. We discuss the assumption A4 in more details in the proof scheme. Now we formulate the main result of this section. We analyze only the setting of polynomially decaying step size with $\gamma = 1/2$, since decay rate of (17) essentially depends on the approximation rate of Theorem 2, with the fastest rate achieved when $\gamma = 1/2$. For other learning rates the decay rate of right-hand side in Theorem 3 will be slower. For simplicity, we do not trace the dependence of the bound below on the parameter $c_0$.

**Theorem 3.** *Assume A1, A2, A3 with $\gamma = 1/2$, and A4. Then with $\mathbb{P}$ – probability at least $1 - 6/n$ it holds that*

$$\sup_{B \in \mathrm{Conv}(\mathbb{R}^d)} |\mathbb{P}^{\mathsf{b}}(\sqrt{n}(\bar{\theta}_n^{\mathsf{b}} - \bar{\theta}_n) \in B) - \mathbb{P}(\sqrt{n}(\bar{\theta}_n - \theta^\star) \in B)| \lesssim \frac{\kappa_Q^2(\mathrm{C}_{\mathbf{A}}^4 \vee 1)(1 + \|\varepsilon\|_\infty^2) \log n}{a^{5/2}\lambda_{\min} n^{1/4}}$$

$$+ \frac{\sqrt{d}}{\sqrt{n}} \left(\frac{\|\varepsilon\|_\infty^3}{\lambda_{\min}^{3/2}} + \kappa_Q\|\varepsilon\|_\infty \frac{\sqrt{\log n}}{\sqrt{\lambda_{\min}}} + \frac{\kappa_Q(1 + \|\varepsilon\|_\infty^2/\lambda_{\min}) \log n}{\sqrt{n}}\right) + \frac{\Delta_3 \mathrm{e}^{-(c_0/2)a\sqrt{n}}}{\lambda_{\min}}\|\theta_0 - \theta^\star\|,$$

*where $\Delta_3 = \Delta_3(n, a, \mathrm{C}_{\mathbf{A}}, \|\varepsilon\|_\infty)$ is a polynomial function defined in Appendix C, eq. (46).*

The proof of Theorem 3 is based on the Gaussian approximation performed both in the "real" world and bootstrap world together with an appropriate Gaussian comparison inequality. The main steps of the proof are illustrated by the following scheme:

Real world: $\quad \sqrt{n}\bar{\mathbf{A}}(\bar{\theta}_n - \theta^\star) \xleftarrow{\text{Gaussian approximation, Th. 2}} \xi \sim \mathcal{N}(0, \Sigma_\varepsilon)$

$$\Big\updownarrow \text{Gaussian comparison, Theorem 5}$$

Bootstrap world: $\sqrt{n}\bar{\mathbf{A}}(\bar{\theta}_n^{\mathsf{b}} - \bar{\theta}_n) \xleftarrow{\text{Gaussian approx. in Bootstrap world, Th. 4}} \xi^{\mathsf{b}} \sim \mathcal{N}(0, \Sigma_\varepsilon^{\mathsf{b}})$

In the above scheme we have denoted by $\Sigma_\varepsilon^{\mathsf{b}} = n^{-1}\sum_{\ell=n}^{2n-1} \varepsilon_\ell \varepsilon_\ell^\top$ the sample covariance matrix approximating $\Sigma_\varepsilon$. Gaussian approximation for the true distribution of $\sqrt{n}\bar{\mathbf{A}}(\bar{\theta}_n - \theta^\star)$ follows from Theorem 2. Proof of Gaussian approximation in the Bootstrap world Theorem 4 is also based on the inequality (13), but is more complicated and involves the expansion analysis of the LSA error from [1]. This technique allows to separate the LSA error into different scales with respect to the step sizes $\{\alpha_k\}$, see Appendix C.4 for details. However, this technique requires to impose additional

assumption A4 - eq. (19). Proof of the Gaussian comparison part of Theorem 5 is based on Pinsker's inequality and matrix Bernstein inequality. The latter result requires that $n$ is large enough to ensure that minimal eigenvalue of $\Sigma_\varepsilon^b$ is close to $\lambda_{\min}$, justifying the assumption A4 - eq. (20). Detailed proof if provided in Appendix C.

**Discussion.** We emphasize that the Gaussian approximation result of Theorem 2 (with Bootstrap world generalization in Theorem 4) is a key result to prove the above bootstrap validity. This argument was missing in the earlier works studying confidence intervals for stochastic optimization algorithms [11, 73, 74], where the authors considered procedures to estimate $\Sigma_\infty$ in (11). They combine *non-asymptotic* bounds on the accuracy of recovering $\Sigma_\infty$ with only *asymptotic* validity of the resulting confidence intervals. We expect that our proof technique for Theorem 2 can be used to provide similar non-asymptotic validity results for outlined approaches for constructing confidence intervals based on the estimation of the asymptotic covariance matrix.

**Corollary 1.** *(Set of Euclidean balls or ellipsoids) Suppose that we are interested in estimating quantile of a given order $\alpha \in (0,1)$ and some matrix $B \in \mathbb{R}^{d \times d}$, that is, the quantity*

$$t_\alpha = \inf\{t > 0 : \mathbb{P}(\sqrt{n}\|B(\bar{\theta}_n - \theta^\star)\| \geq t) \leq \alpha\}.$$

*We define its counterpart in the Bootstrap world, $t_\alpha^b$, as*

$$t_\alpha^b = \inf\{t > 0 : \mathbb{P}^b(\sqrt{n}\|B(\bar{\theta}_n^b - \bar{\theta}_n)\| \geq t) \leq \alpha\}.$$

*Note that $t_\alpha^b$ is defines with respect to the bootstrap measure, therefore, it depends on the data $\mathcal{Z}^{2n}$. This bootstrap critical value $t_\alpha^b$ is applied in the Bootstrap world to build the confidence set*

$$\mathcal{E}(\alpha) = \{\theta \in \mathbb{R}^d : \sqrt{n}\|B(\theta - \bar{\theta}_n)\| \leq t_\alpha^b\} \,.$$

*Theorem 3 justifies this construction and evaluate the coverage probability of the true value $\theta^\star$ by this set. It states that*

$$\mathbb{P}(\theta^\star \notin \mathcal{E}(\alpha)) = \mathbb{P}(\sqrt{n}\|B(\bar{\theta}_n - \theta^\star)\| > t_\alpha^b) \approx \alpha \,,$$

*with the error of order $n^{-1/4}$ in the right-hand side. Although an analytic expression for $t_\alpha^b$ is not available, one can approximate it by generating a large number $M$ of independent samples of $\mathcal{W}_n$ and computing from them the empirical distribution function of $\sqrt{n}\|B(\bar{\theta}_n^b - \bar{\theta}_n)\|$, following (16).*

**Remark 3.** *A natural question that arises after Theorem 3 is whether it is possible to prove similar bounds for the iterates of first-order stochastic optimization algorithms. There are several MSE bounds for corresponding algorithms with explicit dependence on the step size $\alpha_k$; see, for example, [45, 5]. Therefore, we expect that it is possible to obtain a counterpart to Theorem 2. At the same time, for general first-order stochastic optimization algorithms, unlike LSA, there are no counterparts to the precise error expansions of [1]. Thus, proving the counterpart of Theorem 3 in this setting is more challenging. Similarly, we emphasize that generalizations of the procedure (16) to cases where $\{Z_k\}_{k \in \mathbb{N}}$ are dependent, for example, form a Markov chain, are complicated. The approach of [63] is not directly applicable in this setting, and appropriate generalization of (13) is a separate and challenging research direction.*

## 5 Applications to the TD learning and numerical results

We illustrate our findings for the setting of temporal difference (TD) learning algorithm [66, 67] for policy evaluation in RL. Non-asymptotic error bounds for this algorithm attracted lot of contributions [43, 16, 30, 51, 38]. At the same time, confidence intervals for TD were studied in [22, 56] only in terms of their asymptotic validity. In the TD algorithm we consider a discounted MDP (Markov Decision Process) given by a tuple $(\mathcal{S}, \mathcal{A}, \mathrm{P}, r, \gamma)$. Here $\mathcal{S}$ and $\mathcal{A}$ stand for state and action spaces, and $\gamma \in (0,1)$ is a discount factor. Assume that $\mathcal{S}$ is a complete metric space with metric $\mathrm{d}_\mathcal{S}$ and Borel $\sigma$-algebra $\mathcal{B}(\mathcal{S})$. $\mathrm{P}$ stands for the transition kernel $\mathrm{P}(B|s,a)$, which determines the probability of moving from state $s$ to a set $B \in \mathcal{B}(\mathcal{S})$ when action $a$ is performed. Reward function $r \colon \mathcal{S} \times \mathcal{A} \to [0,1]$ is assumed to be deterministic. *Policy* $\pi(\cdot|s)$ is the distribution over action space $\mathcal{A}$ corresponding to agent's action preferences in state $s \in \mathcal{S}$. We aim to estimate *value function*

$$V^\pi(s) = \mathbb{E}\big[\sum_{k=0}^\infty \gamma^k r(s_k, a_k)|s_0 = s\big] \,,$$

where $a_k \sim \pi(\cdot|s_k)$, and $s_{k+1} \sim \mathrm{P}(\cdot|s_k, a_k)$ for any $k \in \mathbb{N}$. Define the transition kernel under $\pi$,

$$\mathrm{P}_\pi(B|s) = \int_{\mathcal{A}} \mathrm{P}(B|s, a)\pi(\mathrm{d}a|s) , \tag{21}$$

which corresponds to the 1-step transition probability from state $s$ to a set $B \in \mathcal{B}(\mathcal{S})$. The state space $\mathcal{S}$ here can be arbitrary. It is a common option to consider the *linear function approximation* for $V^\pi(s)$, defined for $s \in \mathcal{S}$, $\theta \in \mathbb{R}^d$, and a feature mapping $\varphi: \mathcal{S} \to \mathbb{R}^d$ as $V_\theta^\pi(s) = \varphi^\top(s)\theta$. Here $d$ is the dimension of feature space. Our goal is to find a parameter $\theta^\star$ which is defined as a unique solution to the projected Bellman equation, see [70]. We denote by $\mu$ the invariant distribution over the state space $\mathcal{S}$ induced by $\mathrm{P}^\pi(\cdot|s)$ in (21). We define the *design matrix* $\Sigma_\varphi$ as

$$\Sigma_\varphi = \mathbb{E}_\mu[\varphi(s)\varphi(s)^\top] \in \mathbb{R}^{d \times d} . \tag{22}$$

Consider the following assumptions on the generative mechanism and on the feature mapping $\varphi(\cdot)$:

**TD 1.** *Tuples $(s, a, s')$ are generated i.i.d. with $s \sim \mu$, $a \sim \pi(\cdot|s)$, $s' \sim \mathrm{P}(\cdot|s, a)$.*

**TD 2.** *Matrix $\Sigma_\varphi$ is non-degenerate with the minimal eigenvalue $\lambda_{\min}(\Sigma_\varphi) > 0$. Moreover, the feature mapping $\varphi(\cdot)$ satisfies $\sup_{s \in \mathcal{S}} \|\varphi(s)\| \leq 1$.*

In the setting of linear function approximation the estimation of $V^\pi(s)$ reduces to estimating $\theta^\star \in \mathbb{R}^d$, which can be done via the LSA procedure. Here, the $k$-th step randomness is given by the tuple $Z_k = (s_k, a_k, s_k')$. Then, the corresponding LSA update can be written as

$$\theta_k = \theta_{k-1} - \alpha_k(\mathbf{A}_k\theta_{k-1} - \mathbf{b}_k) , \tag{23}$$

where $\mathbf{A}_k$ and $\mathbf{b}_k$ are given, respectively, by

$$\mathbf{A}_k = \varphi(s_k)\{\varphi(s_k) - \gamma\varphi(s_k')\}^\top , \quad \mathbf{b}_k = \varphi(s_k)r(s_k, a_k) .$$

We provide the expressions for the corresponding system matrix $\bar{\mathbf{A}} = \mathbb{E}[\mathbf{A}_k]$ and the right-hand side $\bar{\mathbf{b}}$ in Appendix E. We verify that assumption A2 holds and, furthermore, we provide a tighter counterpart to the result of Proposition 1. This result closely follows [51] and [61].

**Proposition 2.** *Let $\{\theta\}_{k \in \mathbb{N}}$ be a sequence of TD updates generated by (23) under **TD 1** and **TD 2**. Then this update scheme satisfies assumption A2 with*

$$\mathrm{C}_{\mathbf{A}} = 2(1 + \gamma) , \quad \|\varepsilon\|_\infty = 2(1 + \gamma)(\|\theta^\star\| + 1) ,$$

*moreover, one can check that $\|\mathrm{I} - \alpha\bar{A}\|^2 \leq 1 - \alpha a$ with*

$$a = (1 - \gamma)\lambda_{\min}(\Sigma_\varphi) , \quad \alpha_\infty = (1 - \gamma)/(1 + \gamma)^2 ,$$

*that is, Proposition 1 holds with $Q = \mathrm{I}$.*

Proof of Proposition 2 is provided in Appendix E. Since all the assumptions in A2 are fulfilled, we can verify tightness of the bound Theorem 2 for different learning rate schedules $\alpha_k$ in (23).

**Numerical results.** Efficiency of the multiplier bootstrap approach (16) to the problems of constructing confidence sets in online algorithms has been demonstrated in the works [22] and [56]. We aim to illustrate the tightness of our bounds for normal approximation outlined in Theorem 2 in the setting of TD learning with linear function approximation. To this end, we consider the classical Garnet problem [3], in the simplified version proposed by [25]. This problem is characterized by the number of states $N_s$, number of actions $a$, and branching factor $b$ (i.e. the number of neighbors of each state in the MDP). We set these values to $N_s = 10$, $a = 2$ and $b = 3$, and aim to evaluate the value function of the randomly generated policy $\pi(\cdot|s)$. Details on the way the policy $\pi$ is set can be found in Appendix F. We consider the problem of policy evaluation in this MDP using the TD learning algorithm with identity feature mapping, that is, $\phi(s) = e_s$ (that is, $s$-th coordinate vector) for $s \in \{1, \ldots, N_s\}$. We run the procedure (23) with the learning rates $\alpha_k = c_0/k^\gamma$ and different powers $\gamma \in \{0.5, 0.65, 0.7\}$. For each of the experiments we aim to estimate the supremum

$$\Delta_n := \sup_{x \in \mathbb{R}} \left| \mathbb{P}(\sqrt{n}\|\bar{\theta}_n - \theta^\star\| \leq x) - \mathbb{P}(\|\Sigma_\infty^{1/2}\eta\| \leq x) \right| , \tag{24}$$

$\eta \sim \mathcal{N}(0, \mathrm{I}_{N_s})$, and show that this supremum scales as $n^{-1/4}$ when $\gamma = 1/2$ and admits slower decay for other powers of $\gamma$. We approximate true probability $\mathbb{P}(\|\Sigma_\infty^{1/2}\eta\| \leq x)$ by the corresponding empirical probabilities based on sample of size $M \gg n$. Second, for $n \in \{1600, \ldots, 1638400\}$, where next sample size is twice larger than the previous one, we generate $N = 6553600$ trajectories of TD algorithm and approximate the distribution of $\sqrt{n}\|\bar{\theta}_n - \theta^\star\|$ based on the corresponding empirical distribution. We report our results in Figure 1, showing that the smallest values of $\Delta_n$ correspond to the step size schedule $\gamma = 1/2$, moreover, the decay rate $n^{-1/4}$ seems to be tight, otherwise one should expect further decay of $\Delta_n n^{1/4}$. Additional simulations are provided in Appendix F.

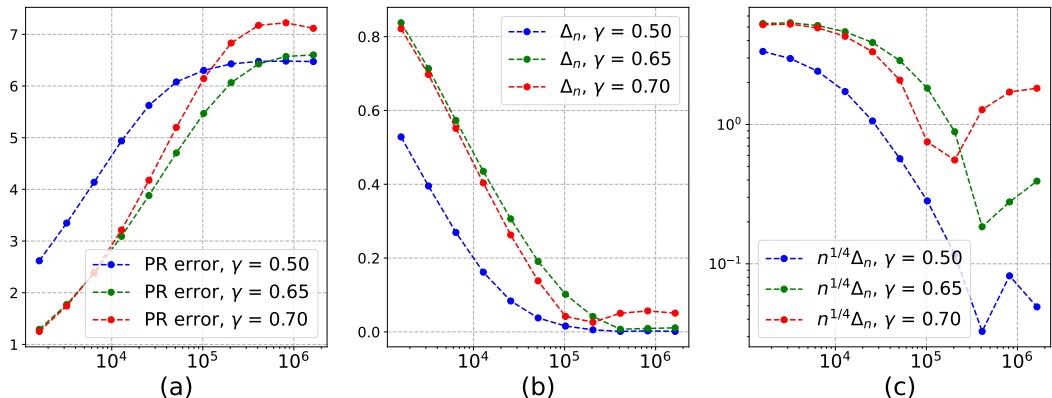

Figure 1: Subfigure (a): Rescaled error $\sqrt{n}\|\bar{\theta}_n - \theta^\star\|$, averaged over $N$ independent TD trajectories for different trajectory lengths $n$. Subfigure (b): approximate quantity $\Delta_n$ from (24) for different powers $\gamma$ and $n$. Subfigure (c): $\Delta_n$, rescaled by a factor $n^{1/4}$, predicted by Theorem 2.

## 6 Conclusion

In this paper, we have established, to the best of our knowledge, the first fully non-asymptotic confidence bounds for parameter estimation in the LSA algorithm using the multiplier bootstrap. This result is based on a novel Berry-Esseen bound for the Polyak-Ruppert averaged LSA iterates, which is of independent interest. Our paper suggests several interesting directions for further research. First, our Berry-Esseen bounds are obtained using the randomized concentration inequality [63], and it would be valuable to generalize this approach to the setting of Markov chains. Second, it is natural to extend our results to the first-order gradient methods, both for stochastic optimization and variational inequalities. Third, it becomes possible to prove the fully non-asymptotic validity of confidence intervals obtained with plug-in techniques or other estimators of the asymptotic covariance matrix of $\bar{\theta}_n$. These could then be compared with the multiplier bootstrap confidence intervals in terms of their dependence on problem dimension $d$ and other instance-dependent quantities.

## Acknowledgement

The work of S. Samsonov and A. Naumov was prepared within the framework of the HSE University Basic Research Program. The work of E. Moulines has been partly funded by the European Union (ERC-2022-SYG-OCEAN-101071601). Views and opinions expressed are however those of the author(s) only and do not necessarily reflect those of the European Union or the European Research Council Executive Agency. Neither the European Union nor the granting authority can be held responsible for them. The work of Q.-M. Shao is partially supported by National Nature Science Foundation of China NSFC 12031005 and Shenzhen Outstanding Talents Training Fund, China. The work of Z.-S. Zhang is partially supported by National Nature Science Foundation of China NSFC 12301183 and National Nature Science Found for Excellent Young Scientists Fund. This research was supported in part through computational resources of HPC facilities at HSE University [33].

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

# A    Proofs for accuracy of normal approximation

## A.1    Expansion of the error of LSA equipped with the Polyak-Ruppert averaging

**Proposition 3.** *The following expansion holds:*

$$
\sqrt{n}\bar{\mathbf{A}}(\bar{\theta}_n - \theta^\star) = -\underbrace{\frac{1}{\sqrt{n}}\sum_{k=n+1}^{2n}\varepsilon_k}_{W} + \underbrace{\frac{1}{\sqrt{n}}\frac{\theta_n - \theta^\star}{\alpha_n}}_{D_1} - \underbrace{\frac{1}{\sqrt{n}}\frac{\theta_{2n} - \theta^\star}{\alpha_{2n}}}_{D_2}
$$

$$
-\underbrace{\frac{1}{\sqrt{n}}\sum_{k=n+1}^{2n}(\mathbf{A}_k - \bar{\mathbf{A}})(\theta_{k-1} - \theta^\star)}_{D_3} + \underbrace{\frac{1}{\sqrt{n}}\sum_{k=n+1}^{2n}\left(\theta_{k-1} - \theta^\star\right)\left(\frac{1}{\alpha_k} - \frac{1}{\alpha_{k-1}}\right)}_{D_4} \quad (25)
$$

*Proof.* We use the recurrence (3) and rewrite it as

$$
\theta_k - \theta^\star = (\mathrm{I} - \alpha_k\bar{\mathbf{A}})(\theta_{k-1} - \theta^\star) - \alpha_k(\mathbf{A}_k - \bar{\mathbf{A}})(\theta_{k-1} - \theta^\star) - \alpha_k\varepsilon_k \ .
$$

The previous equation implies, after algebraic manipulation and division by $\alpha_k$, that

$$
\bar{\mathbf{A}}(\theta_{k-1} - \theta^\star) = \frac{\theta_{k-1} - \theta^\star}{\alpha_k} - \frac{\theta_k - \theta^\star}{\alpha_k} - (\mathbf{A}_k - \bar{\mathbf{A}})(\theta_{k-1} - \theta^\star) - \varepsilon_k \ .
$$

Taking average for $k$ from $n+1$ to $2n$ and multiplying by $\sqrt{n}$, we obtain (25).    □

## A.2    Bounding the error of the LSA algorithm last iterate

We begin with of technical lemma on the behavior of the last iterate $\theta_k$ of the LSA procedure given in (1). We aim to show that $\mathbb{E}^{1/p}[\|\theta_k - \theta^\star\|^p]$ scales as $\sqrt{\alpha_k}$, provided that $k$ is large enough. This result is classical and appears in a number of papers, e.g. [7, 14, 43, 17]. We provide the proof here for completeness. Our analysis of the bootstrap procedure and the last iterate error of LSA procedure is based on the error expansion technique from [1], see also [16]. Namely, to perform the expansion, we decompose the LSA iterates $\theta_k$ defined in (1) into a transient and fluctuation terms:

$$
\theta_k - \theta^\star = \tilde{\theta}_k^{(\mathrm{tr})} + \tilde{\theta}_k^{(\mathrm{fl})} \ ,
$$

where we have defined the quantities

$$
\tilde{\theta}_k^{(\mathrm{tr})} = \Gamma_{1:k}\{\theta_0 - \theta^\star\} \ , \quad \tilde{\theta}_k^{(\mathrm{fl})} = -\sum_{j=1}^{k}\alpha_j\Gamma_{j+1:k}\varepsilon_j \ , \quad (26)
$$

setting

$$
\Gamma_{m:k} = \prod_{i=m}^{k}(\mathrm{I} - \alpha_i\mathbf{A}(Z_i)) \ , \quad m, k \in \mathbb{N}, m \le k \ , \text{ with the convention } \Gamma_{m:k} = \mathrm{I} \ , m > k \ . \quad (27)
$$

The dependence of $\Gamma_{m:k}$ upon the stepsizes $(\alpha_j)$ is implicit in (27). Here the quantity $\tilde{\theta}_k^{(\mathrm{tr})}$ is the transient component of the error, which determines the rate at which the initial error $\theta_0 - \theta^\star$ is forgotten. The term $\tilde{\theta}_k^{(\mathrm{fl})}$ corresponds to the fluctuation component of the error and is determined by the oscillations of the last iterate $\theta_k$ around $\theta^\star$.

**Proposition 4.** *Assume A 1, A 2, and A 3. Then for any $k \ge n$, where $n$ satisfies (9), it holds for $2 \le p \le \log n^2$, that*

$$
\mathbb{E}^{1/p}[\|\theta_k - \theta^\star\|^p] \le \sqrt{\kappa_Q}\mathrm{e}\exp\Big\{-(a/2)\sum_{\ell=1}^{k}\alpha_\ell\Big\}\|\theta_0 - \theta^\star\| + \frac{4\mathrm{e}\sqrt{\kappa_Q}\|\varepsilon\|_\infty p}{\sqrt{a}}\sqrt{\alpha_k} \ .
$$

*Proof.* Expanding the decomposition (26), we obtain that

$$\mathbb{E}^{1/p}[\|\theta_k - \theta^\star\|^p] \leq \mathbb{E}^{1/p}[\|\Gamma_{1:k}\{\theta_0 - \theta^\star\}\|^p] + \mathbb{E}^{1/p}[\|\sum_{j=1}^{k}\alpha_j\Gamma_{j+1:k}\varepsilon_j\|^p] \,, \tag{28}$$

and we bound both terms separately. Since the sample size $n$ satisfies (9), we get applying Corollary 4 (see equation (71)), that for $2 \leq p \leq \log n^2$ it holds

$$\mathbb{E}^{1/p}[\|\Gamma_{1:k}\{\theta_0 - \theta^\star\}\|^p] \leq \sqrt{\kappa_Q}\mathrm{e}\exp\{-(a/2)\sum_{\ell=1}^{k}\alpha_\ell\}\|\theta_0 - \theta^\star\| \,.$$

Now we proceed with the second term in (28). Applying Burholder's inequality [50, Theorem 8.6] and Lemma 3 with $b = a/4$, we obtain that

$$\mathbb{E}^{1/p}[\|\sum_{j=1}^{k}\alpha_j\Gamma_{j+1:k}\varepsilon_j\|^p] \leq p\left(\mathbb{E}^{2/p}\left[\left(\sum_{j=1}^{k}\alpha_j^2\|\Gamma_{j+1:k}\varepsilon_j\|^2\right)^{p/2}\right]\right)^{1/2}$$

$$\leq p\left(\sum_{j=1}^{k}\alpha_j^2\mathbb{E}^{2/p}\left[\|\Gamma_{j+1:k}\varepsilon_j\|^p\right]\right)^{1/2}$$

$$\leq p\sqrt{\kappa_Q}\mathrm{e}\|\varepsilon\|_\infty\left(\sum_{j=1}^{k}\alpha_j^2\prod_{\ell=j+1}^{k}\left(1 - \frac{a\alpha_\ell}{4}\right)\right)^{1/2}$$

$$\leq \frac{4\mathrm{e}\sqrt{\kappa_Q}\|\varepsilon\|_\infty p}{\sqrt{a}}\sqrt{\alpha_k} \,.$$

$\square$

**Corollary 2.** *Under assumptions of Proposition 4, it holds that*

$$\mathbb{P}\left(\exists k \in [n, 2n - 1] : \|\theta_k - \theta^\star\| \geq g(k, \|\theta_0 - \theta^\star\|, n)\right) \leq \frac{1}{n} \,,$$

*where we have defined*

$$g(k, \|\theta_0 - \theta^\star\|, n) = \sqrt{\kappa_Q}\mathrm{e}^2\exp\{-(a/2)\sum_{\ell=1}^{k}\alpha_\ell\}\|\theta_0 - \theta^\star\| + \frac{8\mathrm{e}^2\sqrt{\kappa_Q}\|\varepsilon\|_\infty\log n}{\sqrt{a}}\sqrt{\alpha_k} \,.$$

*Proof.* We first note that Lemma 1 implies, setting $\delta = 1/n^2$, that for every fixed $k \in [n; 2n - 1]$,

$$\mathbb{P}\left(\|\theta_k - \theta^\star\| \geq \sqrt{\kappa_Q}\mathrm{e}^2\exp\{-(a/2)\sum_{\ell=1}^{k}\alpha_\ell\}\|\theta_0 - \theta^\star\| + \frac{8\mathrm{e}^2\sqrt{\kappa_Q}\|\varepsilon\|_\infty\log n}{\sqrt{a}}\sqrt{\alpha_k}\right) \leq \frac{1}{n^2} \,.$$

Application of the union bound concludes the proof. $\square$

We conclude this part with a simple consequence of Markov's inequality.

**Lemma 1.** *Fix $\delta \in (0, 1/\mathrm{e}^2)$ and let $Y$ be a positive random variable, such that*

$$\mathbb{E}^{1/p}[Y^p] \leq C_1 + C_2 p$$

*for any $2 \leq p \leq \log(1/\delta)$. Then it holds with probability at least $1 - \delta$, that*

$$Y \leq \mathrm{e}C_1 + \mathrm{e}C_2\log(1/\delta) \,.$$

*Proof.* Applying Markov's inequality, for any $t \geq 0$ we get that

$$\mathbb{P}(Y \geq t) \leq \frac{\mathbb{E}[Y^p]}{t^p} \leq \frac{(C_1 + C_2 p)^p}{t^p} \,.$$

Now we set $p = \log(1/\delta)$, $t = \mathrm{e}C_1 + \mathrm{e}C_2\log(1/\delta)$, and aim to check that

$$\frac{(C_1 + C_2\log(1/\delta))^{\log(1/\delta)}}{(\mathrm{e}C_1 + \mathrm{e}C_2\log(1/\delta))^{\log(1/\delta)}} \leq \delta \,.$$

Taking logarithms from both sides, the latter inequality is equivalent to

$$\log(1/\delta)\log\frac{C_1 + C_2\log(1/\delta)}{\mathrm{e}(C_1 + C_2\log(1/\delta))} \leq \log\delta \,,$$

which turns into exact equality. $\square$

## A.3 Proof of Theorem 1

We first define explicitly the remainder term outlined in the statement of Theorem 1:

$$\Delta_1(n, a, \mathrm{C_A}, c_0) = \frac{n^{\gamma-1/2}}{c_0} + \frac{\mathrm{C_A}}{n^{(1-\gamma)/2}\sqrt{c_0 a}} + \frac{n^{2\gamma-3/2}}{ac_0^2} \ . \tag{29}$$

*Proof.* Since both terms in the right-hand side of the error bound of Proposition 4 scales linearly with $\sqrt{\kappa_Q}$, for simplicity we do not trace it in the subsequent bounds (i.e. assume $\kappa_Q = 1$), and then keep the required scaling with $\kappa_Q$ only in the final bounds. The decomposition (25) is a key element of our proof and allows to treat different error sources $D_1 - D_4$ separately. For the last iterate we have, using Proposition 4, that

$$\mathbb{E}^{1/2}\big[\|\theta_n - \theta^\star\|^2\big] \lesssim \frac{\|\varepsilon\|_\infty}{\sqrt{a}}\sqrt{\alpha_n} + \exp\bigg\{-(a/2)\sum_{\ell=1}^{n}\alpha_\ell\bigg\}\|\theta_0 - \theta^\star\|$$

$$\mathbb{E}^{1/2}\big[\|\theta_{2n} - \theta^\star\|^2\big] \lesssim \frac{\|\varepsilon\|_\infty}{\sqrt{a}}\sqrt{\alpha_{2n}} + \exp\bigg\{-(a/2)\sum_{\ell=1}^{2n}\alpha_\ell\bigg\}\|\theta_0 - \theta^\star\| \ .$$

Thus, using that $\sum_{k=1}^{n}\alpha_k \geq \frac{c_0(n^{1-\gamma}-1)}{1-\gamma}$ and $c_0 \leq 1 - \gamma$, we obtain that

$$\mathbb{E}^{1/2}\big[\|D_1\|^2\big] \lesssim \frac{\|\varepsilon\|_\infty}{\sqrt{ac_0}n^{(1-\gamma)/2}} + \frac{n^{\gamma-1/2}}{c_0}\exp\bigg\{-\frac{c_0 an^{1-\gamma}}{2(1-\gamma)}\bigg\}\|\theta_0 - \theta^\star\| \ ,$$

$$\mathbb{E}^{1/2}\big[\|D_2\|^2\big] \lesssim \frac{\|\varepsilon\|_\infty}{\sqrt{ac_0}n^{(1-\gamma)/2}} + \frac{n^{\gamma-1/2}}{c_0}\exp\bigg\{-\frac{c_0 a(2n)^{1-\gamma}}{1-\gamma}\bigg\}\|\theta_0 - \theta^\star\| \ .$$

Now we proceed with $D_3$. Since it is a sum of a martingale-difference sequence w.r.t. $\mathcal{F}_k = \sigma(Z_\ell, \ell \leq k)$, we get using Proposition 4, that

$$\mathbb{E}\big[\|D_3\|^2\big] \lesssim \frac{\mathrm{C_A^2}}{n}\sum_{k=n}^{2n-1}\mathbb{E}[\|\theta_k - \theta^\star\|^2]$$

$$\lesssim \frac{\mathrm{C_A^2}}{n}\sum_{k=n+1}^{2n}\frac{\|\varepsilon\|_\infty^2\alpha_k}{a} + \frac{\mathrm{C_A^2}}{n}\sum_{k=n+1}^{2n}\exp\bigg\{-a\sum_{\ell=1}^{k}\alpha_\ell\bigg\}\|\theta_0 - \theta^\star\|^2$$

$$\lesssim \frac{\mathrm{C_A^2}}{n}\sum_{k=n+1}^{2n}\frac{\|\varepsilon\|_\infty^2\alpha_k}{a} + \frac{\mathrm{C_A^2}}{n\alpha_{2n}}\exp\bigg\{-a\sum_{\ell=1}^{n}\alpha_\ell\bigg\}\underbrace{\sum_{k=n+1}^{2n}\alpha_k\exp\bigg\{-a\sum_{\ell=n+1}^{k}\alpha_\ell\bigg\}}_{S_1}\|\theta_0 - \theta^\star\|^2$$

$$\lesssim \frac{c_0\mathrm{C_A^2}\|\varepsilon\|_\infty^2}{a(1-\gamma)n^\gamma} + \frac{\mathrm{C_A^2}}{n^{1-\gamma}c_0 a}\exp\bigg\{-\frac{c_0 an^{1-\gamma}}{1-\gamma}\bigg\}\|\theta_0 - \theta^\star\|^2 \ ,$$

where we additionally used that $S_1 \lesssim 1/a$ due to Lemma 3. Now it remains to bound the term $D_4$ from the representation (25). Using Minkowski's inequality and Proposition 4, we get that

$$\mathbb{E}^{1/2}\big[\|D_4\|^2\big] \lesssim \frac{1}{\sqrt{n}}\sum_{k=n}^{2n-1}\mathbb{E}^{1/2}\big[\|\theta_k - \theta^\star\|^2\big]\left(\frac{1}{\alpha_k} - \frac{1}{\alpha_{k-1}}\right)$$

$$\lesssim \frac{1}{\sqrt{n}}\sum_{k=n}^{2n-1}\frac{\|\varepsilon\|_\infty(k^\gamma - (k-1)^\gamma)}{c_0\sqrt{a}}\sqrt{\alpha_k}$$

$$+ \frac{1}{c_0\sqrt{n}}\sum_{k=n}^{2n-1}(k^\gamma - (k-1)^\gamma)\exp\bigg\{-(a/2)\sum_{\ell=1}^{k}\alpha_\ell\bigg\}\|\theta_0 - \theta^\star\|$$

$$\overset{(a)}{\lesssim} \frac{\|\varepsilon\|_\infty}{\sqrt{ac_0}\sqrt{n}}\sum_{k=n}^{2n-1}\frac{1}{k^{1-\gamma/2}} + \frac{n^{2\gamma-3/2}}{ac_0^2}\exp\bigg\{-\frac{c_0 an^{1-\gamma}}{2(1-\gamma)}\bigg\}\|\theta_0 - \theta^\star\|$$

$$\lesssim \frac{\|\varepsilon\|_\infty}{\sqrt{ac_0}n^{(1-\gamma)/2}} + \frac{n^{2\gamma-3/2}}{ac_0^2}\exp\bigg\{-\frac{c_0 an^{1-\gamma}}{2(1-\gamma)}\bigg\}\|\theta_0 - \theta^\star\| \ .$$

Here in (a) we additionally used that $k^\gamma - (k-1)^\gamma \lesssim k^{1-\gamma}$ together with Lemma 3. Combining the estimates above yields the result of Theorem 1. $\qquad\square$

We conclude this section with some technical lemmas.

**Lemma 2** (Lemma 24 in [18]). *Let $b > 0$ and $(\alpha_k)_{k\geq 0}$ be a non-increasing sequence such that $\alpha_0 \leq 1/b$. Then*

$$\sum_{j=1}^{n+1} \alpha_j \prod_{l=j+1}^{n+1} (1 - \alpha_l b) = \frac{1}{b}\left\{ 1 - \prod_{l=1}^{n+1}(1 - \alpha_l b) \right\}$$

*Proof.* The proof of this statement is given in [18], we provide it here for completeness. Let us denote $u_{j:n+1} = \prod_{l=j}^{n+1}(1 - \alpha_l b)$. Then, for $j \in \{1, \ldots, n+1\}$, $u_{j+1:n+1} - u_{j:n+1} = b\alpha_j u_{j+1:n+1}$. Hence,

$$\sum_{j=1}^{n+1} \alpha_j \prod_{l=j+1}^{n+1} (1 - \alpha_l b) = \frac{1}{b}\sum_{j=1}^{n+1}(u_{j+1:n+1} - u_{j:n+1}) = b^{-1}(1 - u_{1:n+1}) \,,$$

and the statement follows. $\qquad\square$

**Lemma 3** (Modified Lemma 25 in [18]). *Let $b > 0$ and let $\alpha_\ell = c_0/\ell^\gamma$, $\gamma \in [1/2; 1)$, such that $c_0 \leq 1/b$. Then for any $n$ satisfying*

$$n \geq 2 + 2\left(\frac{2\gamma}{c_0 b}\right)^{1/(1-\gamma)} \,, \quad \text{and} \quad \frac{n^{1-\gamma}}{1 + \log(n)} \geq \frac{2\gamma(1-\gamma)}{c_0 b(1 - (1/2)^{1-\gamma})} \,, \tag{30}$$

*and any $k \geq n$, it holds that*

$$\sum_{j=1}^{k+1} \alpha_j^2 \prod_{\ell=j+1}^{k+1} (1 - \alpha_\ell b) \leq (4/b)\alpha_{k+1} \,.$$

*Proof.* From elementary algebra, we obtain that

$$\alpha_\ell - \alpha_{\ell+1} = \frac{c_0}{\ell^\gamma} - \frac{c_0}{(\ell+1)^\gamma} = \frac{c_0((1 + 1/\ell)^\gamma - 1)}{(\ell+1)^\gamma} \leq \frac{c_0}{(\ell+1)^\gamma}\frac{\gamma}{\ell} \,, \tag{31}$$

where we used the fact that $(1 + x)^\gamma \leq 1 + \gamma x$ for $\gamma \in [1/2; 1)$ and $x \in [0, 1]$. Hence,

$$\frac{\alpha_\ell}{\alpha_{\ell+1}} \leq 1 + \frac{\gamma}{\ell} \,.$$

Thus we obtain that, since $k \geq n$,

$$\sum_{j=1}^{k+1}\alpha_j^2 \prod_{\ell=j+1}^{k+1} (1 - \alpha_\ell b) = \alpha_{k+1}\sum_{j=1}^{k+1}\alpha_j \prod_{\ell=j+1}^{k+1} \left(\frac{\alpha_{\ell-1}}{\alpha_\ell}\right)(1 - \alpha_\ell b)$$

$$\leq \alpha_{k+1}\sum_{j=1}^{k+1}\alpha_j \prod_{\ell=j+1}^{k+1} \left(1 + \frac{\gamma}{\ell-1}\right)(1 - \alpha_\ell b)$$

$$\leq \alpha_{k+1}\sum_{j=1}^{k+1}\alpha_j \exp\left\{\sum_{\ell=j+1}^{n}\frac{\gamma}{\ell-1}\right\}\exp\left\{-\sum_{\ell=j+1}^{n}\alpha_\ell b\right\}\exp\left\{\sum_{\ell=n+1}^{k+1}\frac{\gamma}{\ell-1}\right\}\exp\left\{-\sum_{\ell=n+1}^{k+1}\alpha_\ell b\right\}$$

$$\leq \alpha_{k+1}\sum_{j=1}^{k+1}\alpha_j \exp\left\{\sum_{\ell=j+1}^{n}\frac{\gamma}{\ell-1}\right\}\exp\left\{-\sum_{\ell=j+1}^{n}\alpha_\ell b\right\}\exp\left\{-\frac{b}{2}\sum_{\ell=n+1}^{k+1}\alpha_\ell b\right\} \,.$$

In the last identity we used the fact that, since $n$ satisfies (30), it holds for $\ell \geq n/2$ that

$$\frac{\gamma}{\ell-1} \leq \alpha_\ell b/2 \,. \tag{32}$$

We will now prove that for $j \leq n - 1$, it holds

$$\sum_{\ell=j+1}^{n} \frac{\gamma}{\ell - 1} \leq (b/2) \sum_{\ell=j+1}^{n} \alpha_\ell \,, \tag{33}$$

For $j \geq n/2$, the bound (33) directly follows from (32). We now turn to the proof of (33) for $j \leq \lceil n/2 \rceil$. Note first that

$$\sum_{\ell=j+1}^{n} \frac{\gamma}{\ell - 1} \leq \sum_{\ell=2}^{n} \frac{\gamma}{\ell - 1} \leq \gamma(\log(n) + 1) \,.$$

On the other hand, we get

$$\sum_{\ell=j+1}^{n} \frac{1}{\ell^\gamma} \geq \int_{j+1}^{n+1} \frac{\mathrm{d}x}{x^\gamma} = \frac{\left((n+1)^{1-\gamma} - (j+1)^{1-\gamma}\right)}{1 - \gamma} \,.$$

Comparing the above bounds, to ensure that (33) holds, it is enough to check that

$$\gamma(1 + \log(n)) \leq \frac{c_0 b}{2(1 - \gamma)} \left(n^{1-\gamma} - (n/2)^{1-\gamma}\right). \tag{34}$$

Note that (34) is guaranteed by (30). Using that $e^{-x} \leq 1 - x/2$ for $x \in [0, 1]$, we obtain that

$$\sum_{j=1}^{k+1} \alpha_j^2 \prod_{\ell=j+1}^{k+1} (1 - \alpha_\ell b) \leq \alpha_{k+1} \sum_{j=1}^{k+1} \alpha_j \exp\left\{-(b/2) \sum_{\ell=j+1}^{k+1} \alpha_\ell\right\}$$

$$\leq \alpha_{k+1} \sum_{j=1}^{k+1} \alpha_j \prod_{\ell=j+1}^{k+1} \left(1 - (b/4)\alpha_\ell\right)$$

$$\leq (4/b) \, \alpha_{k+1} \,,$$

where the last inequality follows from Lemma 2. $\qquad\square$

## B  Proof of Theorem 2

We first define explicitly the remainder term outlined in the statement of Theorem 2:

$$\Delta_2(n, a, \mathrm{C_A}, \mathrm{Tr}\,\Sigma_\varepsilon, c_0) = \kappa_Q \left(\sqrt{\mathrm{Tr}\,\Sigma_\varepsilon}\Delta_1 + \frac{\sqrt{\mathrm{Tr}\,\Sigma_\varepsilon}(\mathrm{C_A} \vee 1)^2 n^{\gamma-1/2}}{ac_0}\right) \,, \tag{35}$$

and constants $\mathsf{C}_1, \mathsf{C}_2, \mathsf{C}_3, \mathsf{C}_4$ from Theorem 2, optimized bound (15), and Remark 1, respectively:

$$\begin{aligned}
\mathsf{C}_1 &= \frac{\sqrt{\kappa_Q}\|\varepsilon\|_\infty\sqrt{\mathrm{Tr}\,\Sigma_\varepsilon}}{\sqrt{ac_0}} + \frac{\kappa_Q(\mathrm{Tr}\,\Sigma_\varepsilon + \mathrm{C_A}\sqrt{\mathrm{Tr}\,\Sigma_\varepsilon}\|\varepsilon\|_\infty)}{ac_0} \,, \\
\mathsf{C}_2 &= \frac{\sqrt{\kappa_Q}\|\varepsilon\|_\infty\sqrt{\mathrm{Tr}\,\Sigma_\varepsilon}c_0\,\mathrm{C_A}}{\sqrt{ac_0(1 - \gamma)}} + \kappa_Q\,\mathrm{C_A}\sqrt{\mathrm{Tr}\,\Sigma_\varepsilon}(\|\varepsilon\|_\infty + \sqrt{\mathrm{Tr}\,\Sigma_\varepsilon} + \mathrm{C_A}\|\varepsilon\|_\infty) \,, \\
\mathsf{C}_3 &= \frac{\kappa_Q(c_0\,\mathrm{C_A} \vee 1)\sqrt{\mathrm{Tr}\,\Sigma_\varepsilon}(\|\varepsilon\|_\infty + \sqrt{\mathrm{Tr}\,\Sigma_\varepsilon} + \mathrm{C_A}\|\varepsilon\|_\infty)}{ac_0} \,, \\
\mathsf{C}_4 &= \frac{\kappa_Q(c_0\,\mathrm{C_A} \vee 1)\sqrt{\mathrm{Tr}\,\Sigma_\varepsilon^{(\Pi)}}(\|\varepsilon\|_\infty + \sqrt{\mathrm{Tr}\,\Sigma_\varepsilon^{(\Pi)}} + \mathrm{C_A}\|\varepsilon\|_\infty)}{ac_0} \,.
\end{aligned} \tag{36}$$

To complete the proof we only need to combine (13) with the bounds of Theorem 1. Note that we apply (13) with

$$\xi_\ell = \frac{\varepsilon_\ell}{\sqrt{n}} \,.$$

Thus, for $\Upsilon_n$ defined in (13) we have

$$\Upsilon_n \leq \frac{\|\varepsilon\|_\infty^3}{n^{1/2}} \,.$$

Applying the Cauchy-Schwartz inequality, we get

$$\mathbb{E}[\|D\|\|W\|] \leq \mathbb{E}^{1/2}[\|D\|^2]\mathbb{E}^{1/2}[\|W\|^2] \lesssim \frac{\sqrt{\kappa_Q}\|\varepsilon\|_\infty\sqrt{\operatorname{Tr}\Sigma_\varepsilon}}{\sqrt{ac_0}}\left(\frac{1}{n^{(1-\gamma)/2}} + \frac{c_0\,\mathrm{C_A}}{\sqrt{1-\gamma}n^{\gamma/2}}\right)$$
$$+ \sqrt{\kappa_Q}\sqrt{\operatorname{Tr}\Sigma_\varepsilon}\Delta_1\exp\left\{-\frac{c_0 a n^{1-\gamma}}{2(1-\gamma)}\right\}\|\theta_0 - \theta^\star\|\;.$$

Now it remains to bound the last term in (13). Using the Cauchy-Schwartz inequality and Lemma 4, we obtain that

$$n^{-1/2}\mathbb{E}[\sum_{i=n}^{2n-1}\|\varepsilon_i\|\|D - D^{(i)}\|] \leq n^{-1/2}\mathbb{E}^{1/2}[\|\varepsilon_1\|^2]\sum_{i=n}^{2n-1}\mathbb{E}^{1/2}[\|D - D^{(i)}\|^2]$$
$$\lesssim \frac{\kappa_Q(\operatorname{Tr}\Sigma_\varepsilon + \mathrm{C_A}\sqrt{\operatorname{Tr}\Sigma_\varepsilon}\|\varepsilon\|_\infty)}{ac_0 n^{1-\gamma}} + \frac{\kappa_Q\,\mathrm{C_A}\sqrt{\operatorname{Tr}\Sigma_\varepsilon}(\|\varepsilon\|_\infty + \sqrt{\operatorname{Tr}\Sigma_\varepsilon} + \mathrm{C_A}\|\varepsilon\|_\infty)}{n^{\gamma/2}}$$
$$+ \frac{\kappa_Q\sqrt{\operatorname{Tr}\Sigma_\varepsilon}(\mathrm{C_A}\vee 1)^2 n^{\gamma-1/2}}{ac_0}\exp\left\{-\frac{c_0 a n^{1-\gamma}}{2(1-\gamma)}\right\}\|\theta_0 - \theta^\star\|\;,$$

and the statement follows from [63, Corollary 2.3].

## B.1 Proof of auxiliary lemmas for Theorem 2.

Our proof of Theorem 2 is based on the key lemma below, which allows us to bound $\mathbb{E}^{1/2}[\|D - D^{(i)}\|^2]$ for $i \in \{n+1, \ldots, 2n\}$.

**Lemma 4.** *Assume A1, A2, and A3. Then*

$$\sum_{i=n+1}^{2n}\mathbb{E}^{1/2}[\|D - D^{(i)}\|^2] \lesssim \frac{\kappa_Q(\sqrt{\operatorname{Tr}\Sigma_\varepsilon} + \mathrm{C_A}\|\varepsilon\|_\infty)}{ac_0}n^{\gamma-1/2}$$
$$+ \kappa_Q\,\mathrm{C_A}\left(\|\varepsilon\|_\infty + \sqrt{\operatorname{Tr}\Sigma_\varepsilon} + \mathrm{C_A}\|\varepsilon\|_\infty\right)n^{\frac{1-\gamma}{2}}$$
$$+ \frac{\kappa_Q(\mathrm{C_A}\vee 1)^2 n^{\gamma-1/2}}{ac_0}\exp\left\{-\frac{c_0 a n^{1-\gamma}}{2(1-\gamma)}\right\}\|\theta_0 - \theta^\star\|\;.$$

*Proof.* Since both terms in the right-hand side of the error bound of Proposition 4 scales linearly with $\sqrt{\kappa_Q}$, for simplicity we do not trace it in the subsequent bounds (i.e. assume $\kappa_Q = 1$), and then keep the required scaling with $\kappa_Q$ only in the final bounds. Consider the sequences of noise variables

$$(Z_1, \ldots, Z_{i-1}, Z_i, Z_{i+1}, \ldots, Z_{2n}) \text{ and } (Z_1, \ldots, Z_{i-1}, Z_i', Z_{i+1}, \ldots, Z_{2n})\;,$$

which differ only in position $i$, $n+1 \leq i \leq 2n$, with $Z_i'$ being an independent copy of $Z_i$. Consider the associated SA processes

$$\theta_k = \theta_{k-1} - \alpha_k\{\mathbf{A}(Z_k)\theta_{k-1} - \mathbf{b}(Z_k)\}\;, \quad k \geq 1\;, \quad \theta_0 = \theta_0 \in \mathbb{R}^d$$
$$\theta_k^{(i)} = \theta_{k-1}^{(i)} - \alpha_k\{\mathbf{A}(Y_k)\theta_{k-1}^{(i)} - \mathbf{b}(Y_k)\}\;, \quad k \geq 1\;, \quad \theta_0^{(i)} = \theta_0 \in \mathbb{R}^d\;, \tag{37}$$

where $Y_k = Z_k$ for $k \neq i$ and $Y_i = Z_i'$. From the above representations we easily observe that $\theta_k = \theta_k^{(i)}$ for $k < i$, moreover,

$$\theta_i - \theta_i^{(i)} = \alpha_i\{(\mathbf{A}(Z_i') - \mathbf{A}(Z_i))\theta_{i-1} - \mathbf{b}(Z_i') + \mathbf{b}(Z_i)\}$$
$$= \alpha_i(\mathbf{A}(Z_i') - \mathbf{A}(Z_i))(\theta_{i-1} - \theta^\star) - \alpha_i(\varepsilon_i - \varepsilon_i')\;, \tag{38}$$

where $\varepsilon_i = \varepsilon(Z_i)$ and $\varepsilon_i' = \varepsilon(Z_i')$. Representation (38) implies, together with Proposition 4 and $c_0 \leq a$, that

$$\mathbb{E}^{1/2}[\|\theta_i - \theta_i^{(i)}\|^2] \lesssim \alpha_i\sqrt{\operatorname{Tr}\Sigma_\varepsilon} + \frac{\mathrm{C_A}\|\varepsilon\|_\infty\alpha_i^{3/2}}{\sqrt{a}} + \alpha_i\,\mathrm{C_A}\exp\left\{-\frac{a}{2}\sum_{j=1}^{i-1}\alpha_j\right\}\|\theta_0 - \theta^\star\|$$
$$\lesssim \alpha_i(\sqrt{\operatorname{Tr}\Sigma_\varepsilon} + \mathrm{C_A}\|\varepsilon\|_\infty) + \alpha_i\,\mathrm{C_A}\exp\left\{-\frac{a}{2}\sum_{j=1}^{i-1}\alpha_j\right\}\|\theta_0 - \theta^\star\|\;. \tag{39}$$

Moreover, for any $j > i$ one observes, expanding (37), that

$$\theta_j - \theta_j^{(i)} = \left\{ \prod_{k=i+1}^{j} (\mathrm{I} - \alpha_k \mathbf{A}(Z_k)) \right\} (\theta_i - \theta_i^{(i)}) . \tag{40}$$

We use the above representations to estimate $\mathbb{E}^{1/2}[\|D - D^{(i)}\|^2]$. Using Minkowski's inequality,

$$\mathbb{E}^{1/2}[\|D - D^{(i)}\|^2] \leq \sum_{j=1}^{4} \mathbb{E}^{1/2}[\|D_j - D_j^{(i)}\|^2] , \tag{41}$$

and bound the respective differences separately. Recall that here $D_1 - D_4$ are defined in (25), and $D_1^{(i)} - D_4^{(i)}$ are their respective counterparts with $Z_i$ substituted with $Z_i'$. First we note that the term $D_1 = D_1^{(i)}$ for any $n + 1 \leq i \leq 2n$. Next, using (40) and (39), we get

$$
\begin{aligned}
\mathbb{E}^{1/2}[\|D_2 - D_2^{(i)}\|^2] &= \frac{1}{\sqrt{n}\alpha_{2n}} \mathbb{E}^{1/2}[\|\theta_{2n} - \theta_{2n}^{(i)}\|^2] \\
&\leq \frac{1}{\sqrt{n}\alpha_{2n}} \mathbb{E}^{1/2}\Big[\|\prod_{k=i+1}^{2n} (\mathrm{I} - \alpha_k \mathbf{A}_k)\|^2\Big] \mathbb{E}^{1/2}[\|\theta_i - \theta_i^{(i)}\|^2] \\
&\stackrel{(a)}{\lesssim} \frac{\alpha_i(\sqrt{\mathrm{Tr}\,\Sigma_\varepsilon} + \mathrm{C_A}\,\|\varepsilon\|_\infty)}{\sqrt{n}\alpha_{2n}} \exp\Big\{-\frac{a}{2} \sum_{k=i+1}^{2n} \alpha_k\Big\} \\
&\quad + \frac{\alpha_i\,\mathrm{C_A}}{\sqrt{n}\alpha_{2n}} \exp\Big\{-\frac{a}{2} \sum_{k=1}^{2n} \alpha_k\Big\} \|\theta_0 - \theta^\star\| .
\end{aligned}
$$

In the inequality (a) above we additionally used the stability of matrix product introduced from Corollary 4. Summing the above inequality for $i = n + 1$ to $2n$ and applying Lemma 2, we get

$$
\begin{aligned}
\sum_{i=n+1}^{2n} \mathbb{E}^{1/2}[\|D_2 - D_2^{(i)}\|^2] &\lesssim \frac{\sqrt{\mathrm{Tr}\,\Sigma_\varepsilon} + \mathrm{C_A}\,\|\varepsilon\|_\infty}{a\sqrt{n}\alpha_{2n}} + \frac{\mathrm{C_A}}{a\sqrt{n}\alpha_{2n}} \exp\Big\{-\frac{a}{2} \sum_{k=1}^{n} \alpha_k\Big\} \|\theta_0 - \theta^\star\| \\
&\lesssim \frac{\sqrt{\mathrm{Tr}\,\Sigma_\varepsilon} + \mathrm{C_A}\,\|\varepsilon\|_\infty}{a c_0} n^{\gamma - 1/2} + \frac{\mathrm{C_A}\, n^{\gamma - 1/2}}{a c_0} \exp\Big\{-\frac{c_0 a n^{1-\gamma}}{2(1-\gamma)}\Big\} \|\theta_0 - \theta^\star\| . \tag{42}
\end{aligned}
$$

Now we proceed with the difference $D_3 - D_3^{(i)}$. Using (25), we get

$$D_3 - D_3^{(i)} = \frac{1}{\sqrt{n}} (\mathbf{A}_i - \mathbf{A}_i')(\theta_{i-1} - \theta^\star) + \frac{1}{\sqrt{n}} \sum_{k=i+1}^{2n} (\mathbf{A}_k - \bar{\mathbf{A}})(\theta_{k-1} - \theta_{k-1}^{(i)}) .$$

The expression above is a sum of martingale-difference terms w.r.t. filtration $\mathcal{F}_k' = \sigma(Z_i', Z_\ell, \ell \leq k)$. Hence, we get, using (40) and Proposition 4, that

$$
\begin{aligned}
\mathbb{E}[\|D_3 - D_3^{(i)}\|^2] &\lesssim \frac{\mathrm{C_A^2}}{n} \mathbb{E}[\|\theta_{i-1} - \theta^\star\|^2] + \frac{\mathrm{C_A^2}}{n} \sum_{k=i+1}^{2n} \mathbb{E}[\|\theta_{k-1} - \theta_{k-1}^{(i)}\|^2] \tag{43} \\
&\lesssim \frac{\mathrm{C_A^2}\,\|\varepsilon\|_\infty^2 \alpha_i}{na} + \frac{\mathrm{C_A^2}\,\|\theta_0 - \theta^\star\|^2}{n} \exp\Big\{-a \sum_{j=1}^{i-1} \alpha_j\Big\} \\
&\quad + \frac{\mathrm{C_A^2}}{n} \mathbb{E}[\|\theta_i - \theta_i^{(i)}\|^2] \sum_{k=i+1}^{2n} \exp\Big\{-a \sum_{j=i+1}^{k-1} \alpha_j\Big\} .
\end{aligned}
$$

Using now the bound (39), we obtain that

$$\mathbb{E}[\|\theta_i - \theta_i^{(i)}\|^2] \sum_{k=i+1}^{2n} \exp\{-a \sum_{j=i+1}^{k-1} \alpha_j\}$$

$$\lesssim \alpha_i^2 \left(\operatorname{Tr}\Sigma_\varepsilon + \mathrm{C}_{\mathbf{A}}^2 \|\varepsilon\|_\infty^2\right) \sum_{k=i+1}^{2n} \exp\{-a \sum_{j=i+1}^{k-1} \alpha_j\} + \alpha_i^2 \, \mathrm{C}_{\mathbf{A}}^2 \|\theta_0 - \theta^\star\|^2 \sum_{k=i+1}^{2n} \exp\{-a \sum_{j=1}^{k-1} \alpha_i\}$$

$$\lesssim \frac{\alpha_i^2}{\alpha_{2n}} \left(\operatorname{Tr}\Sigma_\varepsilon + \mathrm{C}_{\mathbf{A}}^2 \|\varepsilon\|_\infty^2\right) \sum_{k=i+1}^{2n} \alpha_k \exp\{-a \sum_{j=i+1}^{k-1} \alpha_j\} + \frac{\alpha_i^2 \, \mathrm{C}_{\mathbf{A}}^2}{\alpha_{2n}} \|\theta_0 - \theta^\star\|^2 \sum_{k=i+1}^{2n} \alpha_k \exp\{-a \sum_{j=1}^{k-1} \alpha_j\}$$

$$\overset{(a)}{\lesssim} \frac{\alpha_i^2 (\operatorname{Tr}\Sigma_\varepsilon + \mathrm{C}_{\mathbf{A}}^2 \|\varepsilon\|_\infty^2)}{a\alpha_{2n}} + \frac{\alpha_i^2 \, \mathrm{C}_{\mathbf{A}}^2}{a\alpha_{2n}} \|\theta_0 - \theta^\star\|^2 \exp\{-a \sum_{j=1}^{i-1} \alpha_j\} \; .$$

In the above formula in (a) we additionally used that, since $\alpha_i a \leq 1/2$,

$$\sum_{k=i+1}^{2n} \alpha_k \exp\{-a \sum_{j=i+1}^{k-1} \alpha_j\} \lesssim \int_0^{+\infty} \exp\{-ax\}\,dx = \frac{1}{a} \; . \tag{44}$$

Hence, combining everything in (43), and using additionally that $\alpha_i \leq a$, we get

$$\mathbb{E}^{1/2}[\|D_3 - D_3^{(i)}\|^2] \lesssim \frac{\mathrm{C}_{\mathbf{A}}}{\sqrt{na}} \left(\|\varepsilon\|_\infty \sqrt{\alpha_i} + \frac{\alpha_i(\sqrt{\operatorname{Tr}\Sigma_\varepsilon} + \mathrm{C}_{\mathbf{A}} \|\varepsilon\|_\infty)}{\sqrt{\alpha_{2n}}}\right) +$$

$$\frac{\mathrm{C}_{\mathbf{A}}}{\sqrt{n}} \left(1 + \frac{\alpha_i \, \mathrm{C}_{\mathbf{A}}}{\sqrt{a\alpha_{2n}}}\right) \exp\{-\frac{a}{2} \sum_{j=1}^{i-1} \alpha_i\} \|\theta_0 - \theta^\star\| \; .$$

Summing the above inequality for $i = n+1$ to $2n$, and using that $\alpha_k = c_0/k^\gamma$, we get

$$\sum_{i=n+1}^{2n} \sqrt{\alpha_i} \lesssim \sqrt{c_0} n^{1-\gamma/2} \; , \qquad \sum_{i=n+1}^{2n} \frac{\alpha_i}{\sqrt{\alpha_{2n}}} \lesssim \sqrt{c_0} n^{1-\gamma/2} \; ,$$

and, hence, using again $\alpha_i \leq a$, we get

$$\sum_{i=n+1}^{2n} \mathbb{E}^{1/2}[\|D_3 - D_3^{(i)}\|^2]$$

$$\lesssim \frac{\mathrm{C}_{\mathbf{A}} \sqrt{c_0}}{\sqrt{a}} \left(\|\varepsilon\|_\infty + \sqrt{\operatorname{Tr}\Sigma_\varepsilon} + \mathrm{C}_{\mathbf{A}} \|\varepsilon\|_\infty\right) n^{\frac{1-\gamma}{2}} + \frac{(\mathrm{C}_{\mathbf{A}} \vee 1)^2 \|\theta_0 - \theta^\star\|}{\sqrt{n}\alpha_{2n}} \sum_{i=n+1}^{2n} \alpha_i \exp\{-\frac{a}{2} \sum_{j=1}^{i-1} \alpha_j\}$$

$$\lesssim \frac{\mathrm{C}_{\mathbf{A}} \sqrt{c_0}}{\sqrt{a}} \left(\|\varepsilon\|_\infty + \sqrt{\operatorname{Tr}\Sigma_\varepsilon} + \mathrm{C}_{\mathbf{A}} \|\varepsilon\|_\infty\right) n^{\frac{1-\gamma}{2}} + \frac{n^{\gamma-1/2}(\mathrm{C}_{\mathbf{A}} \vee 1)^2 \|\theta_0 - \theta^\star\|}{ac_0} \exp\left\{-\frac{c_0 a n^{1-\gamma}}{2(1-\gamma)}\right\} \; ,$$

where for the last identity we used the fact that $\alpha_k = c_0/k^\gamma$, and (44). It remains to upper bound the difference $D_4 - D_4^{(i)}$. Note first that, proceeding as in (31), we get

$$\alpha_{k-1} - \alpha_k \leq \frac{\gamma}{k-1} \frac{1}{\alpha_{k-1}} \lesssim \frac{1}{(k-1)^{1-\gamma}} \; .$$

Using now the definition of $D_4$ in (25), we have that

$$\mathbb{E}^{1/2}[\|D_4 - D_4^{(i)}\|^2] = \frac{1}{\sqrt{n}} \mathbb{E}^{1/2}[\| \sum_{k=i+1}^{2n} (\theta_{k-1} - \theta_{k-1}^{(i)}) \left(\frac{1}{\alpha_k} - \frac{1}{\alpha_{k-1}}\right) \|^2]$$

$$\leq \frac{1}{\sqrt{n}} \mathbb{E}^{1/2}[\|\theta_i - \theta_i^{(i)}\|^2] \sum_{k=i+1}^{2n} \left(\frac{1}{\alpha_k} - \frac{1}{\alpha_{k-1}}\right) \exp\{-\frac{a}{2} \sum_{j=i+1}^{k-1} \alpha_j\} \; .$$

Hence, using the bound (39) and taking sum for $i = n+1$ to $2n$,

$$\sum_{i=n+1}^{2n} \mathbb{E}^{1/2}[\|D_4 - D_4^{(i)}\|^2] \lesssim \left(\frac{\sqrt{\operatorname{Tr}\Sigma_\varepsilon} + \mathrm{C_A}\|\varepsilon\|_\infty}{\sqrt{n}}\right) \sum_{i=n+1}^{2n} \alpha_i \sum_{k=i+1}^{2n} \left(\frac{1}{\alpha_k} - \frac{1}{\alpha_{k-1}}\right) \exp\{-a\sum_{j=i+1}^{k-1} \alpha_j\}$$

$$+ \frac{\mathrm{C_A}\|\theta_0 - \theta^\star\|}{\sqrt{n}} \sum_{i=n+1}^{2n} \alpha_i \sum_{k=i+1}^{2n} \left(\frac{1}{\alpha_k} - \frac{1}{\alpha_{k-1}}\right) \exp\{-\frac{a}{2}\sum_{j=1}^{k-1} \alpha_j\} .$$

Changing now the summation order, we obtain that

$$\sum_{i=n+1}^{2n} \alpha_i \sum_{k=i+1}^{2n} \left(\frac{1}{\alpha_k} - \frac{1}{\alpha_{k-1}}\right) \exp\{-a\sum_{j=i+1}^{k-1} \alpha_j\} \lesssim \frac{1}{a} \sum_{k=n+2}^{2n} \left(\frac{1}{\alpha_k} - \frac{1}{\alpha_{k-1}}\right)$$

$$= \frac{1}{a}\left(\frac{1}{\alpha_{n+1}} - \frac{1}{\alpha_{2n}}\right) \lesssim \frac{1}{a\alpha_{2n}} .$$

Hence, combining the above bounds, we get

$$\sum_{i=n+1}^{2n} \mathbb{E}^{1/2}[\|D_4 - D_4^{(i)}\|^2] \lesssim \frac{\sqrt{\operatorname{Tr}\Sigma_\varepsilon} + \mathrm{C_A}\|\varepsilon\|_\infty}{\sqrt{n}a\alpha_{2n}} + \sum_{i=n+1}^{2n} \mathbb{E}^{1/2}[\|D_4 - D_4^{(i)}\|^2] \qquad (45)$$

$$\lesssim \frac{\sqrt{\operatorname{Tr}\Sigma_\varepsilon} + \mathrm{C_A}\|\varepsilon\|_\infty}{ac_0} n^{\gamma-1/2} + \frac{\mathrm{C_A}\, n^{\gamma-1/2}}{ac_0} \exp\left\{-\frac{c_0 a n^{1-\gamma}}{2(1-\gamma)}\right\}\|\theta_0 - \theta^\star\| .$$

It remains now to combine (42), (43), and (45) in (41) and use that $c_0 \leq a$. $\qquad\square$

## B.2 Relations between $\rho_n^{\mathrm{Conv}}$ and integral probability metrics

In this section we closely follow the exposition outlined in [24]. Consider two $\mathbb{R}^d$-valued random variables $X$ and $Y$. Then the integral probability metric [75], associated with the class of functions $\mathcal{H} = \{h : \mathbb{R}^d \to \mathbb{R}, \mathbb{E}[|h(X)|] < \infty, \mathbb{E}[|h(Y)|] < \infty\}$, is defined as

$$\mathsf{d}_\mathcal{H}(X,Y) = \sup_{h\in\mathcal{H}}\left|\mathbb{E}[h(X)] - \mathbb{E}[h(Y)]\right| .$$

Different choices of $\mathcal{H}$ induce different metrics, in particular, we mention the following:

$$\mathcal{H}_K = \{\mathbf{1}_{x\leq u}, \quad u = (u_1, \ldots, u_d) \in \mathbb{R}^d\}$$
$$\mathcal{H}_{Conv} = \{\mathbf{1}_{x\in B}, \quad B \in \operatorname{Conv}(\mathbb{R}^d)\}$$
$$\mathcal{H}_W = \{h : \mathbb{R}^d \to \mathbb{R}, \quad \|h\|_{\mathrm{Lip}} \leq 1\}$$
$$\mathcal{H}_{[m]} = \{h : \mathbb{R}^d \to \mathbb{R}, \quad h^{m-1} \text{ is Lipschitz with } |h|_j \leq 1 , \quad 1 \leq j \leq m\} ,$$

where $\operatorname{Conv}(\mathbb{R}^d)$ refers to the set of convex sets in $\mathbb{R}^d$, $\|h\|_{\mathrm{Lip}} = \sup_{x\neq y}\frac{\|h(x)-h(y)\|}{\|x-y\|}$, and the quantity $|h|_j$ is defined as

$$|h|_j = \max_{i_1,\ldots,i_j\in\{1,\ldots,d\}} \|\frac{\partial^j h(u)}{\partial u_{i_1}\ldots\partial u_{i_j}}\|_\infty .$$

In other words, for $m \in \mathbb{N}$, the class $\mathcal{H}_{[m]}$ corresponds to the functions with bounded derivatives up to the $(m-1)$-th order. The class $\mathcal{H}_K$ induces the Kolmogorov distance between distributions [75], class $\mathcal{H}_{Conv}$ induces the metric $\rho_n^{\mathrm{Conv}}$ defined in (2), which is the main object of studies in the current paper. Class $\mathcal{H}_W$ induces the celebrated Wasserstein distance, and classes $\mathcal{H}_{[m]}$ induce smoothed Wasserstein distances. We will denote the respective metrics by $\mathsf{d}_K, \rho_n^{\mathrm{Conv}}, \mathsf{d}_W$, and $\mathsf{d}_{[m]}$, respectively. Then, obviously,

$$\mathsf{d}_K(X,Y) \leq \rho_n^{\mathrm{Conv}}(X,Y)$$

for any random vectors $X$ and $Y$. Other relations are more involved. When $Y$ is a multivariate normal vector, it is known (see e.g. [49]) that

$$\rho_n^{\mathrm{Conv}}(X,Y) \leq C\sqrt{\mathsf{d}_W(X,Y)} ,$$

where the constant $C$ in the above inequality depends on the covariance matrix of vector $Y$. This inequality justifies comparison of our bounds of Theorem 2 with the result of [65]. The authors in [2] considered integral probability metric $\mathsf{d}_{[2]}$ and obtained rate of convergence

$$\mathsf{d}_{[2]}(\sqrt{n}(\bar{\theta}_n - \theta^\star), Y) \leq \frac{C_1}{\sqrt{n}} \, ,$$

where $Y \sim \mathcal{N}(0, \Sigma_\infty)$, and $C_1$ in the above inequality stands for a constant depending upon problem dimension $d$ and other instance-dependent parameters from A2. Applying the result of [24, Proposition 2.6] yields

$$\mathsf{d}_K(\sqrt{n}(\bar{\theta}_n - \theta^\star), Y) \lesssim \left(\mathsf{d}_{[2]}(\sqrt{n}(\bar{\theta}_n - \theta^\star), Y)\right)^{1/3} \lesssim \frac{1}{n^{1/6}} \, .$$

Thus, the result of [2] implies rate of convergence of $\sqrt{n}(\bar{\theta}_n - \theta^\star)$ to normal law $\mathcal{N}(0, \Sigma_\infty)$ of order $n^{-1/6}$ in a sense of Kolmogorov distance $\mathsf{d}_K$. Our result of Theorem 2 implies the respective rate of order $n^{-1/4}$. At the same time, it is not clear if $\rho_n^{\mathrm{Conv}}$ can be directly related to $\mathsf{d}_{[2]}$.

## C  Bootstrap validity proof

### C.1  Proof of Theorem 3

We first define explicitly the remainder term $\Delta_3$ outlined in the statement of Theorem 3, that is,

$$\Delta_3(n, a, \mathrm{C_A}, \|\varepsilon\|_\infty) = \frac{\kappa_Q^{3/2}(\mathrm{C_A^3} \vee 1)\|\varepsilon\|_\infty n^{1/4}\sqrt{\log n}}{a^{3/2}} \, . \tag{46}$$

In the above bounds we do not trace the precise dependence on the constant $c_0$ from the definition of the step size. We now define the following sets, with the convention $\alpha_\ell = c_0/\sqrt{\ell}$:

$$\Omega_1 = \left\{\forall k \in [n, 2n-1] : \|\theta_k - \theta^\star\| \geq \sqrt{\kappa_Q}\mathrm{e}^2 \exp\left\{-\frac{a}{2}\sum_{\ell=1}^{k} \alpha_\ell\right\}\|\theta_0 - \theta^\star\| \right. \tag{47}$$
$$\left. + \frac{8\mathrm{e}^2\sqrt{\kappa_Q}\|\varepsilon\|_\infty \log n}{\sqrt{a}}\sqrt{\alpha_k}\right\} \, ,$$

$$\Omega_2 = \left\{n+1 \leq m \leq k \leq 2n : \ \|\Gamma_{m:k}\| \leq \sqrt{\kappa_Q}\mathrm{e}^2 \prod_{j=m}^{k}\left(1 - \frac{a\alpha_j}{4}\right)\right\} \, ,$$

$$\Omega_3 = \left\{\|\Sigma_\varepsilon^{-1/2}\Sigma_\varepsilon^{\mathrm{b}}\Sigma_\varepsilon^{-1/2} - \mathrm{I}\| \leq 4\|\varepsilon\|_\infty\sqrt{\frac{\log(n)}{\sigma n}} + \frac{4(1 + \|\varepsilon\|_\infty^2/\sigma^2)\log(n)}{n}\right\} \, ,$$

$$\Omega_4 = \left\{\forall \ell \in [n, 2n-1] : \ \left\|\sum_{k=\ell+1}^{2n}(\mathbf{A}_k - \bar{\mathbf{A}})\Gamma_{\ell+1:k-1}\right\| \leq \frac{8\,\mathrm{C_A}\sqrt{\kappa_Q}\mathrm{e}^2\sqrt{\log n}}{\sqrt{a\alpha_\ell}} + 6\,\mathrm{C_A}\sqrt{\kappa_Q}\mathrm{e}\log n\right\} \, ,$$

$$\Omega_5 = \left\{\forall h \in [1; n]\, , \ \forall m \in [n, 2n-h] : \ \|\sum_{\ell=m+1}^{m+h}\alpha_\ell(\mathbf{A}_\ell - \bar{\mathbf{A}})\|_Q \leq 2\,\mathrm{C_A}\sqrt{\kappa_Q}\sqrt{\sum_{\ell=m+1}^{m+h}\alpha_\ell^2\log(2n^4)}\right\} \, ,$$

$$\Omega_6 = \left\{\|\Sigma_\varepsilon^{\mathrm{b}} - \Sigma_\varepsilon\| \leq 4\|\varepsilon\|_\infty\sqrt{\frac{\|\Sigma_\varepsilon\| \log(n)}{n}} + \frac{4(\|\Sigma_\varepsilon\| + \|\varepsilon\|_\infty^2)\log(n)}{n}\right\} \, ,$$

Then, due to Corollary 2, we have that $\mathbb{P}(\Omega_1) \geq 1 - \frac{1}{n}$. Similarly, due to Corollary 5, $\mathbb{P}(\Omega_2) \geq 1 - \frac{1}{n}$. The bounds on $\mathbb{P}(\Omega_3)$ and $\mathbb{P}(\Omega_4)$ follows from Lemma 5 and Lemma 6, respectively. Similarly, Proposition 7 implies that $\mathbb{P}(\Omega_5) \geq 1 - \frac{1}{n}$. Hence, based on the sets above, we can construct

$$\Omega_0 = \Omega_1 \cap \Omega_2 \cap \Omega_3 \cap \Omega_4 \cap \Omega_5 \cap \Omega_6 \, ,$$

such that $\mathbb{P}(\Omega_0) \geq 1 - \frac{6}{n}$. All further on, we restrict ourselves to the event $\Omega_0$. Restricting to this event, we obtain that, with Minkowski's inequality,

$$\sup_{B \in \mathrm{Conv}(\mathbb{R}^d)} |\mathbb{P}^{\mathsf{b}}(\sqrt{n}(\bar{\theta}_n^{\mathsf{b}} - \bar{\theta}_n) \in B) - \mathbb{P}(\sqrt{n}(\bar{\theta}_n - \theta^\star) \in B)|$$

$$\leq \sup_{B \in \mathrm{Conv}(\mathbb{R}^d)} \left|\mathbb{P}^{\mathsf{b}}(\sqrt{n}(\bar{\theta}_n^{\mathsf{b}} - \bar{\theta}_n) \in B) - \mathbb{P}^{\mathsf{b}}(\xi^{\mathsf{b}} \in B)\right|$$

$$+ \sup_{B \in \mathrm{Conv}(\mathbb{R}^d)} \left|\mathbb{P}(\xi \in B) - \mathbb{P}^{\mathsf{b}}(\xi^{\mathsf{b}} \in B)\right|$$

$$+ \sup_{B \in \mathrm{Conv}(\mathbb{R}^d)} \left|\mathbb{P}(\sqrt{n}(\bar{\theta}_n - \theta^\star) \in B) - \mathbb{P}(\xi \in B)\right|,$$

where we set $\xi^{\mathsf{b}} \sim \mathcal{N}(0, \bar{\mathbf{A}}^{-1}\Sigma_\varepsilon^{\mathsf{b}}\bar{\mathbf{A}}^{-\top})$, $\Sigma_\varepsilon^{\mathsf{b}} = n^{-1}\sum_{\ell=1}^n \varepsilon_\ell\varepsilon_\ell^\top$, and $\xi \sim \mathcal{N}(0, \Sigma_\infty)$, where $\Sigma_\infty = \bar{\mathbf{A}}^{-1}\Sigma_\varepsilon\bar{\mathbf{A}}^{-\top}$. Now we control the first supremum using Theorem 4, second one using Theorem 5, and third with Theorem 2.

**Lemma 5.** *Assume A1, A2, A3 with $\gamma = 1/2$, and A4. Then*

$$\mathbb{P}(\Omega_3) \geq 1 - 1/n.$$

*Proof.* The proof follows directly from the matrix Bernstein inequality, e.g. [69]. We note that

$$\|\Sigma_\varepsilon^{-1/2}\varepsilon_\ell\varepsilon_\ell^\top\Sigma_\varepsilon^{-1/2} - \mathrm{I}\| \leq 1 + \|\Sigma_\varepsilon^{-1/2}\varepsilon\|_\infty^2 .$$

and

$$\|\sum_{k=n+1}^{2n} \mathbb{E}[(\Sigma_\varepsilon^{-1/2}\varepsilon_\ell\varepsilon_\ell^\top\Sigma_\varepsilon^{-1/2} - \mathrm{I})^2]\| \leq n\mathbb{E}\|\Sigma_\varepsilon^{-1/2}\varepsilon\|_\infty^2 .$$

$\square$

**Lemma 6.** *Assume A1, A2, A3 with $\gamma = 1/2$, and A4. Then*

$$\mathbb{P}(\Omega_4 \cap \Omega_2) \geq 1 - \frac{1}{n}.$$

*Proof.* Denote

$$X_k = (\mathbf{A}_k - \bar{\mathbf{A}})\Gamma_{\ell+1:k-1}.$$

and let $\mathcal{F}_{k,l+1} = \sigma\{Z_j, \ell+1 \leq j \leq k\}$, $\ell+1 \leq k \leq 2n$. Then $\mathbb{E}[X_k|\mathcal{F}_{k-1,\ell+1}] = 0$. Let $S_\ell = \sum_{k=\ell+1}^n X_k$. Note that on $\Omega_2$, quadratic variation of $S_\ell$ can be controlled as

$$\mathrm{Var}^2 := \max(\|\sum_{k=\ell+1}^{2n} \mathbb{E}[X_kX_k^\top|\mathcal{F}_{k-1,l+1}]\|, \|\sum_{k=\ell+1}^{2n} \mathbb{E}[X_k^\top X_k|\mathcal{F}_{k-1,l+1}]\|)$$

$$\leq \kappa_Q \mathrm{e}^4 \mathrm{C}_{\mathbf{A}}^2 \sum_{k=\ell+1}^{2n} \prod_{j=\ell+1}^{k-1} (1 - a\alpha_j/4)^2 \leq \frac{4\kappa_Q \mathrm{e}^4 \mathrm{C}_{\mathbf{A}}^2}{a\alpha_\ell}$$

Furthermore, on $\Omega_2$

$$\|X_k\| \leq \sqrt{\kappa_Q}\mathrm{e}^2 \mathrm{C}_{\mathbf{A}} \prod_{j=\ell+1}^{k-1} (1 - a\alpha_j/4) \leq \sqrt{\kappa_Q}\mathrm{e}^2 \mathrm{C}_{\mathbf{A}} .$$

It remains to apply the Freedman inequality for matrix-values martingales [68] and use the union bound over $\ell \in [n, 2n-1]$.  $\square$

**Lemma 7.** *Assume A1, A2, A3 with $\gamma = 1/2$, and A4. Then*

$$\mathbb{P}(\Omega_5) \geq 1 - 1/n.$$

*Proof.* We first fix $h \in [1; n]$, $m \in [n, 2n - h]$, and consider the random variable

$$T_n = \| \sum_{\ell=m+1}^{m+h} \alpha_\ell (\mathbf{A}_\ell - \bar{\mathbf{A}}) \| .$$

Then we control its variance as

$$\max( \| \sum_{\ell=m+1}^{m+h} \alpha_\ell^2 \mathbb{E}[(\mathbf{A}_\ell - \bar{\mathbf{A}})(\mathbf{A}_\ell - \bar{\mathbf{A}})^\top] \|, \| \sum_{\ell=m+1}^{m+h} \alpha_\ell^2 \mathbb{E}[(\mathbf{A}_\ell - \bar{\mathbf{A}})^\top (\mathbf{A}_\ell - \bar{\mathbf{A}})] \|) \le C_{\mathbf{A}}^2 \sum_{\ell=m+1}^{m+h} \alpha_\ell^2 ,$$

moreover, $\|(\mathbf{A}_\ell - \bar{\mathbf{A}})(\mathbf{A}_\ell - \bar{\mathbf{A}})^\top\| \le C_{\mathbf{A}}^2$. Applying now the matrix Bernstein inequality [69], we obtain that with probability at least $1 - 1/n^3$, we have

$$T_n \le C_{\mathbf{A}} \sqrt{2 \sum_{\ell=m+1}^{m+h} \alpha_\ell^2} \sqrt{\log(2n^3 d)} + \frac{\alpha_{m+1} C_{\mathbf{A}}}{3} \log(2n^3 d) \le 2 C_{\mathbf{A}} \sqrt{\sum_{\ell=m+1}^{m+h} \alpha_\ell^2 \log(2n^4)} .$$

In the last line here we used that $d \le n$. Rest of the proof follows by taking union bound over $h$ and $m$ together with $\|B\|_Q^2 \le \kappa_Q \|B\|^2$ valid for any matrix $B \in \mathbb{R}^{d \times d}$. $\qquad\square$

**Lemma 8.** *Assume A1, A2, A3 with $\gamma = 1/2$, and A4. Then*

$$\mathbb{P}(\Omega_6) \ge 1 - 1/n.$$

*Proof.* It is easy to check that $\|\varepsilon_\ell \varepsilon_\ell^\top - \Sigma_\varepsilon\| \le \|\Sigma_\varepsilon\| + \|\varepsilon\|_\infty^2$. Moreover,

$$\mathrm{Var}^2 := \| \sum_{\ell=1}^n (\varepsilon_\ell \varepsilon_\ell^\top - \Sigma_\varepsilon)^2 \| \le n \|\varepsilon\|_\infty^2 \|\Sigma_\varepsilon\|.$$

It remains to apply the matrix Bernstein inequality together with the bound $n \ge d$ from A3. $\qquad\square$

## C.2 Rate of Gaussian approximation in the bootstrap world

The main result of this section is the following theorem.

**Theorem 4.** *Assume A1, A2, A3 with $\gamma = 1/2$, and A4. Then, conditionally on the event $\Omega_0$, the following error bound holds:*

$$\sup_{B \in \mathrm{Conv}(\mathbb{R}^d)} \left| \mathbb{P}^b(\sqrt{n}(\bar{\theta}_n^b - \bar{\theta}_n) \in B) - \mathbb{P}^b(\xi^b \in B) \right| \lesssim \frac{d^{1/2} \|\varepsilon\|_\infty^3}{\lambda_{\min}^{3/2} \sqrt{n}} + \frac{\kappa_Q^{3/2}(C_{\mathbf{A}}^3 \vee 1) \|\varepsilon\|_\infty^2 \log n}{a^{3/2} \lambda_{\min} n^{1/4}}$$

$$+ \frac{\kappa_Q^{3/2}(C_{\mathbf{A}}^3 \vee 1) \|\varepsilon\|_\infty n^{1/4} \sqrt{\log n}}{a^{3/2} \lambda_{\min}} \exp\{-\frac{a}{4} \sum_{j=1}^n \alpha_j\} \|\theta_0 - \theta^\star\| ,$$

*where $\xi^b \sim \mathcal{N}(0, \bar{\mathbf{A}}^{-1} \Sigma_\varepsilon^b \bar{\mathbf{A}}^{-\top})$ and $\Sigma_\varepsilon^b = n^{-1} \sum_{\ell=1}^n \varepsilon_\ell \varepsilon_\ell^\top$.*

*Proof.* Since both terms in the right-hand side of the error bound of Proposition 4 scales linearly with $\sqrt{\kappa_Q}$, for simplicity we do not trace it in the subsequent bounds (i.e. assume $\kappa_Q = 1$), and then keep the required scaling with $\kappa_Q$ only in the final bounds. Recall first that the quantities $\theta_k^b$ and $\theta_k$ are defined in (16). We start from the following decomposition:

$$\theta_k^b - \theta_k = (\mathrm{I} - \alpha_k \bar{\mathbf{A}})(\theta_{k-1}^b - \theta_{k-1}) - \alpha_k(w_k - 1)\varepsilon_k - \alpha_k(\mathbf{A}_k - \bar{\mathbf{A}})(\theta_{k-1}^b - \theta_{k-1}) - \alpha_k(w_k - 1)\mathbf{A}_k(\theta_{k-1}^b - \theta^\star) .$$

Taking average for $k$ from $n+1$ to $2n$, we get after multiplying by $\sqrt{n}$ that

$$\sqrt{n}\bar{\mathbf{A}}(\bar{\theta}_n^{\mathsf{b}} - \bar{\theta}_n) = -\underbrace{\frac{1}{\sqrt{n}}\sum_{k=n+1}^{2n}(w_k-1)\varepsilon_k}_{W^{\mathsf{b}}} + \underbrace{\frac{1}{\sqrt{n}}\frac{\theta_n^{\mathsf{b}} - \theta_n}{\alpha_n}}_{D_1^{\mathsf{b}}} - \underbrace{\frac{1}{\sqrt{n}}\frac{\theta_{2n}^{\mathsf{b}} - \theta_{2n}}{\alpha_{2n}}}_{D_2^{\mathsf{b}}}$$

$$-\underbrace{\frac{1}{\sqrt{n}}\sum_{k=n+1}^{2n}(w_k-1)\mathbf{A}_k(\theta_{k-1}^{\mathsf{b}} - \theta^\star)}_{D_3^{\mathsf{b}}} + \underbrace{\frac{1}{\sqrt{n}}\sum_{k=n+1}^{2n}(\theta_{k-1}^{\mathsf{b}} - \theta_{k-1})\left(\frac{1}{\alpha_k} - \frac{1}{\alpha_{k-1}}\right)}_{D_4^{\mathsf{b}}}$$

$$-\underbrace{\frac{1}{\sqrt{n}}\sum_{k=n+1}^{2n}(\mathbf{A}_k - \bar{\mathbf{A}})(\theta_{k-1}^{\mathsf{b}} - \theta_{k-1})}_{D_5^{\mathsf{b}}}. \quad (48)$$

The formula (48) resembles the key representation $T^{\mathsf{b}} := \sqrt{n}\bar{\mathbf{A}}(\bar{\theta}_n^{\mathsf{b}} - \bar{\theta}_n) = W^{\mathsf{b}} + D^{\mathsf{b}}$, where

$$D^{\mathsf{b}} = D_1^{\mathsf{b}} + \ldots + D_5^{\mathsf{b}}, \quad (49)$$

and $D_1^{\mathsf{b}} - D_5^{\mathsf{b}}$ are defined in (48). Now we aim to apply the result of [63]:

$$\sup_{B\in\mathrm{Conv}(\mathbb{R}^d)}|\mathbb{P}^{\mathsf{b}}(T^{\mathsf{b}}\in B) - \mathbb{P}^{\mathsf{b}}(\xi^{\mathsf{b}}\in B)| \leq 259d^{1/2}\Upsilon + 2\mathbb{E}^{\mathsf{b}}[\|W^{\mathsf{b}}\|\|D^{\mathsf{b}}\|]$$

$$+ 2\sum_{\ell=n+1}^{2n}\mathbb{E}^{\mathsf{b}}[\|\xi_\ell\|\|D^{\mathsf{b}} - D^{(\mathsf{b},\ell)}\|], \quad (50)$$

where $\xi_\ell = \frac{1}{\sqrt{n}}(w_\ell-1)\varepsilon_\ell$. We finish the proof by the application of the formula (50). In order to bound the quantities $\mathbb{E}^{\mathsf{b}}\|D^{\mathsf{b}}\|^2$ and $\mathbb{E}^{\mathsf{b}}\|D^{(\mathsf{b},i)}\|^2$, we apply the respective results of Proposition 5 and Proposition 6, respectively. Namely, applying the Cauchy-Schwartz inequality together with Proposition 5, we get that on the event $\Omega_0$ it holds

$$\mathbb{E}^{\mathsf{b}}[\|D^{\mathsf{b}}\|\|W^{\mathsf{b}}\|] \leq \{\mathbb{E}^{\mathsf{b}}[\|D^{\mathsf{b}}\|^2]\}^{1/2}\{\mathbb{E}^{\mathsf{b}}[\|W^{\mathsf{b}}\|^2]\}^{1/2} \lesssim \frac{\kappa_Q^2(\mathrm{C}_{\mathbf{A}}^4\vee1)\|\varepsilon\|_\infty\sqrt{\mathrm{Tr}\,\Sigma_\varepsilon^{\mathsf{b}}}\log n}{n^{1/4}a^{5/2}}$$

$$+ \kappa_Q^{3/2}\|\varepsilon\|_\infty\left(\frac{(\mathrm{C}_{\mathbf{A}}^3\vee1)n^{1/4}}{\sqrt{a}} + \frac{(\mathrm{C}_{\mathbf{A}}^5\vee1)\sqrt{\log n}}{\sqrt{n}a}\right)\exp\left\{-\frac{c_0a\sqrt{n}}{2}\right\}\|\theta_0 - \theta^\star\|.$$

Similarly, applying Minkowski's inequality and Proposition 6, we obtain that

$$\mathbb{E}^{\mathsf{b}}\Big[\sum_{i=n}^{2n-1}\|\xi_i\|\|D^{\mathsf{b}} - D^{(\mathsf{b},i)}\|\Big] \leq \{\mathbb{E}^{\mathsf{b}}[\|\xi_1\|^2]\}^{1/2}\sum_{i=n}^{2n-1}\{\mathbb{E}^{\mathsf{b}}[\|D^{\mathsf{b}} - D^{(\mathsf{b},i)}\|^2]\}^{1/2}$$

$$\lesssim \frac{\kappa_Q^{3/2}(\mathrm{C}_{\mathbf{A}}^3\vee1)\|\varepsilon\|_\infty^2\log n}{a^{3/2}n^{1/4}} + \frac{\kappa_Q\|\varepsilon\|_\infty^2\mathrm{C}_{\mathbf{A}}^2}{a^2\sqrt{n}} + \frac{\kappa_Q^{3/2}(\mathrm{C}_{\mathbf{A}}^3\vee1)\|\varepsilon\|_\infty n^{1/4}\sqrt{\log n}}{a^{3/2}}\exp\Big\{-\frac{a}{4}\sum_{j=1}^n\alpha_j\Big\}\|\theta_0 - \theta^\star\|.$$

Now it remains to combine the bounds above in (50). $\qquad\square$

**Proposition 5.** *Assume A1, A2, A3 with $\gamma = 1/2$, and A4. Then, conditionally on the event $\Omega_0$, the following error bound holds:*

$$\{\mathbb{E}^{\mathsf{b}}\left[\|D^{\mathsf{b}}(w_1,\ldots,w_{2n},Z_1,\ldots,Z_{2n})\|^2\right]\}^{1/2} \lesssim \frac{\kappa_Q^2(\mathrm{C}_{\mathbf{A}}^4\vee1)\|\varepsilon\|_\infty\log n}{n^{1/4}a^{5/2}}$$

$$+ \kappa_Q^{3/2}\left(\frac{(\mathrm{C}_{\mathbf{A}}^3\vee1)n^{1/4}}{\sqrt{a}} + \frac{(\mathrm{C}_{\mathbf{A}}^5\vee1)\sqrt{\log n}}{\sqrt{n}a}\right)\exp\left\{-\frac{c_0a\sqrt{n}}{2}\right\}\|\theta_0 - \theta^\star\|,$$

*where $\lesssim$ stands for inequality up to an absolute constant.*

Proof of Proposition 5 is provided below in Appendix C.5. The lemma below is a direct counterpart of Lemma 4.

**Proposition 6.** *Assume A1, A2, A3 with $\gamma = 1/2$, and A4. Then, conditionally on the event $\Omega_0$, the following error bound holds:*

$$\sum_{i=n+1}^{2n} \mathbb{E}^{\mathsf{b}}[\|D^{\mathsf{b}} - D^{(\mathsf{b},i)}\|^2]^{1/2} \lesssim \frac{(\mathrm{C}_{\mathbf{A}}^3 \vee 1)\|\varepsilon\|_\infty}{a^{3/2}} n^{1/4} \log n + \frac{\|\varepsilon\|_\infty \mathrm{C}_{\mathbf{A}}^2}{a^2}$$
$$+ \frac{(\mathrm{C}_{\mathbf{A}}^3 \vee 1)n^{3/4}\sqrt{\log n}}{a^{3/2}} \exp\Big\{-\frac{a}{4}\sum_{j=1}^n \alpha_j\Big\}\|\theta_0 - \theta^\star\| \, ,$$

Proof of Proposition 6 is provided below in Appendix C.6.

**Lemma 9.** *For any $k \geq n$ on the set $\Omega_0$ the following inequality holds:*

$$\mathbb{E}^{\mathsf{b}}[\|\theta_k^{\mathsf{b}} - \theta_k\|^2] \lesssim \frac{\alpha_k\|\varepsilon\|_\infty^2 \mathrm{C}_{\mathbf{A}}^2}{a^3} + \mathrm{C}_{\mathbf{A}}^2 \, k \prod_{j=1}^k (1 - a\alpha_j/4)^2 \|\theta_0 - \theta^\star\|^2 \, .$$

*Proof.* A direct application of Lemma 11 with $L = 0$ yields that

$$\mathbb{E}^{\mathsf{b}}[\|\theta_k^{\mathsf{b}} - \theta_k\|^2] \lesssim \frac{\alpha_k\|\varepsilon\|_\infty^2}{a}\left(1 + \frac{\mathrm{C}_{\mathbf{A}}^2}{a^2}\right) + \mathrm{C}_{\mathbf{A}}^2 \, k \prod_{j=1}^k (1 - a\alpha_j/4)^2 \|\theta_0 - \theta^\star\|^2 \, .$$

Now to complete the proof it remains to notice that $\mathrm{C}_{\mathbf{A}} \geq a$. $\qquad\square$

**Lemma 10.** *For any matrix-valued sequences $(U_n)_{n\in\mathbb{N}}$, $(V_n)_{n\in\mathbb{N}}$ and for any $M \in \mathbb{N}$, it holds that:*

$$\prod_{k=1}^M U_k - \prod_{k=1}^M V_k = \sum_{k=1}^M \Big\{ \prod_{j=k+1}^M V_j \Big\}(U_k - V_k)\Big\{\prod_{j=1}^{k-1} U_j\Big\} \, .$$

## C.3 Gaussian comparison inequality

**Theorem 5.** *Assume A1 and A2. Then on the set $\Omega_3$*

$$\sup_{B\in\mathrm{Conv}(\mathbb{R}^d)} |\mathbb{P}(\xi \in B) - \mathbb{P}^{\mathsf{b}}(\xi^{\mathsf{b}} \in B)| \leq 4\|\Sigma_\varepsilon^{-1/2}\varepsilon\|_\infty \sqrt{\frac{d\log n}{n}} + \frac{4\sqrt{d}(1 + \|\Sigma_\varepsilon^{-1/2}\varepsilon\|_\infty^2)\log n}{n}$$

*Proof.* We will use the following inequality

$$\|\mathcal{N}(0,\Sigma_1) - \mathcal{N}(0,\Sigma_2)\|_{\mathsf{TV}} \leq \frac{1}{2}\|\Sigma_1^{-1/2}\Sigma_2\Sigma^{-1/2} - \mathrm{I}\|_{\mathsf{Fr}} \tag{51}$$

Applying (51) we obtain

$$\sup_{B\in\mathrm{Conv}(\mathbb{R}^d)} |\mathbb{P}(\xi \in B) - \mathbb{P}^{\mathsf{b}}(\xi^{\mathsf{b}} \in B)\| \leq \frac{\sqrt{d}}{2}\|\Sigma_\varepsilon^{-1/2}\Sigma_\varepsilon^{\mathsf{b}}\Sigma_\varepsilon^{-1/2} - \mathrm{I}\|.$$

It remains to apply definition of $\Omega_3$. $\qquad\square$

## C.4 Auxiliary technical results.

For the analysis of the difference term $\theta_k^{\mathsf{b}} - \theta_k$ we use the perturbation expansion technique introduced in [1], see also [16]. Within this approach, we represent the fluctuation component of the error $\tilde{\theta}_n^{(\mathsf{fl})}$ defined in (26) as

$$\tilde{\theta}_n^{(\mathsf{fl})} = J_n^{(0)} + H_n^{(0)} \, ,$$

where the latter terms are defined by the following pair of recursions

$$J_n^{(0)} = \left(\mathrm{I} - \alpha_n\bar{\mathbf{A}}\right) J_{n-1}^{(0)} - \alpha_n\varepsilon(Z_n) \, , \qquad\qquad J_0^{(0)} = 0 \, , \tag{52}$$

$$H_n^{(0)} = (\mathrm{I} - \alpha_n\mathbf{A}(Z_n)) H_{n-1}^{(0)} - \alpha_n\tilde{\mathbf{A}}(Z_n)J_{n-1}^{(0)} \, , \qquad\qquad H_0^{(0)} = 0 \, .$$

Moreover, it is known that for $L \geq 1$ the term $H_n^{(0)}$ can be further decomposed as follows:

$$H_n^{(0)} = \sum_{\ell=1}^{L} J_n^{(\ell)} + H_n^{(L)} \ .$$

Here the terms $J_n^{(\ell)}$ and $H_n^{(\ell)}$ are given by the following recurrences:

$$
\begin{aligned}
J_n^{(\ell)} &= \left(\mathrm{I} - \alpha_n \bar{\mathbf{A}}\right) J_{n-1}^{(\ell)} - \alpha_n \tilde{\mathbf{A}}(Z_n) J_{n-1}^{(\ell-1)} \ , & J_0^{(\ell)} &= 0 \ , \\
H_n^{(\ell)} &= \left(\mathrm{I} - \alpha_n \mathbf{A}(Z_n)\right) H_{n-1}^{(\ell)} - \alpha_n \tilde{\mathbf{A}}(Z_n) J_{n-1}^{(\ell)} \ , & H_0^{(\ell)} &= 0 \ .
\end{aligned}
\tag{53}
$$

The expansion depth $L$ here controls the desired approximation accuracy. Informally, one can show that $\mathbb{E}^{1/p}[\||J_n^{(\ell)}\||^p] \lesssim \alpha_n^{(\ell+1)/2}$, and similarly $\mathbb{E}^{1/p}[\||H_n^{(\ell)}\||^p] \lesssim \alpha_n^{(\ell+1)/2}$. Using the outlined expansion, we prove the following lemma:

**Lemma 11.** *Assume A1, A2, A3 with $\gamma = 1/2$, and A4. Then for any $k \geq n$ and $L \in \mathbb{N}$ the following decomposition holds:*

$$\theta_k^{\mathrm{b}} - \theta_k = J_k^{\mathrm{b},0} + \sum_{j=1}^{L} J_k^{\mathrm{b},j} + H_k^{\mathrm{b},L}, \tag{54}$$

*where*

$$
\begin{aligned}
J_k^{\mathrm{b},0} &= - \sum_{\ell=n+1}^{k} \alpha_\ell (w_\ell - 1) \Gamma_{\ell+1:k} \tilde{\varepsilon}_\ell, \\
J_k^{\mathrm{b},j} &= - \sum_{\ell=n+1}^{k} \alpha_\ell (w_\ell - 1) \Gamma_{\ell+1:k} A_\ell J_{\ell-1}^{\mathrm{b},j-1}, \quad j \in [1, L] \\
H_k^{\mathrm{b},L} &= - \sum_{\ell=n+1}^{k} \alpha_\ell (w_\ell - 1) \Gamma_{\ell+1:k}^{\mathrm{b}} A_\ell J_{\ell-1}^{\mathrm{b},L} \ ,
\end{aligned}
\tag{55}
$$

*and the quantities $\tilde{\varepsilon}_\ell$ are defined as*

$$\tilde{\varepsilon}_\ell = \mathbf{A}_\ell(\theta_{\ell-1} - \theta^\star) + \varepsilon_\ell \ .$$

*Moreover, on the event $\Omega_0$,*

$$\mathbb{E}^{\mathrm{b}}[\||J_k^{\mathrm{b},j}\||^2] \lesssim \frac{\alpha_k^{j+1} \|\varepsilon\|_\infty^2 \, \mathrm{C}_{\mathbf{A}}^{2j}}{a^{j+1}} + \mathrm{C}_{\mathbf{A}}^{2j+2} \prod_{j=1}^{k} (1 - a\alpha_j/4)^2 \|\theta_0 - \theta^\star\|^2 \ , \quad j \in [0, L] \tag{56}$$

$$\mathbb{E}^{\mathrm{b}}[\||H_k^{\mathrm{b},L}\||^2] \lesssim \frac{\alpha_k^{L+1} \mathrm{C}_{\mathbf{A}}^{2(L+1)} \|\varepsilon\|_\infty^2}{a^{L+3}} + \mathrm{C}_{\mathbf{A}}^{2(L+1)} k \prod_{j=1}^{k} (1 - a\alpha_j/4)^2 \|\theta_0 - \theta^\star\|^2 \ . \tag{57}$$

*Proof.* We start from the decomposition

$$\theta_k^{\mathrm{b}} - \theta_k = (\mathrm{I} - \alpha_k w_k \mathbf{A}_k)(\theta_{k-1}^{\mathrm{b}} - \theta_{k-1}) - \alpha_k (w_k - 1) \tilde{\varepsilon}_k. \tag{58}$$

Expanding the recurrence above till $k = n$, and using the fact that $\theta_n^{\mathrm{b}} = \theta_n$, we get running the recurrence (58), that

$$\theta_k^{\mathrm{b}} - \theta_k = - \sum_{\ell=n+1}^{k} \alpha_\ell (w_\ell - 1) \Gamma_{\ell+1:k}^{\mathrm{b}} \tilde{\varepsilon}_\ell \ .$$

Hence, proceeding as in (52), we obtain the representation

$$
\begin{aligned}
J_k^{(\mathrm{b},0)} &= (\mathrm{I} - \alpha_k \mathbf{A}_k) J_{k-1}^{(\mathrm{b},0)} - \alpha_k (w_k - 1) \tilde{\varepsilon}_k \ , & J_0^{(\mathrm{b},0)} &= 0 \ , \\
H_k^{(\mathrm{b},0)} &= (\mathrm{I} - \alpha_k w_k \mathbf{A}_k) H_{k-1}^{(\mathrm{b},0)} - \alpha_k (w_k - 1) \mathbf{A}_k J_{k-1}^{(\mathrm{b},0)} \ , & H_0^{(\mathrm{b},0)} &= 0 \ .
\end{aligned}
$$

It is easy to check that $J_k^{(b,0)} + H_k^{(b,0)} = \theta_k^b - \theta_k$. Similarly, with further expansion of $H_k^{(b,0)}$ along the lines of (53), we arrive at the decomposition (54). Since $w_k$ for $k = n+1, \ldots, 2n$ are i.i.d., we get using the definition of the events $\Omega_1$ and $\Omega_2$, that on the event $\Omega_0$:

$$
\begin{aligned}
\|\tilde{\varepsilon}_\ell\|^2 &\lesssim \|\varepsilon\|_\infty^2 + \mathrm{C_A^2} \exp\Big\{-a \sum_{j=1}^{\ell-1} \alpha_j\Big\} \|\theta_0 - \theta^\star\|^2 + \frac{\alpha_\ell \|\varepsilon\|_\infty^2 \log^2 n}{a} \\
&\lesssim \|\varepsilon\|_\infty^2 + \mathrm{C_A^2} \prod_{j=1}^{\ell-1} (1 - a\alpha_j/2)^2 \|\theta_0 - \theta^\star\|^2 + \frac{\alpha_\ell \|\varepsilon\|_\infty^2 \log^2 n}{a} \qquad (59) \\
&\lesssim \|\varepsilon\|_\infty^2 + \mathrm{C_A^2} \prod_{j=1}^{\ell-1} (1 - a\alpha_j/2)^2 \|\theta_0 - \theta^\star\|^2 \,,
\end{aligned}
$$

where for the last bound we have additionally used that $\alpha_\ell \log^2 n / a \leq 1$ for $\ell \geq n$. The latter bound is guaranteed by A4. Hence, using the bound (59) together with the definition of $J_k^{b,0}$, we obtain that

$$
\begin{aligned}
\mathbb{E}^b[\|J_k^{b,0}\|^2] &= \sum_{\ell=n+1}^k \alpha_\ell^2 \|\Gamma_{\ell+1:k} \tilde{\varepsilon}_\ell\|^2 = \sum_{\ell=n+1}^k \alpha_\ell^2 \|\Gamma_{\ell+1:k} \big( \mathbf{A}_\ell(\theta_{\ell-1} - \theta^\star) + \varepsilon_\ell \big)\|^2 \\
&\lesssim \|\varepsilon\|_\infty^2 \sum_{\ell=n+1}^k \alpha_\ell^2 \prod_{j=\ell+1}^k (1 - a\alpha_j/4)^2 + \mathrm{C_A^2} \sum_{\ell=n+1}^k \alpha_\ell^2 \prod_{j=1}^k (1 - a\alpha_j/4)^2 \|\theta_0 - \theta^\star\|^2 \\
&\quad + \frac{\|\varepsilon\|_\infty^2 \mathrm{C_A^2} \log^2 n}{a} \sum_{\ell=n+1}^k \alpha_\ell^3 \prod_{j=\ell+1}^k (1 - a\alpha_j/4)^2 \\
&\lesssim \frac{\|\varepsilon\|_\infty^2 \alpha_k}{a} + \mathrm{C_A^2} \log\left(\frac{k}{n}\right) \prod_{j=1}^k (1 - a\alpha_j/4)^2 \|\theta_0 - \theta^\star\|^2 + \frac{\alpha_k^2 \|\varepsilon\|_\infty^2 \mathrm{C_A^2} \log^2 n}{a^2} \\
&\lesssim \frac{\|\varepsilon\|_\infty^2 \alpha_k}{a} + \mathrm{C_A^2} \prod_{j=1}^k (1 - a\alpha_j/4)^2 \|\theta_0 - \theta^\star\|^2 \,,
\end{aligned}
$$

where we additionally used the fact that $k \in [n; 2n]$ and $n$ satisfies A4. Assume now that the bound on $J_k^{b,j-1}$ has a form

$$
\mathbb{E}^b[\|J_k^{b,j-1}\|^2] \lesssim \frac{\|\varepsilon\|_\infty^2 \alpha_k^j}{a^j} + \mathrm{C_A^{2j}} \prod_{\ell=1}^k (1 - a\alpha_\ell/4)^2 \|\theta_0 - \theta^\star\|^2 \,.
$$

Then, using the martingale property of $J_k^{b,j}$, we write that

$$
\begin{aligned}
\mathbb{E}^b[\|J_k^{b,j}\|^2] &= \sum_{\ell=n+1}^k \alpha_\ell^2 \mathbb{E}^b[\|\Gamma_{\ell+1:k} A_\ell J_{\ell-1}^{b,j-1}\|^2] \\
&\lesssim \sum_{\ell=n+1}^k \frac{\alpha_\ell^{j+2} \|\varepsilon\|_\infty^2 \mathrm{C_A^{2j}}}{a^j} \prod_{j=\ell+1}^k (1 - a\alpha_j/4)^2 + \mathrm{C_A^{2j+2}} \sum_{\ell=n+1}^k \alpha_\ell^2 \prod_{j=1}^k (1 - a\alpha_j/4)^2 \|\theta_0 - \theta^\star\|^2 \\
&\lesssim \frac{\alpha_k^{j+1} \|\varepsilon\|_\infty^2 \mathrm{C_A^{2j}}}{a^{j+1}} + \mathrm{C_A^{2j+2}} \prod_{j=1}^k (1 - a\alpha_j/4)^2 \|\theta_0 - \theta^\star\|^2 \,,
\end{aligned}
$$

and thus the bound (56) is proved. Moreover, using (55) and Minkowski's inequality, we obtain that

$$(\mathbb{E}^{\mathsf{b}}[\|H_k^{\mathsf{b},L}\|^2])^{1/2} \leq C_{\mathbf{A}} \sum_{\ell=n+1}^k \alpha_\ell (\mathbb{E}^{\mathsf{b}}[\|\Gamma_{\ell+1:k}^{\mathsf{b}}\|^2])^{1/2} (\mathbb{E}^{\mathsf{b}}[\|J_{\ell-1}^{\mathsf{b},L}\|^2])^{1/2}$$

$$\lesssim C_{\mathbf{A}} \sum_{\ell=n+1}^k \frac{\alpha_\ell^{(L+3)/2} \|\varepsilon\|_\infty C_{\mathbf{A}}^L}{a^{(L+1)/2}} \prod_{j=\ell+1}^k (1 - a\alpha_j/4)$$

$$+ C_{\mathbf{A}}^{L+1} \sum_{\ell=n+1}^k \alpha_\ell \prod_{j=1}^k (1 - a\alpha_j/4)^2 \|\theta_0 - \theta^\star\|$$

$$\lesssim \frac{\alpha_k^{(L+1)/2} C_{\mathbf{A}}^{L+1} \|\varepsilon\|_\infty}{a^{(L+3)/2}} + C_{\mathbf{A}}^{L+1} \sqrt{k} \prod_{j=1}^k (1 - a\alpha_j/4)^2 \|\theta_0 - \theta^\star\| \,,$$

and (57) follows. $\qquad\square$

## C.5  Proof of Proposition 5

Recall that the quantity $D^{\mathsf{b}}$ is defined in (49). Since $\theta_n^{\mathsf{b}} = \theta_n$, we conclude that $D_1^{\mathsf{b}} = 0$. To estimate other terms we will use the main error decomposition outlined in Lemma 11, that is, the expansion

$$\theta_k^{\mathsf{b}} - \theta_k = \sum_{\ell=0}^L J_k^{\mathsf{b},\ell} + H_k^{\mathsf{b},L},$$

applied with different $L \geq 0$. To bound $D_2^{\mathsf{b}}$ we take $L = 0$ and obtain

$$\mathbb{E}^{\mathsf{b}}[\|D_2^{\mathsf{b}}\|^2] \lesssim \mathbb{E}^{\mathsf{b}}[\|J_{2n}^{\mathsf{b},j}\|^2] + \mathbb{E}^{\mathsf{b}}[\|H_{2n}^{\mathsf{b},L}\|^2] \lesssim \frac{\|\varepsilon\|_\infty^2}{n\alpha_{2n}a}\left(1 + \frac{C_{\mathbf{A}}^2}{a^2}\right) + \frac{C_{\mathbf{A}}^2}{\alpha_{2n}^2} \prod_{j=1}^{2n} (1 - a\alpha_j/4)^2 \|\theta_0 - \theta^\star\|^2$$

$$\lesssim \frac{\|\varepsilon\|_\infty^2}{a\sqrt{n}}\left(1 + \frac{C_{\mathbf{A}}^2}{a^2}\right) + n\, C_{\mathbf{A}}^2 \prod_{j=1}^{2n} (1 - a\alpha_j/4)^2 \|\theta_0 - \theta^\star\|^2 \,.$$

To estimate $D_3^{\mathsf{b}}$ we note that

$$D_3^{\mathsf{b}} = \frac{1}{\sqrt{n}} \sum_{k=n+1}^{2n} (w_k - 1)\mathbf{A}_k(\theta_{k-1}^{\mathsf{b}} - \theta^\star) = D_{3,1}^{\mathsf{b}} + D_{3,2}^{\mathsf{b}} \,,$$

where we have set, respectively,

$$D_{3,1}^{\mathsf{b}} = \frac{1}{\sqrt{n}} \sum_{k=n+1}^{2n} (w_k - 1)\mathbf{A}_k(\theta_{k-1}^{\mathsf{b}} - \theta_{k-1}),$$

$$D_{3,2}^{\mathsf{b}} = \frac{1}{\sqrt{n}} \sum_{k=n+1}^{2n} (w_k - 1)\mathbf{A}_k(\theta_{k-1} - \theta^\star) \,.$$

It follows from Lemma 9 that on the event $\Omega_0$ it holds

$$\mathbb{E}^{\mathsf{b}}[\|D_{3,1}^{\mathsf{b}}\|^2] \leq \frac{C_{\mathbf{A}}^2}{n} \sum_{k=n+1}^{2n} \mathbb{E}^{\mathsf{b}}[\|\theta_{k-1}^{\mathsf{b}} - \theta_{k-1}\|^2]$$

$$\lesssim \frac{C_{\mathbf{A}}^2}{n} \sum_{k=n+1}^{2n} \frac{\alpha_k \|\varepsilon\|_\infty^2}{a}\left(1 + \frac{C_{\mathbf{A}}^2}{a^2}\right) + C_{\mathbf{A}}^4 \sum_{k=n+1}^{2n} \prod_{j=1}^k (1 - a\alpha_j/4)^2 \|\theta_0 - \theta^\star\|^2$$

$$\lesssim \frac{C_{\mathbf{A}}^2 \|\varepsilon\|_\infty^2}{a\sqrt{n}}\left(1 + \frac{C_{\mathbf{A}}^2}{a^2}\right) + \frac{C_{\mathbf{A}}^4 \sqrt{n}}{a} \exp\left\{-c_0 a\sqrt{n}\right\} \|\theta_0 - \theta^\star\|^2$$

Moreover, on the set $\Omega_0$ it holds (since $\Omega_0 \subseteq \Omega_1$), that

$$
\mathbb{E}^{\mathsf{b}}[\|D_{3,2}^{\mathsf{b}}\|^2] = \frac{1}{n} \sum_{k=n+1}^{2n} \|\mathbf{A}_k(\theta_{k-1} - \theta^\star)\|^2
$$

$$
\lesssim \frac{\mathrm{C}_{\mathbf{A}}^2}{n} \sum_{k=n+1}^{2n} \left( \exp\left\{-a \sum_{\ell=1}^{k} \alpha_\ell\right\} \|\theta_0 - \theta^\star\|^2 + \frac{\alpha_k \|\varepsilon\|_\infty^2 \log^2 n}{a} \right)
$$

$$
\lesssim \frac{\mathrm{C}_{\mathbf{A}}^2}{n a \alpha_{2n}} \exp\left\{-a \sum_{\ell=1}^{n} \alpha_\ell\right\} \|\theta_0 - \theta^\star\|^2 + \frac{\mathrm{C}_{\mathbf{A}}^2 \|\varepsilon\|_\infty^2 \log^2 n}{na} \sum_{k=n+1}^{2n} \alpha_k
$$

$$
\lesssim \frac{\mathrm{C}_{\mathbf{A}}^2}{a\sqrt{n}} \exp\left\{-c_0 a \sqrt{n}\right\} \|\theta_0 - \theta^\star\|^2 + \frac{\mathrm{C}_{\mathbf{A}}^2 \|\varepsilon\|_\infty^2 \log^2 n}{a\sqrt{n}} .
$$

Combining the above bounds, we get

$$
\mathbb{E}^{\mathsf{b}}[\|D_3^{\mathsf{b}}\|^2] \lesssim \mathbb{E}^{\mathsf{b}}[\|D_{3,1}^{\mathsf{b}}\|^2] + \mathbb{E}^{\mathsf{b}}[\|D_{3,2}^{\mathsf{b}}\|^2]
$$

$$
\lesssim \frac{\mathrm{C}_{\mathbf{A}}^2 \|\varepsilon\|_\infty^2 \log^2 n}{a\sqrt{n}} \left(1 + \frac{\mathrm{C}_{\mathbf{A}}^2}{a^2}\right) + \frac{\mathrm{C}_{\mathbf{A}}^4 \sqrt{n}}{a} \exp\left\{-c_0 a \sqrt{n}\right\} \|\theta_0 - \theta^\star\|^2 .
$$

Now we proceed with the term $D_4^{\mathsf{b}}$. Applying Minkowski's inequality, we get

$$
\{\mathbb{E}^{\mathsf{b}}[\|D_4^{\mathsf{b}}\|^2]\}^{1/2} \leq \frac{1}{\sqrt{n}} \sum_{k=n+1}^{2n} \left(\frac{1}{\alpha_k} - \frac{1}{\alpha_{k-1}}\right) \{\mathbb{E}^{\mathsf{b}}[\|\theta_{k-1}^{\mathsf{b}} - \theta_{k-1}\|^2]\}^{1/2}
$$

$$
\lesssim \frac{1}{\sqrt{n}} \sum_{k=n+1}^{2n} \frac{1}{\sqrt{k}} \left( \frac{\sqrt{\alpha_k}\|\varepsilon\|_\infty}{\sqrt{a}} \left(1 + \frac{\mathrm{C}_{\mathbf{A}}}{a}\right) + \mathrm{C}_{\mathbf{A}}\sqrt{k} \prod_{j=1}^{k}(1 - a\alpha_j/4)\|\theta_0 - \theta^\star\| \right)
$$

$$
\lesssim \frac{\|\varepsilon\|_\infty}{n^{1/4}\sqrt{a}} \left(1 + \frac{\mathrm{C}_{\mathbf{A}}}{a}\right) + \frac{\mathrm{C}_{\mathbf{A}}}{\sqrt{n}} \sum_{k=n+1}^{2n} \prod_{j=1}^{k}(1 - a\alpha_j/4)\|\theta_0 - \theta^\star\|
$$

$$
\lesssim \frac{\|\varepsilon\|_\infty}{n^{1/4}\sqrt{a}} \left(1 + \frac{\mathrm{C}_{\mathbf{A}}}{a}\right) + \mathrm{C}_{\mathbf{A}} \exp\left\{-\frac{c_0 a \sqrt{n}}{2}\right\} \|\theta_0 - \theta^\star\| .
$$

It remains to upper bound the term $D_5^{\mathsf{b}}$. Using the decomposition, suggested by Lemma 11 with $L = 2$, we get that

$$
D_5^{\mathsf{b}} = \frac{1}{\sqrt{n}} \sum_{k=n+1}^{2n} (\mathbf{A}_k - \bar{\mathbf{A}})(\theta_{k-1}^{\mathsf{b}} - \theta_{k-1}) = \underbrace{\frac{1}{\sqrt{n}} \sum_{k=n+1}^{2n} (\mathbf{A}_k - \bar{\mathbf{A}})J_{k-1}^{\mathsf{b},0}}_{D_{5,1}^{\mathsf{b}}}
$$

$$
+ \underbrace{\frac{1}{\sqrt{n}} \sum_{k=n+1}^{2n} (\mathbf{A}_k - \bar{\mathbf{A}})J_{k-1}^{\mathsf{b},1}}_{D_{5,2}^{\mathsf{b}}} + \underbrace{\frac{1}{\sqrt{n}} \sum_{k=n+1}^{2n} (\mathbf{A}_k - \bar{\mathbf{A}})J_{k-1}^{\mathsf{b},2}}_{D_{5,3}^{\mathsf{b}}} + \underbrace{\frac{1}{\sqrt{n}} \sum_{k=n+1}^{2n} (\mathbf{A}_k - \bar{\mathbf{A}})H_{k-1}^{\mathsf{b},2}}_{D_{5,4}^{\mathsf{b}}} .
$$

Here we have to consider expansion until $H^{\mathsf{b},2}$, since dealing with the latter term (outlined as $D_{5,4}^{\mathsf{b}}$ in the above expansion) is possible only with Minkowski's inequality. Now we consider the summands $D_{5,1}^{\mathsf{b}} - D_{5,4}^{\mathsf{b}}$ separately. Consider first the term $D_{5,1}^{\mathsf{b}}$. Changing the summation order, we obtain

$$
D_{5,1}^{\mathsf{b}} = -\frac{1}{\sqrt{n}} \sum_{k=n+1}^{2n} (\mathbf{A}_k - \bar{\mathbf{A}}) \sum_{\ell=n+1}^{k-1} \alpha_\ell (w_\ell - 1) \Gamma_{\ell+1:k-1} \tilde{\varepsilon}_\ell
$$

$$
= -\frac{1}{\sqrt{n}} \sum_{\ell=n+1}^{2n-1} \alpha_\ell (w_\ell - 1) \left( \sum_{k=\ell+1}^{2n} (\mathbf{A}_k - \bar{\mathbf{A}}) \Gamma_{\ell+1:k-1} \right) \tilde{\varepsilon}_\ell .
$$

Then on the event $\Omega_0$ we get, since $\Omega_0 \subseteq \Omega_4$, and using that $n$ satisfies A4,

$$\| \sum_{k=\ell+1}^{2n} (\mathbf{A}_k - \bar{\mathbf{A}})\Gamma_{\ell+1:k-1}\|^2 \lesssim \frac{\log n}{a\alpha_\ell} \ . \tag{60}$$

Combining the above bound together with the one provided by (59), we obtain that

$$\mathbb{E}^{\mathsf{b}}[\|D_{5,1}^{\mathsf{b}}\|^2] \lesssim \frac{\|\varepsilon\|_\infty^2}{n} \sum_{\ell=n+1}^{2n-1} \frac{\alpha_\ell \log n}{a} + \frac{\mathrm{C}_{\mathbf{A}}^2}{n} \sum_{\ell=n+1}^{2n-1} \frac{\alpha_\ell \log n}{a} \prod_{j=1}^{\ell-1}(1-a\alpha_j/2)^2\|\theta_0 - \theta^\star\|^2$$

$$\lesssim \frac{\|\varepsilon\|_\infty^2 \log n}{\sqrt{n}a} + \frac{\mathrm{C}_{\mathbf{A}}^2 \log n}{a^2 n} \exp\{-a\sum_{j=1}^n \alpha_j\}\|\theta_0 - \theta^\star\|^2 \ .$$

Similarly, for the term $D_{5,2}^{\mathsf{b}}$ we get, changing the order of summation, that

$$D_{5,2}^{\mathsf{b}} = \frac{1}{\sqrt{n}} \sum_{\ell=n+1}^{2n-1} \alpha_\ell(w_\ell - 1)\Big( \sum_{k=\ell+1}^{2n} (\mathbf{A}_k - \bar{\mathbf{A}})\Gamma_{\ell+1:k-1}\Big) A_\ell J_{\ell-1}^{\mathsf{b},0} \ .$$

Hence, using the bound (60) together with (56), we get

$$\mathbb{E}^{\mathsf{b}}[\|D_{5,2}^{\mathsf{b}}\|^2] \lesssim \frac{1}{n} \sum_{\ell=n+1}^{2n-1} \frac{\alpha_\ell \log n}{a} \mathrm{C}_{\mathbf{A}}^2 \Big( \frac{\alpha_\ell\|\varepsilon\|_\infty^2}{a} + \mathrm{C}_{\mathbf{A}}^2 \prod_{j=1}^{\ell-1}(1-a\alpha_j/4)^2\|\theta_0 - \theta^\star\|^2 \Big)$$

$$\lesssim \frac{\mathrm{C}_{\mathbf{A}}^2 \|\varepsilon\|_\infty^2 \log n}{na^2} \sum_{\ell=n+1}^{2n-1} \alpha_\ell^2 + \frac{\mathrm{C}_{\mathbf{A}}^4 \log n}{na} \sum_{\ell=n+1}^{2n-1} \alpha_\ell \prod_{j=1}^{\ell-1}(1-a\alpha_j/4)^2\|\theta_0 - \theta^\star\|^2$$

$$\lesssim \frac{\mathrm{C}_{\mathbf{A}}^2 \|\varepsilon\|_\infty^2 \log n}{na^2} + \frac{\mathrm{C}_{\mathbf{A}}^4 \log n}{na} \exp\{-(a/2)\sum_{j=1}^n \alpha_j\}\|\theta_0 - \theta^\star\|^2 \ .$$

We proceed with $D_{5,3}^{\mathsf{b}}$. We change the summation order and proceed exactly as with $D_{5,2}^{\mathsf{b}}$. Indeed,

$$D_{5,3}^{\mathsf{b}} = \frac{1}{\sqrt{n}} \sum_{\ell=n+1}^{2n-1} \alpha_\ell(w_\ell - 1)\Big( \sum_{k=\ell+1}^{2n} (\mathbf{A}_k - \bar{\mathbf{A}})\Gamma_{\ell+1:k-1}\Big) A_\ell J_{\ell-1}^{\mathsf{b},1} \ ,$$

and

$$\mathbb{E}^{\mathsf{b}}[\|D_{5,3}^{\mathsf{b}}\|^2] \lesssim \frac{1}{n} \sum_{\ell=n+1}^{2n-1} \frac{\alpha_\ell \log n}{a} \mathrm{C}_{\mathbf{A}}^2 \Big( \frac{\alpha_\ell^2\|\varepsilon\|_\infty^2 \mathrm{C}_{\mathbf{A}}^2}{a^2} + \mathrm{C}_{\mathbf{A}}^4 \prod_{j=1}^{\ell-1}(1-a\alpha_j/4)^2\|\theta_0 - \theta^\star\|^2 \Big)$$

$$\lesssim \frac{\mathrm{C}_{\mathbf{A}}^4 \|\varepsilon\|_\infty^2 \log n}{n^{3/2}a^3} + \frac{\mathrm{C}_{\mathbf{A}}^6 \log n}{na} \exp\{-(a/2)\sum_{j=1}^n \alpha_j\}\|\theta_0 - \theta^\star\|^2 \ .$$

It remains to upper bound $D_{5,4}^{\mathsf{b}}$. Proceeding as above, we change the summation order, and obtain

$$D_{5,4}^{\mathsf{b}} = \frac{1}{\sqrt{n}} \sum_{\ell=n+1}^{2n-1} \alpha_\ell(w_\ell - 1)\Big( \sum_{k=\ell+1}^{2n} (\mathbf{A}_k - \bar{\mathbf{A}})\Gamma_{\ell+1:k-1}^{\mathsf{b}}\Big) A_\ell J_{\ell-1}^{\mathsf{b},2} \ .$$

Applying Minkowski's inequality, we get

$$
(\mathbb{E}^{\mathsf{b}}[\|D_{5,4}^{\mathsf{b}}\|^2])^{1/2} \lesssim \frac{C_{\mathbf{A}}^2}{\sqrt{n}} \sum_{\ell=n+1}^{2n-1} \alpha_\ell \sum_{k=\ell+1}^{2n} (\mathbb{E}^{\mathsf{b}}[\|\Gamma_{\ell+1:k-1}^{\mathsf{b}}\|^2])^{1/2} (\mathbb{E}^{\mathsf{b}}[\|J_{\ell-1}^{\mathsf{b},2}\|^2])^{1/2}
$$

$$
\lesssim \frac{C_{\mathbf{A}}^2}{\sqrt{n}} \sum_{\ell=n+1}^{2n-1} \alpha_\ell^{5/2} \sum_{k=\ell+1}^{2n} \exp\Big\{-\frac{a}{4}\sum_{j=\ell+1}^{k-1}\alpha_j\Big\} \frac{\|\varepsilon\|_\infty C_{\mathbf{A}}^2}{a^{3/2}}
$$

$$
+ \frac{C_{\mathbf{A}}^5}{\sqrt{n}} \sum_{\ell=n+1}^{2n-1} \alpha_\ell \sum_{k=\ell+1}^{2n-1} \exp\Big\{-\frac{a}{4}\sum_{j=1}^{k-1}\alpha_j\Big\} \|\theta_0 - \theta^\star\|
$$

$$
\lesssim \frac{C_{\mathbf{A}}^4 \|\varepsilon\|_\infty}{\sqrt{n}a^{5/2}} \sum_{\ell=n+1}^{2n-1} \alpha_\ell^{3/2} + \frac{C_{\mathbf{A}}^5}{\sqrt{n}a} \exp\Big\{-\frac{a}{4}\sum_{j=1}^{n}\alpha_j\Big\} \|\theta_0 - \theta^\star\|
$$

$$
\lesssim \frac{C_{\mathbf{A}}^4 \|\varepsilon\|_\infty}{n^{1/4}a^{5/2}} + \frac{C_{\mathbf{A}}^5}{\sqrt{n}a} \exp\Big\{-\frac{a}{4}\sum_{j=1}^{n}\alpha_j\Big\} \|\theta_0 - \theta^\star\| \,.
$$

Now the result follows from the representation (49) and combinations of the above bounds for $D_1^{\mathsf{b}} - D_5^{\mathsf{b}}$.

### C.6 Proof of Proposition 6

Consider the sequences of weights

$$
(w_1, \ldots, w_{i-1}, w_i, w_{i+1}, \ldots, w_{2n}) \text{ and } (w_1, \ldots, w_{i-1}, w_i', w_{i+1}, \ldots, w_{2n}) \,, \tag{61}
$$

which differs only in position $i$, $n + 1 \le i \le 2n$, with $w_i'$ being an independent copy of $w_i$. Consider the associated SA processes

$$
\begin{aligned}
\theta_k^{\mathsf{b}} &= \theta_{k-1}^{\mathsf{b}} - \alpha_k w_k \{\mathbf{A}(Z_k)\theta_{k-1} - \mathbf{b}(Z_k)\}\,, \quad k \ge n+1\,, \quad \theta_n^{\mathsf{b}} = \theta_n \in \mathbb{R}^d \\
\theta_k^{(\mathsf{b},i)} &= \theta_{k-1}^{(\mathsf{b},i)} - \alpha_k w_k^{(i)} \{\mathbf{A}(Z_k)\theta_{k-1}^{(\mathsf{b},i)} - \mathbf{b}(Z_k)\}\,, \quad k \ge n+1\,, \quad \theta_n^{(\mathsf{b},i)} = \theta_n \in \mathbb{R}^d\,,
\end{aligned} \tag{62}
$$

where $w_k^{(i)} = w_k$ for $k \ne i$ and $w_i^{(i)} = w_i'$. Respective random variables $D^{\mathsf{b}}$ and $D^{(\mathsf{b},i)}$ are based on the first and second sequences from (61), respectively, and are constructed according to the equation (48). From the above representations we easily observe that $\theta_k^{\mathsf{b}} = \theta_k^{(\mathsf{b},i)}$ for $k < i$, moreover,

$$
\begin{aligned}
\theta_i^{\mathsf{b}} - \theta_i^{(\mathsf{b},i)} &= -\alpha_i(w_i - w_i')\{\mathbf{A}(Z_i))\theta_{i-1}^{\mathsf{b}} - \mathbf{b}(Z_i)\} \\
&= -\alpha_i(w_i - w_i')\{\mathbf{A}(Z_i))(\theta_{i-1}^{\mathsf{b}} - \theta_{i-1}) - \mathbf{b}(Z_i)\} \\
&= -\alpha_i(w_i - w_i')\{\mathbf{A}(Z_i))(\theta_{i-1}^{\mathsf{b}} - \theta_{i-1}) + \tilde\varepsilon_i\} \,.
\end{aligned}
$$

where $\varepsilon_i = \varepsilon(Z_i)$ and $\varepsilon_i' = \varepsilon(Z_i')$. From the above representation we get, applying Lemma 9 and (59), that

$$
\{\mathbb{E}^{\mathsf{b}}[\|\theta_i^{\mathsf{b}} - \theta_i^{(\mathsf{b},i)}\|^2]\}^{1/2} \lesssim \alpha_i\, C_{\mathbf{A}} \{\mathbb{E}^{\mathsf{b}}[\|\theta_i^{\mathsf{b}} - \theta_i\|^2]\}^{1/2} + \alpha_i\, C_{\mathbf{A}} \|\tilde\varepsilon_i\| \tag{63}
$$

$$
\lesssim \alpha_i\, C_{\mathbf{A}} \|\varepsilon\|_\infty + \frac{\alpha_i^{3/2}\|\varepsilon\|_\infty C_{\mathbf{A}}^2}{a^{3/2}} + (C_{\mathbf{A}}^2 \vee 1)\sqrt{i} \prod_{j=1}^{i}(1 - a\alpha_j/4)\|\theta_0 - \theta^\star\|
$$

$$
\lesssim \frac{\alpha_i\|\varepsilon\|_\infty C_{\mathbf{A}}^2}{a} + (C_{\mathbf{A}}^2 \vee 1)\sqrt{i} \prod_{j=1}^{i}(1 - a\alpha_j/4)\|\theta_0 - \theta^\star\| \,,
$$

where for the last line we have additionally assumed that $\alpha_i \lesssim a$ for $i \ge n$. Moreover, for any $j > i$ one observes, expanding (62), that

$$
\theta_j^{\mathsf{b}} - \theta_j^{(\mathsf{b},i)} = \Big\{ \prod_{k=i+1}^{j} (\mathrm{I} - \alpha_k w_k \mathbf{A}(Z_k)) \Big\}(\theta_i^{\mathsf{b}} - \theta_i^{(\mathsf{b},i)}) = \Gamma_{i+1:j}^{\mathsf{b}} (\theta_i^{\mathsf{b}} - \theta_i^{(\mathsf{b},i)}) \,.
$$

Thus, similarly to (41), we obtain that

$$\{\mathbb{E}^{\mathsf{b}}[\|D^{\mathsf{b}} - D^{(\mathsf{b},i)}\|^2]\}^{1/2} \le \sum_{j=1}^{5} \{\mathbb{E}^{\mathsf{b}}[\|D_j^{\mathsf{b}} - D_j^{(\mathsf{b},i)}\|^2]\}^{1/2} ,$$

and bound the respective differences separately. By the construction of the process above, we note that $D_1^{\mathsf{b}} = D_1^{(\mathsf{b},i)}$. Proceeding further, and using the equation (48), we obtain that

$$\{\mathbb{E}^{\mathsf{b}}[\|D_2^{\mathsf{b}} - D_2^{(\mathsf{b},i)}\|^2]\}^{1/2} = \frac{1}{\sqrt{n}\alpha_{2n}} \{\mathbb{E}^{\mathsf{b}}[\|\theta_{2n}^{\mathsf{b}} - \theta_{2n}^{(\mathsf{b},i)}\|^2]\}^{1/2}$$

$$\le \frac{1}{\sqrt{n}\alpha_{2n}} \{\mathbb{E}^{\mathsf{b}}[\|\Gamma_{i+1:2n}^{\mathsf{b}}\|^2]\}^{1/2} \{\mathbb{E}^{\mathsf{b}}[\|\theta_i^{\mathsf{b}} - \theta_i^{(\mathsf{b},i)}\|^2]\}^{1/2}$$

$$\lesssim \frac{\alpha_i\|\varepsilon\|_\infty \mathrm{C}_{\mathbf{A}}^2}{\sqrt{n}\alpha_{2n}a} \exp\Big\{-\frac{a}{4}\sum_{j=i+1}^{2n}\alpha_j\Big\} + \frac{(\mathrm{C}_{\mathbf{A}}^2\vee 1)\sqrt{i}}{\sqrt{n}\alpha_{2n}} \exp\Big\{-\frac{a}{4}\sum_{j=1}^{2n}\alpha_j\Big\}\|\theta_0 - \theta^\star\| .$$

Thus, taking sum for $i$ from $n+1$ to $2n$, and applying Lemma 2, we get that

$$\sum_{i=n+1}^{2n} \{\mathbb{E}^{\mathsf{b}}[\|D_2^{\mathsf{b}} - D_2^{(\mathsf{b},i)}\|^2]\}^{1/2} \lesssim \sum_{i=n+1}^{2n} \frac{\alpha_i\|\varepsilon\|_\infty \mathrm{C}_{\mathbf{A}}^2}{\sqrt{n}\alpha_{2n}a} \exp\Big\{-\frac{a}{4}\sum_{j=i+1}^{2n}\alpha_j\Big\}$$

$$+ \sum_{i=n+1}^{2n} \frac{(\mathrm{C}_{\mathbf{A}}^2\vee 1)\sqrt{i}}{\sqrt{n}\alpha_{2n}} \exp\Big\{-\frac{a}{4}\sum_{j=1}^{2n}\alpha_j\Big\}\|\theta_0 - \theta^\star\|$$

$$\lesssim \frac{\|\varepsilon\|_\infty \mathrm{C}_{\mathbf{A}}^2}{\sqrt{n}\alpha_{2n}a^2} + \frac{(\mathrm{C}_{\mathbf{A}}^2\vee 1)n}{\alpha_{2n}} \exp\Big\{-\frac{a}{4}\sum_{j=1}^{2n}\alpha_j\Big\}\|\theta_0 - \theta^\star\|$$

$$\lesssim \frac{\|\varepsilon\|_\infty \mathrm{C}_{\mathbf{A}}^2}{a^2} + (\mathrm{C}_{\mathbf{A}}^2\vee 1)n^{3/2} \exp\Big\{-\frac{a}{4}\sum_{j=1}^{2n}\alpha_j\Big\}\|\theta_0 - \theta^\star\|$$

$$\lesssim \frac{\|\varepsilon\|_\infty \mathrm{C}_{\mathbf{A}}^2}{a^2} + (\mathrm{C}_{\mathbf{A}}^2\vee 1)n^{3/4} \exp\Big\{-\frac{a}{4}\sum_{j=1}^{n}\alpha_j\Big\}\|\theta_0 - \theta^\star\| . \quad (64)$$

Here in the last line above we used a particular form $\alpha_k = c_0/\sqrt{k}$, and relied on the bound

$$n^{3/4} \exp\Big\{-\frac{a}{4}\sum_{j=n+1}^{2n}\alpha_j\Big\} \le 1 ,$$

which is guaranteed by the lower bound on the trajectory length $n$ of the form

$$\frac{\sqrt{n}}{\log n} \ge \frac{3}{2(\sqrt{2}-1)ac_0} .$$

The latter condition is guaranteed by A4. Now we proceed with $D_3^{\mathsf{b}} - D_3^{(\mathsf{b},i)}$. Using its definition in (48), we get

$$D_3^{\mathsf{b}} - D_3^{(\mathsf{b},i)} = \frac{1}{\sqrt{n}}(w_i - w_i')\mathbf{A}_i(\theta_{i-1}^{\mathsf{b}} - \theta^\star) + \frac{1}{\sqrt{n}}\sum_{k=i+1}^{2n}(w_k - 1)\mathbf{A}_k(\theta_{k-1}^{\mathsf{b}} - \theta_{k-1}^{(\mathsf{b},i)}) .$$

Since the latter term is a martingale-difference, we obtain that

$$\mathbb{E}^{\mathsf{b}}[\|D_3^{\mathsf{b}} - D_3^{(\mathsf{b},i)}\|^2] \lesssim \frac{\mathrm{C}_{\mathbf{A}}^2}{n}\mathbb{E}^{\mathsf{b}}[\|\theta_{i-1}^{\mathsf{b}} - \theta^\star\|^2] + \frac{\mathrm{C}_{\mathbf{A}}^2}{n}\sum_{k=i+1}^{2n}\mathbb{E}^{\mathsf{b}}[\|\theta_{k-1}^{\mathsf{b}} - \theta_{k-1}^{(\mathsf{b},i)}\|^2]$$

$$\lesssim \frac{\mathrm{C}_{\mathbf{A}}^2}{n}\mathbb{E}^{\mathsf{b}}[\|\theta_{i-1}^{\mathsf{b}} - \theta_{i-1}\|^2] + \frac{\mathrm{C}_{\mathbf{A}}^2}{n}\|\theta_{i-1} - \theta^\star\|^2 + \frac{\mathrm{C}_{\mathbf{A}}^2}{n}\sum_{k=i+1}^{2n}\mathbb{E}^{\mathsf{b}}[\|\theta_{k-1}^{\mathsf{b}} - \theta_{k-1}^{(\mathsf{b},i)}\|^2]$$

$$\lesssim \frac{\mathrm{C}_{\mathbf{A}}^2}{n}\mathbb{E}^{\mathsf{b}}[\|\theta_{i-1}^{\mathsf{b}} - \theta_{i-1}\|^2] + \frac{\mathrm{C}_{\mathbf{A}}^2}{n}\|\theta_{i-1} - \theta^\star\|^2 + \frac{\mathrm{C}_{\mathbf{A}}^2}{n}\sum_{k=i+1}^{2n}\mathbb{E}^{\mathsf{b}}[\|\Gamma_{i+1:k-1}^{\mathsf{b}}\|^2\|\theta_i^{\mathsf{b}} - \theta_i^{(\mathsf{b},i)}\|^2] .$$

Hence we obtain, using (63) together with the definition of $\Omega_1$, that

$$\mathbb{E}^{\mathsf{b}}[\|D_3^{\mathsf{b}} - D_3^{(\mathsf{b},i)}\|^2] \lesssim \frac{\alpha_{i-1}\|\varepsilon\|_\infty^2 \mathrm{C}_{\mathbf{A}}^4}{na^3} + \frac{\alpha_{i-1}\|\varepsilon\|_\infty^2 \mathrm{C}_{\mathbf{A}}^2 \log^2 n}{an} + \frac{\mathrm{C}_{\mathbf{A}}^2}{n}\exp\Big\{-a\sum_{\ell=1}^{i}\alpha_\ell\Big\}\|\theta_0 - \theta^\star\|^2$$

$$+ \underbrace{\frac{\mathrm{C}_{\mathbf{A}}^2}{n}\left(\frac{\alpha_i^2\|\varepsilon\|_\infty^2 \mathrm{C}_{\mathbf{A}}^4}{a^2} + (\mathrm{C}_{\mathbf{A}}^4 \vee 1)i\prod_{j=1}^{i}(1 - a\alpha_j/4)^2\|\theta_0 - \theta^\star\|^2\right)\sum_{k=i+1}^{2n}\exp\Big\{-\frac{a}{2}\sum_{j=i+1}^{k-1}\alpha_j\Big\}}_{T_1} \;.$$

$$(65)$$

Considering the latter term in the sum, we obtain

$$T_1 \lesssim \frac{\alpha_i^2\|\varepsilon\|_\infty^2 \mathrm{C}_{\mathbf{A}}^6}{n\alpha_{2n}a^2}\sum_{k=i+1}^{2n}\alpha_k\exp\Big\{-\frac{a}{2}\sum_{j=i+1}^{k-1}\alpha_j\Big\} + \frac{(\mathrm{C}_{\mathbf{A}}^6 \vee 1)i}{n}\sum_{k=i+1}^{2n}\exp\Big\{-\frac{a}{2}\sum_{j=1}^{k-1}\alpha_j\Big\}\|\theta_0 - \theta^\star\|^2$$

$$\lesssim \frac{\alpha_i\|\varepsilon\|_\infty^2 \mathrm{C}_{\mathbf{A}}^6}{na^3} + \frac{(\mathrm{C}_{\mathbf{A}}^6 \vee 1)i}{n\alpha_{2n}}\exp\Big\{-\frac{a}{2}\sum_{j=1}^{i}\alpha_j\Big\}\|\theta_0 - \theta^\star\|^2 \;.$$

Thus, summing the equations (65) for $i$ from $n+1$ to $2n$, we obtain that

$$\sum_{i=n+1}^{2n}\{\mathbb{E}^{\mathsf{b}}[\|D_3^{\mathsf{b}} - D_3^{(\mathsf{b},i)}\|^2]\}^{1/2} \lesssim \sum_{i=n+1}^{2n}\left(\frac{\sqrt{\alpha_{i-1}}\|\varepsilon\|_\infty \mathrm{C}_{\mathbf{A}}^2}{\sqrt{n}a^{3/2}} + \frac{\sqrt{\alpha_{i-1}}\|\varepsilon\|_\infty \mathrm{C}_{\mathbf{A}}\log n}{\sqrt{an}} + \frac{\sqrt{\alpha_i}\|\varepsilon\|_\infty \mathrm{C}_{\mathbf{A}}^3}{\sqrt{n}a^{3/2}}\right)$$

$$+ \frac{\mathrm{C}_{\mathbf{A}}^3 \vee 1}{\sqrt{n}}\sum_{i=n+1}^{2n}\exp\Big\{-\frac{a}{2}\sum_{\ell=1}^{i}\alpha_\ell\Big\}\|\theta_0 - \theta^\star\|$$

$$\lesssim \frac{(\mathrm{C}_{\mathbf{A}}^3 \vee 1)\|\varepsilon\|_\infty}{a^{3/2}}n^{1/4}\log n + \frac{\mathrm{C}_{\mathbf{A}}^3 \vee 1}{a}\exp\Big\{-\frac{a}{2}\sum_{\ell=1}^{n}\alpha_\ell\Big\}\|\theta_0 - \theta^\star\| \;.$$

$$(66)$$

Using now the definition of $D_4^{\mathsf{b}}$ in (48) and Minkowski's inequality, we write

$$\{\mathbb{E}^{\mathsf{b}}[\|D_4^{\mathsf{b}} - D_4^{(\mathsf{b},i)}\|^2]\}^{1/2} \leq \frac{1}{\sqrt{n}}\{\mathbb{E}^{\mathsf{b}}[\|\theta_i^{\mathsf{b}} - \theta_i^{(\mathsf{b},i)}\|^2]\}^{1/2}\sum_{k=i+1}^{2n}\left(\frac{1}{\alpha_k} - \frac{1}{\alpha_{k-1}}\right)\exp\Big\{-\frac{a}{4}\sum_{j=i+1}^{k-1}\alpha_j\Big\}$$

$$\lesssim \frac{\alpha_i\|\varepsilon\|_\infty \mathrm{C}_{\mathbf{A}}^2}{\sqrt{n}a}\sum_{k=i+1}^{2n}\alpha_k\exp\Big\{-\frac{a}{4}\sum_{j=i+1}^{k-1}\alpha_j\Big\}$$

$$+ \frac{(\mathrm{C}_{\mathbf{A}}^2 \vee 1)\sqrt{i}}{\sqrt{n}}\sum_{k=i+1}^{2n}\alpha_k\exp\Big\{-\frac{a}{4}\sum_{j=1}^{k-1}\alpha_j\Big\}\|\theta_0 - \theta^\star\|$$

$$\lesssim \frac{\alpha_i\|\varepsilon\|_\infty \mathrm{C}_{\mathbf{A}}^2}{\sqrt{n}a^2} + \frac{(\mathrm{C}_{\mathbf{A}}^2 \vee 1)\sqrt{i}}{\sqrt{n}a}\exp\Big\{-\frac{a}{4}\sum_{j=1}^{i}\alpha_j\Big\}\|\theta_0 - \theta^\star\| \;.$$

Thus, taking sum for $i$ from $n+1$ to $2n$, we get

$$\sum_{i=n+1}^{2n}\{\mathbb{E}^{\mathsf{b}}[\|D_4^{\mathsf{b}} - D_4^{(\mathsf{b},i)}\|^2]\}^{1/2} \lesssim \sum_{i=n+1}^{2n}\frac{\alpha_i\|\varepsilon\|_\infty \mathrm{C}_{\mathbf{A}}^2}{\sqrt{n}a^2} + \sum_{i=n+1}^{2n}\frac{(\mathrm{C}_{\mathbf{A}}^2 \vee 1)\sqrt{i}}{\sqrt{n}a}\exp\Big\{-\frac{a}{4}\sum_{j=1}^{i}\alpha_j\Big\}\|\theta_0 - \theta^\star\|$$

$$\lesssim \frac{\|\varepsilon\|_\infty \mathrm{C}_{\mathbf{A}}^2}{a^2} + \frac{(\mathrm{C}_{\mathbf{A}}^2 \vee 1)\sqrt{n}}{a^2}\exp\Big\{-\frac{a}{4}\sum_{j=1}^{n}\alpha_i\Big\}\|\theta_0 - \theta^\star\| \;. \quad (67)$$

Similarly, with the definition of $D_5^{\mathsf{b}}$ in (48), we write

$$
\begin{aligned}
D_5^{\mathsf{b}} - D_5^{(\mathsf{b},i)} &= \frac{1}{\sqrt{n}} \sum_{k=i+1}^{2n} (\mathbf{A}_k - \bar{\mathbf{A}})(\theta_{k-1}^{\mathsf{b}} - \theta_{k-1}^{(\mathsf{b},i)}) = \frac{1}{\sqrt{n}} \left\{ \sum_{k=i+1}^{2n} (\mathbf{A}_k - \bar{\mathbf{A}})\Gamma_{i+1:k-1}^{\mathsf{b}} \right\} (\theta_i^{\mathsf{b}} - \theta_i^{(\mathsf{b},i)}) \\
&= \underbrace{\frac{1}{\sqrt{n}} \left\{ \sum_{k=i+1}^{2n} (\mathbf{A}_k - \bar{\mathbf{A}})\Gamma_{i+1:k-1} \right\} (\theta_i^{\mathsf{b}} - \theta_i^{(\mathsf{b},i)})}_{T_2} \\
&\quad + \underbrace{\frac{1}{\sqrt{n}} \left\{ \sum_{k=i+1}^{2n} (\mathbf{A}_k - \bar{\mathbf{A}})(\Gamma_{i+1:k-1}^{\mathsf{b}} - \Gamma_{i+1:k-1}) \right\} (\theta_i^{\mathsf{b}} - \theta_i^{(\mathsf{b},i)})}_{T_3} .
\end{aligned}
$$

Now we bound the terms $T_2$ and $T_3$ separately. Indeed, for the term $T_2$ we get, applying the definition of the set $\Omega_4$, that

$$
\begin{aligned}
\mathbb{E}^{\mathsf{b}}[\|T_2\|^2] &\lesssim \frac{1}{n} \left( \frac{\mathrm{C}_{\mathbf{A}}^2 \log n}{a\alpha_i} + \mathrm{C}_{\mathbf{A}}^2 \log^2 n \right) \mathbb{E}^{\mathsf{b}}[\|\theta_i^{\mathsf{b}} - \theta_i^{(\mathsf{b},i)}\|^2] \\
&\lesssim \frac{\mathrm{C}_{\mathbf{A}}^2 \log n}{na\alpha_i} \mathbb{E}^{\mathsf{b}}[\|\theta_i^{\mathsf{b}} - \theta_i^{(\mathsf{b},i)}\|^2] .
\end{aligned}
$$

In the above bounds we have used that $\alpha_\ell \leq \frac{1}{a \log n}$. For the term $T_3$ we get, applying Lemma 10, that for any vector $v \in \mathbb{R}^d$,

$$
\begin{aligned}
\sum_{k=i+1}^{2n} (\mathbf{A}_k - \bar{\mathbf{A}})(\Gamma_{i+1:k-1}^{\mathsf{b}} - \Gamma_{i+1:k-1})v &= \sum_{k=i+1}^{2n} \sum_{\ell=i+1}^{k-1} (\mathbf{A}_k - \bar{\mathbf{A}})\Gamma_{\ell+1:k-1}\alpha_\ell(w_\ell - 1)\mathbf{A}_\ell\Gamma_{i+1:\ell-1}^{\mathsf{b}} \, v \\
&= \sum_{\ell=i+1}^{2n-1} \alpha_\ell(w_\ell - 1)\left\{ \sum_{k=\ell+1}^{2n} (\mathbf{A}_k - \bar{\mathbf{A}})\Gamma_{\ell+1:k-1} \right\} \mathbf{A}_\ell\Gamma_{i+1:\ell-1}^{\mathsf{b}} \, v .
\end{aligned}
$$

From the above representation we obtain, using the definition of the set $\Omega_4$, that

$$
\begin{aligned}
\mathbb{E}^{\mathsf{b}}[\|T_3\|^2] &\lesssim \frac{\mathrm{C}_{\mathbf{A}}^2}{n} \sum_{\ell=i+1}^{2n-1} \alpha_\ell^2 \left( \frac{\mathrm{C}_{\mathbf{A}}^2 \log n}{a\alpha_\ell} + \mathrm{C}_{\mathbf{A}}^2 \log^2 n \right) \exp\left\{ -\frac{a}{4} \sum_{j=i+1}^{\ell-1} \alpha_j \right\} \mathbb{E}^{\mathsf{b}}[\|\theta_i^{\mathsf{b}} - \theta_i^{(\mathsf{b},i)}\|^2] \\
&\lesssim \frac{\mathrm{C}_{\mathbf{A}}^4}{n} \sum_{\ell=i+1}^{2n-1} \frac{\alpha_\ell \log n}{a} \exp\left\{ -\frac{a}{4} \sum_{j=i+1}^{\ell-1} \alpha_j \right\} \mathbb{E}^{\mathsf{b}}[\|\theta_i^{\mathsf{b}} - \theta_i^{(\mathsf{b},i)}\|^2] \\
&\lesssim \frac{\mathrm{C}_{\mathbf{A}}^4 \log n}{na^2} \mathbb{E}^{\mathsf{b}}[\|\theta_i^{\mathsf{b}} - \theta_i^{(\mathsf{b},i)}\|^2] .
\end{aligned}
$$

Combining the above bounds, we obtain that

$$
\begin{aligned}
\mathbb{E}^{\mathsf{b}}[\|D_5^{\mathsf{b}} - D_5^{(\mathsf{b},i)}\|^2] &\lesssim \frac{\mathrm{C}_{\mathbf{A}}^2 \log n}{na} \left( \frac{1}{\alpha_i} + \frac{\mathrm{C}_{\mathbf{A}}^2}{a} \right) \mathbb{E}^{\mathsf{b}}[\|\theta_i^{\mathsf{b}} - \theta_i^{(\mathsf{b},i)}\|^2] \\
&\lesssim \frac{\mathrm{C}_{\mathbf{A}}^2 \log n}{na\alpha_i} \mathbb{E}^{\mathsf{b}}[\|\theta_i^{\mathsf{b}} - \theta_i^{(\mathsf{b},i)}\|^2] ,
\end{aligned}
$$

where we have additionally used that $\alpha_i \leq a/\mathrm{C}_{\mathbf{A}}^2$. Thus, using the upper bound (63), we obtain that

$$\sum_{i=n+1}^{2n} \{\mathbb{E}^{\mathsf{b}}[\|D_5^{\mathsf{b}} - D_5^{(\mathsf{b},i)}\|^2]\}^{1/2}$$

$$\lesssim \sum_{i=n+1}^{2n} \frac{\mathrm{C}_{\mathbf{A}}\sqrt{\log n}}{\sqrt{\alpha_i a n}} \frac{\alpha_i \|\varepsilon\|_\infty \mathrm{C}_{\mathbf{A}}^2}{a} + \sum_{i=n+1}^{2n} \frac{\mathrm{C}_{\mathbf{A}}\sqrt{\log n}}{\sqrt{\alpha_i a n}} (\mathrm{C}_{\mathbf{A}}^2 \vee 1)\sqrt{i} \prod_{j=1}^{i}(1 - a\alpha_j/4)\|\theta_0 - \theta^\star\|$$

$$\lesssim \frac{\mathrm{C}_{\mathbf{A}}^3 \|\varepsilon\|_\infty}{a^{3/2}} n^{1/4}\sqrt{\log n}$$

$$+ \frac{(\mathrm{C}_{\mathbf{A}}^3 \vee 1)\sqrt{\log n}}{a^{1/2}\alpha_{2n}^{3/2}} \exp\Big\{-\frac{a}{4}\sum_{j=1}^{n}\alpha_j\Big\}\Big\{\sum_{i=n+1}^{2n}\alpha_i \prod_{j=n+1}^{i}(1 - a\alpha_j/4)\Big\}\|\theta_0 - \theta^\star\|$$

$$\lesssim \frac{\mathrm{C}_{\mathbf{A}}^3 \|\varepsilon\|_\infty}{a^{3/2}} n^{1/4}\sqrt{\log n} + \frac{(\mathrm{C}_{\mathbf{A}}^3 \vee 1)n^{3/4}\sqrt{\log n}}{a^{3/2}} \exp\Big\{-\frac{a}{4}\sum_{j=1}^{n}\alpha_j\Big\}\|\theta_0 - \theta^\star\| . \tag{68}$$

Now it remains to combine the bounds outlined above in (64), (66), (67), and (68), and the statement follows.

## D Proof of stability of random matrix product

### D.1 Proof of Proposition 1

The fact that there exists a unique matrix $Q$, such that the following Lyapunov equation holds:

$$\bar{\mathbf{A}}^\top Q + Q\bar{\mathbf{A}} = P , \tag{69}$$

follows directly from [54, Lemma 9.1, p. 140]. In order to show the second part of the statement, we note that for any non-zero vector $x \in \mathbb{R}^d$, we have

$$\frac{x^\top (I - \alpha\bar{\mathbf{A}})^\top Q(I - \alpha\bar{\mathbf{A}})x}{x^\top Q x} = 1 - \alpha\frac{x^\top (\bar{\mathbf{A}}^\top Q + Q\bar{\mathbf{A}})x}{x^\top Q x} + \alpha^2\frac{x^\top \bar{\mathbf{A}}^\top Q\bar{\mathbf{A}}x}{x^\top Q x}$$

$$= 1 - \alpha\frac{x^\top P x}{x^\top Q x} + \alpha^2\frac{x^\top \bar{\mathbf{A}}^\top Q\bar{\mathbf{A}}x}{x^\top Q x}$$

$$\leq 1 - \alpha\frac{\lambda_{\min}(P)}{\|Q\|} + \alpha^2\frac{\|\bar{\mathbf{A}}\|_Q^2}{\lambda_{\min}(Q)}$$

$$\leq 1 - \alpha a ,$$

where we set

$$a = \frac{1}{2}\frac{\lambda_{\min}(P)}{\lambda_{\max}(Q)} ,$$

and used the fact that $\alpha \leq \alpha_\infty$, where $\alpha_\infty$ is defined in (7).

### D.2 Proofs for auxiliary results on products of random matrix

In order to bound the moment $\mathbb{E}[\|\theta_k - \theta^\star\|^p]$, we first prove a stability results on the products of random matrices $\Gamma_{m:k}$ arising in the LSA recursion. Towards this aim we first introduce some notations and definitions. For a matrix $B \in \mathbb{R}^{d\times d}$ we denote by $(\sigma_\ell(B))_{\ell=1}^d$ its singular values. For $q \geq 1$, the Shatten $q$-norm of $B$ is denoted by $\|B\|_q = \{\sum_{\ell=1}^d \sigma_\ell^q(B)\}^{1/q}$. For $q, p \geq 1$ and a random matrix $\mathbf{X}$ we write $\|\mathbf{X}\|_{q,p} = \{\mathbb{E}[\|\mathbf{X}\|_q^p]\}^{1/p}$. Our proof technique is based on the stability results arising in [29], see also [16].

**Lemma 12** (Proposition 15 in [16]). *Let $\{\mathbf{Y}_\ell\}_{\ell\in\mathbb{N}}$ be an independent sequence and $P$ be a positive definite matrix. Assume that for each $\ell \in \mathbb{N}$ there exist $m_\ell \in (0,1)$ and $\sigma_\ell > 0$ such that $\|\mathbb{E}[\mathbf{Y}_\ell]\|_P^2 \leq 1 - m_\ell$ and $\|\mathbf{Y}_\ell - \mathbb{E}[\mathbf{Y}_\ell]\|_P \leq \sigma_\ell$ almost surely. Define $\mathbf{Z}_k = \prod_{\ell=0}^k \mathbf{Y}_\ell = \mathbf{Y}_k\mathbf{Z}_{k-1}$, for $k \geq 1$ and starting from $\mathbf{Z}_0$. Then, for any $2 \leq q \leq p$ and $k \geq 1$,*

$$\|\mathbf{Z}_k\|_{p,q}^2 \leq \kappa_P \prod_{\ell=1}^k (1 - m_\ell + (p-1)\sigma_\ell^2)\|P^{1/2}\mathbf{Z}_0 P^{-1/2}\|_{p,q}^2 , \tag{70}$$

*where we recall that* $\kappa_P = \lambda_{\min}^{-1}(P)\lambda_{\max}(P)$.

Now we aim to bound $\Gamma_{m:k}$ defined in (27) using Lemma 12. We identify the latter with $\prod_{\ell=m}^{k} \mathbf{Y}_\ell$, where $\mathbf{Y}_\ell = \mathrm{I} - \alpha_\ell \mathbf{A}_\ell, \ell \geq 1$, and $\mathbf{Y}_0 = \mathrm{I}$. Applying the bound (8), we get $\|\mathbb{E}[\mathbf{Y}_\ell]\|_Q^2 = \|\mathrm{I} - \alpha_\ell \bar{\mathbf{A}}\|_Q^2 \leq 1 - a\alpha_\ell$. Further, assumption A2 implies that almost surely,

$$\|\mathbf{Y}_\ell - \mathbb{E}[\mathbf{Y}_\ell]\|_Q = \alpha_\ell \|\mathbf{A}_\ell - \bar{\mathbf{A}}\|_Q \leq \alpha_\ell \sqrt{\kappa_Q}\, \mathrm{C}_\mathbf{A} = b_Q \alpha_\ell \ .$$

Therefore, (70) holds with $m_\ell = a\alpha_\ell$ and $\sigma_\ell = b_Q \alpha_\ell$. As $\|\mathrm{I}\|_p = d^{1/p}$, we obtain the following corollary.

**Corollary 3.** *Assume A1 and A2. Then, for any* $\alpha_\ell \in [0, \alpha_\infty]$, $2 \leq q \leq p$, *and* $1 \leq m \leq k$, *it holds*

$$\mathbb{E}^{1/q}\left[\|\Gamma_{m:k}\|^q\right] \leq \|\Gamma_{m:k}\|_{p,q} \leq \sqrt{\kappa_Q} d^{1/p} \prod_{\ell=m}^{k} \left(1 - a\alpha_\ell + (p-1)b_Q^2\alpha_\ell^2\right) \ ,$$

*where* $\alpha_\infty$ *was defined in (7), and* $b_Q = \sqrt{\kappa_Q}\, \mathrm{C}_\mathbf{A}$.

**Corollary 4.** *Assume A1, A2, and A3. Then for any* $2 \leq q \leq \log n$, *and any* $k \geq n$, $1 \leq m \leq k$, *it holds that*

$$\mathbb{E}^{1/q}\left[\|\Gamma_{m:k}\|^q\right] \leq \sqrt{\kappa_Q}\mathrm{e}\exp\left\{-(a/2)\sum_{\ell=m}^{k}\alpha_\ell\right\} \ , \tag{71}$$

*where* $\alpha_\infty$ *is defined in (7). Moreover,*

$$\mathbb{E}^{1/q}\left[\|\Gamma_{m:k}\|^q\right] \leq \sqrt{\kappa_Q}\mathrm{e}\prod_{\ell=m}^{k}\left(1 - \frac{a\alpha_\ell}{4}\right) \tag{72}$$

*Proof.* We first apply the result of Corollary 3. Indeed, for $k \geq n$, and any $2 \leq q \leq p$, it holds, setting $b_Q = \sqrt{\kappa_Q}\, \mathrm{C}_\mathbf{A}$, that

$$\mathbb{E}^{1/q}\left[\|\Gamma_{m:k}\|^q\right] \leq \sqrt{\kappa_Q} d^{1/p} \prod_{\ell=m}^{k} \left(1 - a\alpha_\ell + (p-1)b_Q^2\alpha_\ell^2\right)$$

$$\leq \sqrt{\kappa_Q} d^{1/p} \exp\left\{-a\sum_{\ell=m}^{k}\alpha_\ell + (p-1)b_Q^2\sum_{\ell=m}^{k}\alpha_\ell^2\right\} \ .$$

Note that, setting $p = \log n$, and provided that $n$ satisfies (9), we easily obtain that, for $\ell \geq n/2$,

$$(\log n)b_Q^2\alpha_\ell^2 \leq a\alpha_\ell/2 \ . \tag{73}$$

Hence, for $m \geq n/2$, we have

$$\mathbb{E}^{1/q}\left[\|\Gamma_{m:k}\|^q\right] \leq \sqrt{\kappa_Q}\mathrm{e}\exp\left\{-\frac{a}{2}\sum_{\ell=m}^{k}\alpha_\ell\right\} \ ,$$

and the statement follows. Suppose now that $m < n/2$. In such a case we have, applying (73), that

$$\mathbb{E}^{1/q}\left[\|\Gamma_{m:k}\|^q\right] \leq \sqrt{\kappa_Q}\mathrm{e}\exp\left\{-a\sum_{\ell=m}^{k}\alpha_\ell + (\log n)b_Q^2\sum_{\ell=m}^{k}\alpha_\ell^2\right\}$$

$$\leq \sqrt{\kappa_Q}\mathrm{e}\exp\left\{-a\sum_{\ell=m}^{n}\alpha_\ell + (\log n)b_Q^2\sum_{\ell=m}^{n}\alpha_\ell^2\right\}\exp\left\{-(a/2)\sum_{\ell=n+1}^{k}\alpha_\ell\right\} \ , \tag{74}$$

and we need to bound the first term in the product. We first consider $\alpha_\ell = c_0 \ell^{-1/2}$, and use the inequalities

$$\sum_{\ell=m}^{n}\frac{1}{\ell} \leq \left(1 + \int_m^n \frac{dx}{x}\right) \wedge \left(\int_{m-1}^n \frac{dx}{x}\right) = \left(1 + \log\frac{n}{m}\right) \wedge \left(\log\frac{n}{m-1}\right) \ , \tag{75}$$

and

$$\sum_{\ell=m}^{n} \frac{1}{\sqrt{\ell}} \geq \int_{m}^{n} \frac{dx}{\sqrt{x}} = 2(\sqrt{n} - \sqrt{m}) . \tag{76}$$

Thus, it is enough to satisfy the constraint

$$(\log n) b_Q^2 c_0^2 (1 + \log n - \log m) \leq a c_0 (\sqrt{n} - \sqrt{m}) .$$

Since $m < n/2$, it is enough to ensure that

$$(1 + \log n)(\log n) b_Q^2 c_0^2 \leq a c_0 (\sqrt{n} - \sqrt{n/2}) ,$$

or, equivalently,

$$\frac{\sqrt{n}}{(1 + \log n) \log n} \geq \frac{c_0 b_Q^2}{a(1 - 1/\sqrt{2})} ,$$

which is granted by A3. Combining the above bounds in (74), we obtain that the lemma's statement (71) holds for the step size $\alpha_\ell = c_0/\ell^{1/2}$. Similarly, for $\alpha_\ell = c_0/\ell^\gamma$ with $\gamma \in (1/2; 1)$, we get for $m \geq n/2$ that

$$\mathbb{E}^{1/q}\left[\|\Gamma_{m:k}\|^q\right] \leq \sqrt{\kappa_Q} e \exp\left\{-\frac{a}{2} \sum_{\ell=m}^{k} \alpha_\ell\right\} ,$$

since the relation (73) holds. Similarly, for $m < n/2$, the desired upper bound would follow from the inequality

$$\sum_{\ell=m}^{n} \frac{1}{\ell^{2\gamma}} \leq \int_{m-1}^{n} \frac{dx}{x^{2\gamma}} = \frac{(m-1)^{1-2\gamma} - n^{1-2\gamma}}{2\gamma - 1} \leq \frac{1}{2\gamma - 1} ,$$

together with an inequality

$$\frac{(\log n) b_Q^2 c_0^2}{2\gamma - 1} \leq (a/2) c_0 (n^{1-\gamma} - (n/2)^{1-\gamma}) .$$

The latter inequality can be re-written as

$$\frac{n^{1-\gamma}}{\log n} \geq \frac{2c_0 b_Q^2}{a(2\gamma - 1)(1 - (1/2)^{1-\gamma})} ,$$

which is also granted by A3. Combining the above inequalities implies that (71) holds for $\alpha_\ell = c_0/\ell^\gamma$. The bound (72) can be immediately obtained from (71) using the fact that $e^{-x} \leq 1 - x/2$ for $x \in [0; 1]$. $\qquad\square$

**Corollary 5.** *Under conditions of Corollary 4 it holds with $\mathbb{P}$ – probability at least $1 - 1/n^2$ that*

$$\|\Gamma_{m:k}\| \leq \sqrt{\kappa_Q} e^2 \exp\left\{-(a/2) \sum_{\ell=m}^{k} \alpha_\ell\right\} ,$$

*and*

$$\|\Gamma_{m:k}\| \leq \sqrt{\kappa_Q} e^2 \prod_{\ell=m}^{k} \left(1 - \frac{a\alpha_\ell}{4}\right)$$

*Proof.* It is sufficient to choose $q = 2 \log n$ and use Markov's inequality together with the union bound. $\qquad\square$

**Proposition 7.** *Assume A1, A2, A3 with $\gamma = 1/2$, and A4. Then on the set $\Omega_5$ defined in (47), it holds for any $n \leq m \leq k \leq 2n$, that*

$$\left\{\mathbb{E}^{b}[\|\Gamma_{m+1:k}^{b}\|^2]\right\}^{1/2} \leq \kappa_Q^{3/2} e^{9/8} \exp\left\{-\frac{a}{4} \sum_{\ell=m+1}^{k} \alpha_\ell\right\} .$$

*Proof.* Our proof relies on the auxiliary result of Lemma 13 below together with the blocking technique. Indeed, let us represent

$$k - m = Nh + r \,,$$

where $r < h$ and $h = h(n)$ is a block size defined in (18). Then we obtain, using the independence of bootstrap weights $w_{m+1}, \ldots, w_k$, that

$$\{\mathbb{E}^{\mathsf{b}}[\|\Gamma^{\mathsf{b}}_{m+1:k}\|^2]\}^{1/2} \leq \sqrt{\kappa_Q}\{\mathbb{E}^{\mathsf{b}}[\|\Gamma^{\mathsf{b}}_{m+1:k}\|^2_Q]\}^{1/2}$$

$$= \sqrt{\kappa_Q} \prod_{j=1}^{N}\{\mathbb{E}^{\mathsf{b}}[\|\Gamma^{\mathsf{b}}_{m+1+(j-1)h:m+jh}\|^2_Q]\}^{1/2}\{\mathbb{E}^{\mathsf{b}}[\|\Gamma^{\mathsf{b}}_{m+1+Nh:k}\|^2_Q]\}^{1/2}$$

$$\leq \sqrt{\kappa_Q} \exp\left\{-\frac{a}{4}\sum_{\ell=m+1}^{k}\alpha_\ell\right\}\{\mathbb{E}^{\mathsf{b}}[\|\Gamma^{\mathsf{b}}_{m+1+Nh:k}\|^2_Q]\}^{1/2}\exp\left\{\frac{a}{4}\sum_{\ell=m+1+Nh:k}^{k}\alpha_\ell\right\}\,.$$

In the last inequality we applied Lemma 13 to each of the blocks of length $h$ in the first bound. It remains to upper bound the residual terms. Since the remainder block has length less then $h$, we have due to (81) (which holds according to A4), that

$$\exp\left\{\frac{a}{4}\sum_{\ell=m+1+Nh:k}^{k}\alpha_\ell\right\} \leq \exp\left\{\frac{\alpha_\infty a}{4}\right\} \leq \mathrm{e}^{1/8}\,,$$

where the last inequality is due to Proposition 1. Next,

$$\{\mathbb{E}^{\mathsf{b}}[\|\Gamma^{\mathsf{b}}_{m+1+Nh:k}\|^2_Q]\}^{1/2} \leq \kappa_Q \prod_{\ell=m+1+Nh:k}^{k}\{\mathbb{E}^{\mathsf{b}}[\|(I - \alpha_\ell w_\ell \mathbf{A}_\ell)\|^2]\}^{1/2}$$

$$\leq \kappa_Q \prod_{\ell=m+1+Nh:k}^{k}\{\mathbb{E}^{\mathsf{b}}[(1 + \alpha_\ell|w_\ell|\,\mathrm{C}_{\mathbf{A}})^2]\}^{1/2}$$

$$\leq \kappa_Q \prod_{\ell=m+1+Nh:k}^{k}\{\mathbb{E}^{\mathsf{b}}[1 + 2\alpha_\ell|w_\ell|\,\mathrm{C}_{\mathbf{A}} + \alpha_\ell^2 w_\ell^2\,\mathrm{C}_{\mathbf{A}}^2]\}^{1/2}\,.$$

Since

$$\mathbb{E}[|w_\ell|] \leq \sqrt{\mathbb{E}[w_{\ell^2}]} \leq \sqrt{(\mathbb{E}[w_\ell])^2 + \operatorname{Var} w_\ell} = \sqrt{2}\,,$$

we get from previous bound

$$\{\mathbb{E}^{\mathsf{b}}[\|\Gamma^{\mathsf{b}}_{m+1+Nh:k}\|^2_Q]\}^{1/2} \leq \kappa_Q \prod_{\ell=m+1+Nh:k}^{k}(1 + 2\sqrt{2}\alpha_\ell\,\mathrm{C}_{\mathbf{A}} + 2\alpha_\ell^2\,\mathrm{C}_{\mathbf{A}}^2)^{1/2}$$

$$\leq \kappa_Q \exp\left\{\sqrt{2}\,\mathrm{C}_{\mathbf{A}}\sum_{\ell=m+1+Nh:k}^{k}\alpha_\ell\right\} \leq \kappa_Q \mathrm{e}^{\sqrt{2}\,\mathrm{C}_{\mathbf{A}}\,c_0 h/\sqrt{n}} \leq \kappa_Q \mathrm{e}\,,$$

where in the last line we additionally used (19). $\qquad\square$

**Lemma 13.** *Assume A1, A2, A3 with $\gamma = 1/2$, and A4. On the set $\Omega_5$ defined in (47), it holds for $h = h(n)$ defined in (18) and any $m \in [n; 2n - h]$, that*

$$\{\mathbb{E}^{\mathsf{b}}[\|\Gamma^{\mathsf{b}}_{m+1:m+h}\|^2_Q]\}^{1/2} \leq \exp\left\{-\frac{a}{4}\sum_{\ell=m+1}^{m+h}\alpha_\ell\right\}\,.$$

*Proof.* Recall that we use the notation $\mathbb{E}^{\mathsf{b}}[\cdot] = \mathbb{E}[\cdot|\mathcal{Z}^{2n}]$, where $\mathcal{Z}^{2n} = (Z_1, \ldots, Z_{2n})$ are the random variables used in the construction of the iterates $\{\theta_k\}_{1 \leq k \leq n}$ in (1).

Let $h \in \mathbb{N}$ be a block length, which value will be determined later, and consider a product

$$\Gamma^{\mathsf{b}}_{m+1:m+h} = \prod_{\ell=m+1}^{m+h}(I - \alpha_\ell w_\ell \mathbf{A}_\ell)\,. \tag{77}$$

Expanding the product of matrices (77), we obtain

$$\Gamma^{\mathrm{b}}_{m:m+h} = \mathrm{I} - \sum_{\ell=m+1}^{m+h} \alpha_\ell \mathbf{A}_\ell - \mathbf{S} + \mathbf{R} = \mathrm{I} - \sum_{\ell=m+1}^{m+h} \alpha_\ell \bar{\mathbf{A}} - \sum_{\ell=m+1}^{m+h} \alpha_\ell (\mathbf{A}_\ell - \bar{\mathbf{A}}) - \mathbf{S} + \mathbf{R} \;, \quad (78)$$

where $\mathbf{S} = \sum_{\ell=m+1}^{m+h} \alpha_\ell (w_\ell - 1) \mathbf{A}_\ell$ is a linear statistics in $\{w_\ell\}_{\ell=m+1}^{m+h}$, and the remainder $\mathbf{R}$ collects the higher-order terms in the products

$$\mathbf{R} = \sum_{r=2}^{h} (-1)^r \sum_{(i_1,\ldots,i_r) \in \mathsf{I}_r^\ell} \prod_{u=1}^{r} \alpha_{i_u} w_{i_u} \mathbf{A}_{i_u} \;.$$

with $\mathsf{I}_r^\ell = \{(i_1,\ldots,i_r) \in \{m+1,\ldots,m+h\}^r : i_1 < \cdots < i_r\}$. We first consider the contracting part in matrix $Q$-norm. Indeed, applying (8), we obtain that

$$\|\mathrm{I} - \sum_{\ell=m+1}^{m+h} \alpha_\ell \bar{\mathbf{A}}\|_Q^2 \le 1 - a \sum_{\ell=m+1}^{m+h} \alpha_\ell \;,$$

provided that $h$ is set in such a manner that $\sum_{\ell=m+1}^{m+h} \alpha_\ell \le \alpha_\infty$, where $\alpha_\infty$ is defined in (7). Hence, we get from the above inequality that for any $u \in \mathbb{R}^d$, it holds that

$$\|\mathrm{I} - \sum_{\ell=m+1}^{m+h} \alpha_\ell \bar{\mathbf{A}}\|_Q \le 1 - (a/2) \sum_{\ell=m+1}^{m+h} \alpha_\ell \;.$$

Now we need to estimate the remainders in the representation (78). On the set $\Omega_5$, it holds that

$$\| \sum_{\ell=m+1}^{m+h} \alpha_\ell (\mathbf{A}_\ell - \bar{\mathbf{A}})\|_Q \le 2\,\mathrm{C}_{\mathbf{A}}\,\sqrt{\kappa_Q} \sqrt{\sum_{\ell=m+1}^{m+h} \alpha_\ell^2 \log(2n^4)} \;.$$

Moreover, it is straightforward to check that

$$\mathbb{E}^{\mathrm{b}}[\|\mathbf{S}\|_Q^2] \le \mathrm{C}_{\mathbf{A}}^2\,\kappa_Q \sum_{\ell=m+1}^{m+h} \alpha_\ell^2 \;.$$

In order to bound the remainder term $\mathbf{R}$, we note that

$$\mathbb{E}^{\mathrm{b}}[\|\mathbf{R}\|_Q] \le \sum_{r=2}^{h} \binom{h}{r} \alpha_{m+1}^r (2\,\mathrm{C}_{\mathbf{A}})^r \kappa_Q^{r/2} \le \alpha_{m+1}^2 (2\,\mathrm{C}_{\mathbf{A}})^2 \kappa_Q \sum_{r=0}^{h-2} \binom{h}{r+2} \alpha_{m+1}^r (2\,\mathrm{C}_{\mathbf{A}})^r \kappa_Q^{r/2}$$

$$\le \frac{\alpha_{m+1}^2 h^2 (2\,\mathrm{C}_{\mathbf{A}})^2 \kappa_Q}{2} \exp\{2\alpha_{m+1}\,\mathrm{C}_{\mathbf{A}}\,\kappa_Q^{1/2}\}$$

$$\le \frac{\alpha_{m+1}^2 h^2 (2\,\mathrm{C}_{\mathbf{A}})^2 \kappa_Q \mathrm{e}}{2} \;.$$

To complete the proof it remains to set the parameter $h$ in such a way that we can guarantee

$$\mathrm{C}_{\mathbf{A}}\,\sqrt{\kappa_Q} \sqrt{\sum_{\ell=m+1}^{m+h} \alpha_\ell^2 \left(1 + 2\log(2n^4)\right)} + \frac{\alpha_{m+1}^2 h^2\,\mathrm{C}_{\mathbf{A}}^2\,\kappa_Q \mathrm{e}}{2} \le \frac{a}{4} \sum_{\ell=m+1}^{m+h} \alpha_\ell \;, \quad (79)$$

keeping at the same time the constraint

$$\sum_{\ell=m+1}^{m+h} \alpha_\ell \le \alpha_\infty \;. \quad (80)$$

Recall that $\alpha_\ell = c_0/\sqrt{\ell}$. Thus, using the bounds (75) and (76), we obtain that

$$\frac{a}{4} \sum_{\ell=m+1}^{m+h} \alpha_\ell \ge \frac{ac_0}{2}(\sqrt{m+h} - \sqrt{m+1}) \ge \frac{ac_0}{2}(\sqrt{m+h} - \sqrt{m}) \;, \quad (81)$$

and

$$\sum_{\ell=m+1}^{m+h} \alpha_\ell^2 = \sum_{\ell=m+1}^{m+h} \frac{c_0^2}{\ell} \le c_0^2 (\log(m+h) - \log m) . \tag{82}$$

Hence, taking into account (81) and (82), and $\frac{1}{m+1} \le \frac{1}{m}$, the inequality (79) would follow from the bound

$$C_{\mathbf{A}} \sqrt{\kappa_Q} \sqrt{\log(m+h) - \log(m)} \left(1 + 2\log(2n^4)\right) + \frac{c_0 h^2 C_{\mathbf{A}}^2 \kappa_Q \mathrm{e}}{2m} \le \frac{a}{2}(\sqrt{m+h} - \sqrt{m}) . \tag{83}$$

Since $\log(1+x) \le x$ for $x \ge 0$ and $c_0 C_{\mathbf{A}}^2 \kappa_Q \mathrm{e} \le 1$, the latter inequality is satisfied if

$$C_{\mathbf{A}} \sqrt{\kappa_Q} \frac{\sqrt{h}}{\sqrt{m}} \left(1 + 2\log(2n^4)\right) + \frac{h^2}{2m} \le \frac{a}{2}(\sqrt{m+h} - \sqrt{m}) .$$

Now we use one more lower bound

$$\sqrt{m+h} - \sqrt{m} = \sqrt{m}(\sqrt{1+h/m} - 1) \ge \frac{\sqrt{m}(\sqrt{2}-1)h}{m} = \frac{(\sqrt{2}-1)h}{\sqrt{m}} ,$$

which follows from an elementary inequality $\sqrt{1+x} \ge 1 + (\sqrt{2}-1)x$, valid for $0 \le x \le 1$. Hence, (83) would from the inequality

$$C_{\mathbf{A}} \sqrt{\kappa_Q} \frac{\sqrt{h}}{\sqrt{m}} \left(1 + 2\log(2n^4)\right) + \frac{h^2}{2m} \le \frac{a(\sqrt{2}-1)h}{2\sqrt{m}} . \tag{84}$$

Setting $h$ is such a manner that

$$\frac{h}{\sqrt{m}} \le \frac{a(\sqrt{2}-1)}{2} ,$$

inequality (84) would follow from

$$C_{\mathbf{A}} \sqrt{\kappa_Q} \frac{\sqrt{h}}{\sqrt{m}} \left(1 + 2\log(2n^4)\right) \le \frac{a(\sqrt{2}-1)h}{4\sqrt{m}} .$$

The latter inequality is satisfied, if the block size $h$ satisfies

$$h \ge \left(\frac{4 C_{\mathbf{A}} \kappa_Q^{1/2}}{(\sqrt{2}-1)a}\right)^2 (1 + 2\log(2n^4))^2 .$$

Thus, setting $h(n)$ as in (18), all previous inequalities will be fulfilled, provided that

$$\begin{cases} \frac{h(n)}{\sqrt{n}} & \le \frac{a(\sqrt{2}-1)}{2} \\ \frac{c_0 h(n)}{\sqrt{n}} & \le \alpha_\infty . \end{cases}$$

Here last inequality follows from (80) and the following simple bounds, where we use that $m \ge n$ and $\sqrt{1+x} \le 1 + x/2$:

$$\sum_{\ell=m+1}^{m+h} \alpha_\ell \le \sum_{\ell=n+1}^{n+h} \alpha_\ell = c_0 \sum_{\ell=n+1}^{n+h} \frac{1}{\sqrt{\ell}} \le c_0 \int_n^{n+h} \frac{dx}{\sqrt{x}} = 2c_0(\sqrt{n+h} - \sqrt{n}) \le \frac{c_0 h}{\sqrt{n}} .$$

Now (78) implies that

$$\left\{\mathbb{E}^{\mathsf{b}}[\|\Gamma_{m+1:m+h}^{\mathsf{b}}\|_Q^2]\right\}^{1/2} \le 1 - (a/4) \sum_{\ell=m+1}^{m+h} \alpha_\ell ,$$

and the statement follows from an elementary inequality $1 + x \le \mathrm{e}^x$. $\qquad\square$

# E Applications to the TD learning

Recall that the temporal difference learning algorithm in the LSA's setting can be written as

$$\theta_k = \theta_{k-1} - \alpha_k(\mathbf{A}_k\theta_{k-1} - \mathbf{b}_k) , \tag{85}$$

where $\mathbf{A}_k$ and $\mathbf{b}_k$ are given by

$$\begin{aligned}
\mathbf{A}_k &= \varphi(s_k)\{\varphi(s_k) - \gamma\varphi(s_k')\}^\top , \\
\mathbf{b}_k &= \varphi(s_k)r(s_k, a_k) .
\end{aligned} \tag{86}$$

Recall that our aim is to estimate the agent's *value function*

$$V^\pi(s) = \mathbb{E}[\textstyle\sum_{k=0}^\infty \gamma^k r(s_k, a_k)|s_0 = s] ,$$

where $a_k \sim \pi(\cdot|s_k)$, and $s_{k+1} \sim P(\cdot|s_k, a_k)$, for any $k \in \mathbb{N}$. We define the transition kernel under policy $\pi$

$$P_\pi(B|s) = \int_{\mathcal{A}} P(B|s, a)\pi(\mathrm{d}a|s) , \tag{87}$$

which corresponds to the 1-step transition probability from state $s$ to a set $B \in \mathcal{B}(\mathcal{S})$. We denote by $\mu$ the invariant distribution over the state space $\mathcal{S}$ induced by the transition kernel $P_\pi(\cdot|s)$ in (87). In this case the TD learning updates (85) correspond to the approximate solution of the deterministic system $\bar{\mathbf{A}}\theta^\star = \bar{\mathbf{b}}$, where we have set, respectively,

$$\begin{aligned}
\bar{\mathbf{A}} &= \mathbb{E}_{s\sim\mu, s'\sim P_\pi(\cdot|s)}[\varphi(s)\{\varphi(s) - \gamma\varphi(s')\}^\top] \\
\bar{\mathbf{b}} &= \mathbb{E}_{s\sim\mu, a\sim\pi(\cdot|s)}[\varphi(s)r(s, a)] .
\end{aligned} \tag{88}$$

## E.1 Proof of Proposition 2

We first need to check that the matrix $\bar{\mathbf{A}} + \bar{\mathbf{A}}^\top$, where $\bar{\mathbf{A}}$ is defined in (88), is positive-definite. In order to show this fact we closely follow the exposition of [61, Lemma 18] and [51, Lemma 5]. Define a random matrix $\mathbf{A}$ as an independent copy of $\mathbf{A}_k$ from (86), that is,

$$\mathbf{A} = \varphi(s)\{\varphi(s) - \gamma\varphi(s')\}^\top ,$$

where $s \sim \mu$, and $s' \sim P_\pi(\cdot|s)$. With the definition of $\mathbf{A}$, we get that

$$\begin{aligned}
\mathbf{A} + \mathbf{A}^\top &= \varphi(s)\{\varphi(s) - \gamma\varphi(s')\}^\top + \{\varphi(s) - \gamma\varphi(s')\}\varphi(s)^\top \\
&= 2\varphi(s)\varphi(s)^\top - \gamma\{\varphi(s)\varphi(s')^\top + \varphi(s')\varphi(s)^\top\} \\
&\succeq (2 - \gamma)\varphi(s)\varphi(s)^\top - \gamma\varphi(s')\varphi(s')^\top ,
\end{aligned}$$

where we used an elementary inequality $uv^\top + vu^\top \preceq (uu^\top + vv^\top)$ valid for any $u, v \in \mathbb{R}^d$. Hence, with the definition of $\Sigma_\varphi$ in (22), we get

$$\bar{\mathbf{A}} + \bar{\mathbf{A}}^\top = \mathbb{E}[\mathbf{A} + \mathbf{A}^\top] \succeq 2(1 - \gamma)\Sigma_\varphi . \tag{89}$$

Hence, $\bar{\mathbf{A}} + \bar{\mathbf{A}}^\top$ is positive-definite, and we can set $P = \bar{\mathbf{A}} + \bar{\mathbf{A}}^\top$ in the right-hand side of the Lyapunov equation (69). Obviously, $Q = I$ is a solution to the corresponding Lyapunov equation

$$\bar{\mathbf{A}}^\top Q + Q\bar{\mathbf{A}} = \bar{\mathbf{A}} + \bar{\mathbf{A}}^\top .$$

Moreover, applying [61, Lemma 18], we obtain

$$\bar{\mathbf{A}}^\top\bar{\mathbf{A}} \preceq \mathbb{E}[\mathbf{A}^\top\mathbf{A}] \preceq (1 + \gamma)^2\Sigma_\varphi . \tag{90}$$

Hence, we get for $\alpha \leq (1 - \gamma)/(1 + \gamma)^2$, and applying (89) and (90), that

$$\begin{aligned}
(I - \alpha\bar{\mathbf{A}})^\top(I - \alpha\bar{\mathbf{A}}) &= I - \alpha(\bar{\mathbf{A}}^\top + \bar{\mathbf{A}}) + \alpha^2\bar{\mathbf{A}}^\top\bar{\mathbf{A}} \\
&\preceq I - 2\alpha(1 - \gamma)\Sigma_\varphi + \alpha^2(1 + \gamma)^2\Sigma_\varphi \\
&\preceq I - \alpha(1 - \gamma)\Sigma_\varphi \\
&\preceq (1 - \alpha(1 - \gamma)\lambda_{\min}(\Sigma_\varphi))I .
\end{aligned}$$

Hence, the bound (8) holds with $a = (1 - \gamma)\lambda_{\min}(\Sigma_\varphi)$ and $\alpha_\infty = (1 - \gamma)/(1 + \gamma)^2$.

# F  Experimental details for the TD learning

Here we provide some details on numerical experiments. Code to run experiments is provided in https://github.com/svsamsonov/BootstrapLSA. For the considered Garnet problem we choose the policy $\pi$ in the following way. For any $a \in \mathcal{A}$, we set

$$\pi(a|s) = \frac{U_a^{(s)}}{\sum_{i=1}^{|\mathcal{A}|} U_i^{(s)}} \ ,$$

where the $U_i^{(s)}$ are independent random variables following uniform distribution $\mathcal{U}[0,1]$. Here we assume that each action $a \in \mathcal{A}$ can be selected at any state $s \in \{1, \ldots, N_s\}$. We generate an instance of Garnet problem with mentioned parameters, and find analytically the true parameter $\theta^\star$. In order to estimate the supremum

$$\Delta_n := \sup_{x \in \mathbb{R}} \left| \mathbb{P}(\sqrt{n}\|\bar{\theta}_n - \theta^\star\| \leq x) - \mathbb{P}(\|\Sigma_\infty^{1/2}\eta\| \leq x) \right| \ ,$$

$\eta \sim \mathcal{N}(0, \mathrm{I}_{N_s})$, and show that this supremum scales as $n^{-1/4}$ when $\gamma = 1/2$ and admits slower decay for other powers of $\gamma$. We first approximate true probability $\mathbb{P}(\|\Sigma_\infty^{1/2}\eta\| \leq x)$ by the corresponding empirical probabilities based on sample of size $M \gg n$. We fix $M = 5 \cdot 10^7$. We choose trajectory lengths

$$n \in \{1600, 3200, 6400, 12800, 25600, 51200, 102400, 204800, 409600, 819200, 1638400\} \ ,$$

fix the length of burn-in period $n_0 = 102400$, and generate $N = 6553600$ independent trajectories starting in the fixed point $\theta_0 \in \mathbb{R}^{N_s}$. We set the learning rate schedule as $\alpha_k = c_0/k^\gamma$ and try different values $\gamma \in \{0.5, 0.65, 0.7\}$, and $c_0 = 4.0$. Unfortunately, even the chosen order of trajectory length $n$ seems to be insufficient in order to significantly distinguish, for example, between $\gamma = 0.5$ and $\gamma = 0.65$. However, learning rate schedule with faster decay performs worse in terms of $\Delta_n$. Note that the current experiment is already rather computationally intense for artificial problem and takes about 12 hours of compute on a Core i9 - 10920x processor with 12 cores with 3.7 GHz.

