# OpenReview forum: "Gaussian Approximation and Multiplier Bootstrap for Polyak-Ruppert Averaged Linear Stochastic Approximation with Applications to TD Learning"
_NeurIPS.cc/2024/Conference — NeurIPS 2024 poster_

### Official Review · Reviewer_uNZU · 2024-07-02

**Soundness:** 3
**Presentation:** 3
**Contribution:** 2
**Rating:** 5
**Confidence:** 4

**Summary:**

The paper presents advancements in the theoretical understanding of the linear stochastic approximation (LSA) algorithm.

It establishes the Berry–Esseen bound for the normal approximation of Polyak-Ruppert averaged iterates, achieving an optimal rate with an aggressive step size of $\alpha_k \approx k^{-1/2}$. Additionally, it demonstrates the non-asymptotic validity of confidence intervals using a novel multiplier bootstrap procedure, marking a first in this domain.

The practical utility of these theoretical results is showcased through applications in temporal difference (TD) learning for reinforcement learning.

**Strengths:**

1. **Theoretical Advancements**: The paper makes theoretical contributions by establishing the Berry–Esseen bound for the normal approximation of Polyak-Ruppert averaged iterates. Though some previous works have done this in special cases before, this work differs from them in providing a tighter bound.

2. **Good Clarity:** The paper is easy to follow and well-written. The proof seems correct (not checked very carefully).

**Weaknesses:**

1. **Strong Assumptions**: The requirement that $\epsilon(z)$ is uniformly bounded is a strong assumption that might limit the applicability of the results. Relaxing this condition to weaker ones, such as finite moments, could enhance the paper's generality.

2. **Missing References**: There are some missing references, see the limitations.

3. **Discussion on Lower Bounds**: The paper focuses on deriving upper bounds but lacks a discussion on the tightness and potential lower bounds. Addressing this aspect could provide a more comprehensive understanding of the bounds' efficacy and limitations.

**Questions:**

See the Limitations.

**Limitations:**

1. **Missing References on Statistical Inference**: Statistical inference for nonlinear stochastic approximation has been considered by [1*] and [2*], which is highly related to this manuscript but hasn’t been cited. Note that [1*] provides a Berry-Esseen-like bound for the whole trajectory rather than the averaged iterates. These references could be cited after [36] in line 116.
     - [1*] Li, Xiang, Jiadong Liang, and Zhihua Zhang. "Online statistical inference for nonlinear stochastic approximation with Markovian data." arXiv preprint arXiv:2302.07690 (2023).
     - [2*] Li, Xiang, et al. "A statistical analysis of Polyak-Ruppert averaged Q-learning." International Conference on Artificial Intelligence and Statistics. PMLR, 2023.

2. **Assumption on Bounded $\epsilon(z)$**: Theorem 1 requires that $\epsilon(z)$ is uniformly bounded, which is a strong condition. Is it possible to relax this condition to a weaker one, such as $\epsilon(z)$ only having a finite order of moments (such as the fourth order moment or smaller)?

3. **Tightness of Derived Upper Bounds**: The paper provides upper bounds, but it would be insightful to discuss the tightness of these bounds. Are there any thoughts or conjectures regarding potential lower bounds?

#### Minor Corrections:

1. **Figure 1**: The last subfigure should be labeled as (c).

---

> ### Author Rebuttal · Authors · 2024-08-07
>
> We would like to thank the referee uNZU for careful reading of the manuscript and raising interesting questions. Next, we answer the issues raised.
>
> **Missing References on Statistical Inference**
> We thank the referee for provided references and will add them to the revised version of the paper.
>
> **Strong Assumptions: assumption on Bounded $\varepsilon(z)$**
> Indeed, the assumption of bounded $\varepsilon(z)$ is strong, but can be partially relaxed. Following the stablity of matrix products technique, used in [Proposition 3][Durmus et al, 2021], we can generalize the moment bound for products of random matrices (Corollary 4) for the setting when the random variable $\|\|A(Z) - \bar{A}\|\|$ has only finite number of moments.
> In particular, finite $3$rd moment of $\|\|A(Z) - \bar{A}\|\|$ (which implies naturally that $\|\|\varepsilon(z)\|\|$ also admits only a finite $3$rd moment) is sufficient to obtain the first main result of the paper on the Berry-Esseen inequality (Theorem $2$). However, it will be not sufficient to prove the boostrap validity (Theorem $3$), since this result requires high-probability bounds on the product of random matrices $\Gamma_{m:k}$. Yet there are two settings when we can generalize our results without the assumption concerning bounded $\|\|\varepsilon(z)\|\|$:
>
> 1. Random matrix $A(Z)$ is almost sure bounded, but $\|\|\varepsilon(z)\|\|$ is sub-gaussian, or, more generally, for any $p \geq 2$ it holds that
> $$
> \mathsf{E}^{1/p}[\|\| \varepsilon(z)\|\|^p] \leq C_{\varepsilon} p^{\beta}\,,
> $$
> for some $\beta \geq 1/2$. In such a case, we can generalize our bootstrap validity along the lines of the current proof;
>
> 2. The random variable $\|\|A(Z) - \bar{A}\|\|$ is sub-Gaussian, and $\|\|\varepsilon(z)\|\|$ is sub-Gaussian. In such a case, using the high-probability bounds outlined in [Proposition 3][Durmus et al, 2021], we can have a counterpart of Theorem $3$ up to additional powers of $\log{n}$.
> \end{enumerate}
> We will add the discussion above the the revised version of the paper.
>
> **Tightness of Derived Upper Bounds**
> Indeed, it is a very interesting question to obtain the mathching lower bounds that can illustrate the fact that the rate $n^{-1/4}$ is indeed sharp. See the reply for all referees for details on lower bounds.
>
>
> **Minor Corrections**
> Thanks, we will correct this typo and will perform additional proofreading for the revised version of the paper.

---

> > ### Author Response · Authors · 2024-08-14
> >
> > Dear referee,
> >
> > Please kindly let us know if you have any follow-up questions that need further clarification. Your insights are valuable to us, and we stand ready to provide any additional information that might be helpful.

---

### Official Review · Reviewer_ew5L · 2024-07-12

**Soundness:** 2
**Presentation:** 2
**Contribution:** 3
**Rating:** 7
**Confidence:** 3

**Summary:**

The present paper studies linear stochastic approximation with martingale difference noise and diminishing step-sizes. The authors obtain Berry-Esseen bounds for the parameter sequence with Polyak-Ruppert averaging as well as a generalization of finite-time bounds for estimation confidence intervals for parameters in LSA. The obtained Berry-Esseen bounds are illustrated by a TD learning numerical example.

**Strengths:**

To the best of the reviewer's knowledge, both of the contributions are novel. In particular, I believe that this is the first Berry-Esseen bound type bound to be obtained for general linear SA, which is is exciting to see.

The assumptions, contributions and approach to analysis are objectively identified. The authors also did a great job in providing discussions/remarks/intuition for their results.

The paper is well-written but some proofreading is recommended.

**Weaknesses:**

Although the authors did a good job in outlining the scope and contributions of the paper, the analysis and main text are hard to follow given the number of symbols and equations. It is easy for a reader to get lost/distracted midway and it is very hard to keep track of the definitions of each of the terms

I understand that this is an issue with theory papers like this, but would encourage the authors to move unnecessary terms or inequalities to the appendix (e.g. (9) in A3. The exact lower bound adds very little in the main text in my opinion. Its definition could have been postponed to the Appendix.)

The numerical experiments are also weak since many of the plots are not related to the contributions of the paper itself. Plot c seems to be the only one directly related to the theorems of the paper, but it still does not illustrate the theory that well. Maybe including a plot of C k^{-1/4}  to plot (b) where C is a constant for comparison could help in inferring convergence rates.

Also, there is a mistake in the label of Figure 1. Subfigure (b) is mentioned twice.

**Questions:**

- Could the authors clarify if using Polyak Ruppert averaging is necessary to obtain such a Berry-Esseen bound? It would be exciting to see bounds for unaveraged estimates as well.

-Could the authors  run  the experiment  supporting figure (c) for longer to see if the curve with \gamma = 1/2 willnot continue to go upwards eventually?

**Limitations:**

The authors clearly identified the limitations of their results through a clear list of assumptions

---

> ### Author Rebuttal · Authors · 2024-08-07
>
> We would like to thank the referee ew5L for careful reading of the manuscript and valuable suggestions for presentation improvement. Next, we answer the issues raised.
>
> **The analysis and main text are hard to follow**
> Indeed, there are technical details in the main text that complicates reading, yet we were willing to be precise and state exact lower bounds on sample size $n$, especially in $A3$ and $A4$. In the revised version we will modify eq. $9$ and eq. $17$, switching to $\mathcal{O}$ - notation, and move precise bounds to appendix, as was suggested.
>
> **The numerical experiments are also weak since many of the plots are not related to the contributions of the paper itself**
> We respectfully disagree with the referee - the figure (a) explains, why step sizes $\alpha _k = c_0 / k^{\gamma}$, $\gamma > 1/2$ are less preferrable. In this case the plot (a) illustrates slow convergence of the rescaled error $\sqrt{n}(\bar{\theta}_n - \theta^*)$, which is the reason which explains slow convergence of the respective rescaled approximation errors on subfigures (b) and (c). For the new plot (see the attached PDF file veasible for all referees) we included the expression $n^{1/4} \Delta_n$ with and without logarithmic scaling. Thus we expect that this quantity converges to a constant when $\gamma = 1/2$ and grows with $n$ when $\gamma > 1/2$. We will also consider running longer experiments, as for now we have included a figure with one more observation (added point corresponding to $n = 3 276 800$ observations).
>
> **Could the authors clarify if using Polyak Ruppert averaging is necessary to obtain such a Berry-Esseen bound?**
> No, in principle it is not necessary, but the result will be slightly different in this case. It is known that (see e.g. [Fort, 2015]), that the corresponding CLT for the last iterate can be written as
> $$
> \frac{\theta_k - \theta^*}{\sqrt{\alpha_k}} \to \mathcal{N}(0,\Sigma_{\text{last}}),
> $$
> where the covariance matrix $\Sigma_{\text{last}}$ is different from $\Sigma_{\infty}$. Then, using the perturbation-expansion technique from [Aguech et al, 2000], we write that
> $$
> \theta_n - \theta^* = \tilde{\theta_n}^{(tr)} + J_{n}^{(0)} + H_{n}^{(0)},
> $$
> where $\tilde{\theta_n}^{(tr)} = \Gamma_{1:n}(\theta_0 - \theta^*), \quad \Gamma_{1:n} =  \prod_{i=1}^{n} (I - \alpha_{i} A(Z_i) ) $ is the transient component of the error,
> $$
> J_{n}^{(0)} = -\sum_{j=1}^{n}\alpha_j (I - \alpha_j \bar{A})^{n-j} \epsilon(Z_j)
> $$
> is the leading (with respect to step size) component of the error and $H_{n}^{(0)}$ is a remainder term. Thus, using the argument from the current submission, $\tilde{\theta_n}^{(tr)}$ is exponentially small in $n$, $J_{n}^{(0)}$ is the linear statistics in $\epsilon(Z_j)$ that guaranttes asymptotic normality after re-normalization, and $H_{n}^{(0)}$ is the remainder term. It can be shown that
> $$
> \mathsf{E}^{1/2}[\|J_{n}^{(0)}\|^{2}] \lesssim \sqrt{\alpha_n}, \quad \mathsf{E}^{1/2}[\|H_{n}^{(0)}\|^{2}] \lesssim \alpha_n.
> $$
> Thus, applying similar technique of randomized concentration inequalities (formula 13 in the current submission), we will obtain the Berry-Esseen bound for $\frac{\theta_n - \theta^*}{\sqrt{\alpha_n}}$, which should scale as $\sqrt{\alpha_n}$.
>
> **Typos in figure labelling**
> Thanks, we will correct this typo and will perform additional proofreading for the revised version of the paper.
>
> **References:**
>
> [Aguech et al, 2000] Rafik Aguech, Eric Moulines, and Pierre Priouret. On a perturbation approach for the analysis of stochastic tracking algorithms. SIAM Journal on Control and Optimization, 39(3):872–899, 2000.
>
> [Fort, 2015] Central limit theorems for stochastic approximation with controlled Markov chain
> 411 dynamics. ESAIM: PS, 19:60–80, 2015.

---

> > ### Comment · Reviewer_ew5L · 2024-08-11
> >
> > I appreciate the author's responses.
> >
> > I believe that it would be beneficial to include some discussion/remarks on the final version about the extension of their results for estimates without PR as in the response provided by the authors.
> >
> > I apologize for not taking enough time to fully grasp the experiments in the paper, but I understand them now. Thank you for providing a longer run.

---

> ### Author Response · Authors · 2024-08-12
>
> We thank the referee for their comments. We will include a discussion on the Berry-Esseen result for last iterate as well as a longer run for simulations.

---

### Official Review · Reviewer_DkFj · 2024-07-25

**Soundness:** 4
**Presentation:** 2
**Contribution:** 3
**Rating:** 6
**Confidence:** 3

**Summary:**

## Overview

Let $Z, Z_1, \dots, Z_n$ be i.i.d. random elements with a common distribution $\pi$ over $\mathbf{Z}$. Given $A : \mathbf{Z} \to \mathbb{R}^{d\times d}$ and $b : \mathbf{Z} \to \mathbb{R}^d$, the goal of the LSA procedure is to find the unique solution $\theta^\star$ of
$$\mathbb{E}\left(A(Z)\theta^\star - b(Z)\right) = 0. $$

Given a decreasing sequence of step sizes $\alpha_k$ and a starting point $\theta_0$, the standard LSA is given by
$$\theta_k = \theta_{k-1} - \alpha_k(A(Z_k)\theta_{k-1} - b(Z_k)) $$
and the Polyak-Ruppert averaged LSA is given by
$$\overline{\theta_n} = \frac{1}{n} \sum_{k=n}^{2n-1} \theta_k. $$

The authors provide Berry-Essen-type bounds for the Gaussian approximation of $\sqrt{n} \left( \overline{\theta_n} - \theta^\star \right)$ and for the corresponding multiplicative bootstrap process. Namely, for the Gaussian approximation result they upper bound the quantity
$$\rho_n = \sup_\text{B convex} \left| \mathbb{P}\left( \sqrt{n} \left( \overline{\theta_n} - \theta^\star \right) \in B \right) - \mathbb{P}\left( \Sigma_\infty^\frac{1}{2} \eta \in B \right)\right| $$
where $\eta \sim \mathcal{N}(0,I_d)$. And for the bootstrap approximation result they upper bound
$$\rho_n^b = \sup_\text{B convex} \left| \mathbb{P}\left( \left.\sqrt{n} \left( \overline{\theta_n^b} - \overline{\theta_n} \right) \in B \right| Z_1, \dots, Z_{2n} \right) - \mathbb{P}\left( \sqrt{n} \left( \overline{\theta_n} - \theta^\star \right) \in B \right)\right| $$
where $\overline{\theta_n^b}$ are obtained from a multiplier bootstrap process. It is interesting to notice that this multiplier process can be evaluated online without keeping a history in memory.

## Overall proof arguments

To provide the Gaussian approximation result the authors write
$$\theta_n - \theta^\star = (I - \alpha_n A(Z_n))(\theta_{n-1} - \theta^\star) - \alpha_n \varepsilon(Z_n) $$
where
$$\varepsilon(z) = A(z)\theta^\star - b(z). $$

The authors observe that $\varepsilon(Z_n)$ can be viewed as a noise, which is assumed to be bounded. Meanwhile, the operator $ I - \alpha_n A(Z_n) $ is a random perturbation around $ I - \alpha_n \mathbb{E}A(Z) $, which is shown to act as a contraction in an appropriate norm, provided that $-\mathbb{E}A(Z)$ is Hurwitz. Thus, when one take the PR average the noise is expected to behave as the sum of i.i.d. noises and the contraction term must shrink. This is formally done writing the PR average in the form of Theorem 2.1 [reference 60 of the paper] and bounding the terms given by the latter theorem.

The proof the bootstrap approximation result follows the standard practice in this literature:
- First, conditionally on the sample, a Gaussian approximation is obtained relating the bootstrap to a Gaussian with its covariance.
- Second, a Gaussian comparison theorem relates the latter Gaussian with the desired Gaussian, with some concentration results being used to bound their difference in high probability.

## Main claims and observations

The authors show that taking a step size of order $\frac{1}{\sqrt{k}}$ yields the best possible convergence rate in their bounds. They provide empirical evidence that this rate is optimal.
They also claim to be the first ones to fully provide a non-asymptotic bootstrap approximation result.

**Strengths:**

The paper seems to be the first to provide non-asymptotic Gaussian and bootstrap approximation bounds for LSA. Their assumptions are quite mild and are in line with the ones made in similar papers in other domains. For instance, assumption A.2 is similar in nature to the boundness assumptions and the strong-covariance assumption made in [A]. The paper is mathematically sound and poses interesting research directions. I'm particularly curious about the convergence rate of $n^{-\frac{1}{4}}$ suggested by their theoretical results and supported by their experiment. The proposed application to policy evaluation in RL is also interesting. Finally, I point out that the code on the supplementary material was easy to reproduce.

[A] Chernozhukov, Victor, Denis Chetverikov, and Yuta Koike. "Nearly optimal central limit theorem and bootstrap approximations in high dimensions." The Annals of Applied Probability 33.3 (2023): 2374-2425.

**Weaknesses:**

The main claim of the paper is that their bounds suggest an optimal convergence rate of $n^{-\frac{1}{4}}$ when taking $\alpha_k = \frac{c}{\sqrt{k}}$. The bottleneck of this convergence rate comes from an application of Cauchy-Schwarz inequality, the boundness of $\varepsilon$ and the MSE bound on $D$ given by Theorem 1. Meanwhile, the authors also provide experimental evidence that this convergence rate is indeed optimal (although not considering gamma values below 0.5 in Figures 1 or 3). A counter-example or a deeper discussion on this convergence rate would be beneficial to the paper: is it an artifact of the proof or is there reason to believe it cannot be improved?

**Questions:**

Some observations:
- Line 54 has a typo in "Berry-Essee".
- In Equation (15) the $\ell$ can be removed from $\theta_{k}^{b,\ell}$ to enhance clarity. It is only used to explain that to evaluate the probability in practice one must run several samples of the bootstrap process.
- Line 263 has a typo in "date".
- The Equation after line 303 is using $\phi$ instead of $\varphi$.
- Line 307 asks for a sequence of "TD(0) updates", what does the "(0)" stands for?
- Figure 1 lacks x and y labels. The legend can also be improved for clarity.

**Limitations:**

The authors adequately addressed the limitations and the potential impact of their work.

---

> ### Author Rebuttal · Authors · 2024-08-07
>
> We would like to thank the referee DkFj for the work and for the positive feedback! Next, we answer the issues raised.
>
> **Bottleneck of the convergence rate and counter-example or a deeper discussion on this convergence rate**
> This is indeed a very important question. First of all, the analysis of Theorems 1-3 can be adjusted for the whole range of step sizes $\alpha _k = c_0 / k^{\gamma}, \quad \gamma \in (0,1)$. The only modification would affect lower bounds on step size $n$ in the assumptions $A2$ and $A3$, respectively. There is a reason to believe that the moment bound of Theorem~1 is sharp, that is, the best possible bound on $\mathsf{E}^{1/2}[\|\|D\|\|^2]$ is of order $n^{-1/4}$ when setting $\gamma = 1/2$. The corresponding lower bound on moments of the remainder (in $n$) terms can be found for the setting of strongly convex optimization in [Li et al, 2022]. We expect that this result can be generalized for the LSA setting as well. However, the tightness of the moment of Theorem 1 bound does not directly imply the tightness of the bounds of Kolmogorov distance $\rho_n^{\text{(Conv)}}$. It is hard to say if the cross-correlation terms appearing between the linear statistics $W$ and non-linear statistics $D$ in the bound (13) are sharp. See also the general discussion on this topic.
>
> We leave further exploration of this question as a promising direction for a future work.
>
> **Typos and misprints**
> We thank the referee for careful reading of the manuscript and will fix the raised issues in the revised version of the paper. We will also re-generate Figure $1$ with longer trajectories and change legend as suggested by the referee in order to improve readability.
>
>
> **Line 307 asks for a sequence of "TD(0) updates", what does the "(0)" stands for?**
> In general one can perform a policy evaluation using the whole family of TD($\lambda$) algorithms, where $\lambda \in [0,1]$. One can find the details in the paper [Tsitsiklis and Van Roy, 1996]. However, when we choose an instance of the algorithm with parameter $\lambda > 0$, corresponding dynamics of parameter updates $(\theta_k)_{k \geq 0\} $ becomes non-Markovian. TD($0$) is arguably the most popular algorithm of this family, and the only one which falls directly into the stochastic approximation paradigm.
>
> References:
> [Li et al, 2022] Li, C.J., Mou, W., Wainwright, M. and Jordan, M., 2022, June. Root-sgd: Sharp nonasymptotics and asymptotic efficiency in a single algorithm. In Conference on Learning Theory (pp. 909-981). PMLR.
>
> [Tsitsiklis and Van Roy, 1996] Tsitsiklis, John, and Benjamin Van Roy. "Analysis of temporal-diffference learning with function approximation." Advances in neural information processing systems 9 (1996).

---

> > ### Comment · Reviewer_DkFj · 2024-08-09
> >
> > I thank the authors for their rebuttal.
> > The comment made in the rebuttal on the optimal $n^{1/4}$-rate obtained in [Bolthausen, 1982] for the martingale Berry-Essen should be on the main text. I wonder if the authors can find an example matching their upper bound (thus showing it is optimal), maybe drawing inspiration from the example in Section 6 of [Bolthausen, 1982].
> > I thank the authors for the explanation of the meaning of TD(0), it should also be included on the revised text.

---

> > > ### Author Response · Authors · 2024-08-12
> > >
> > > Yes, we will include the corresponding discussion on the optimal rates in martingale CLT in the revised version of the text, as well as a comment on TD(0). Fetching the example provided in [Bolthausen, 1982] into the LSA paradigm is not immediate, but we work in this direction, and, of course, if we succeed to construct such a lower bound, we will include it in the final text.

---

### Author Rebuttal · Authors · 2024-08-07

We thank the reviewers for their thorough feedback. We are pleased that reviewers deemed our contributions to the Berry-Esseen bounds for Polyak-Ruppert averaged LSA and non-asymptotic bootstrap validity as new.

The general question, which was raised by the referees **DkFj** and **uNZU** is related to the tightness of our bounds and availability of certain lower bounds. We do not have a formal proof for the tightness of our bounds and highlight that it is an excellent research question. First of all, we believe that the moment bound on the remainder term $\|D\|$ in Theorem 1 is sharp in terms of its dependence in $n$, since similar results can be traced for the setting of strongly convex optimization in [Li et al, 2022]. However, the tightness of the moment of Theorem 1 bound does not directly imply the tightness of the bounds of Kolmogorov distance $\rho_n^{\text{(Conv)}}$. Second, the key bound (13) in our analysis writes as
$$
\sup_{A \in Conv(\mathbb{R}^d)} | \mathbb{P}(T \in A) - \mathbb{P}(\eta \in A)| \leq 259 d^{1/2} \Upsilon + 2 \mathsf{E}[\|W\| \|D\|] + 2 \sum_{\ell=1}^n \mathsf{E}[\|\xi_\ell\| \|D - D^{(\ell)}\|]\,.
$$
where $W$ and $D$ are the leading (linear) and remainder part of our considered non-linear statistics. It is shown in [Chen and Shao, 2007], Section 4, that the $3$rd term in the above sum can not be removed. However, it is less clear if the same reasoning applies to the correlation term $2 \mathsf{E}[\|W\| \|D\|]$. At the same time, it is this term which explains the $n^{-1/4}$ scaling of the final bound. However, there is another evidence which suggests that $n^{-1/4}$ is a correct order. We can write the statistics of interest $T = n^{1/2} \bar{A} (\bar{\theta_n} - \theta^*)$ within the particular decomposition
$$
T = \frac{1}{\sqrt{n}} \sum_{k=n}^{2n-1} \epsilon_{k+1} + \frac{1}{\sqrt{n}}\sum_{k=n+1}^{2n}(A_k - \bar{A}) (\theta_{k-1} - \theta^*) + R
$$
Here $R$ contains remainder terms of non-linear statistics $D$, which are of smaller order in $n$. The first two terms in the above sum forms a martingale with respect to the natural filtration, and it is known that typical Berry-Esseen rate in martingale CLT is $n^{-1/4}$, see e.g.
[Bolthausen, 1982]. At the same time, the counterexample of Bolthausen has a special structure, which not necessarily falls into the structure of the leading terms above. To conslude, further investigations of lower bounds are needed, but available (incomplete) evidence from both moment point of view and martingale CLT suggests that $n^{-1/4}$ should be optimal.

We also attach a pdf file with slightly increased number of observations $n$ as suggested by the referee **ew5L**. We increased maximal number of observations by a factor of $2$ (till $3276800$) and added scaling of Kolmogorov distance by $n^{1/4}$ without logarithmic scaling of the $y$-axis to better highlight scaling of our approximation rate.

 We address the other more specific concerns directly in the rebuttal to each reviews.

**References:**

[Chen and Shao, 2007] Chen, L. H. and Shao, Q.-M. (2007). Normal approximation for nonlinear statistics using a concentration inequality approach. Bernoulli 13(2) 581–599.
[Bolthausen, 1982] Bolthausen, E. Exact convergence rates in some martingale central limit theorems. The Annals of Probability, pp.672-688, 1982.

---

### Decision · Program_Chairs · 2024-09-25

**Decision:**

Accept (poster)

**Comment:**

The present paper proves normal approximation (CLT) and validity of a bootstrap procedure for the law of the Polyak-Ruppert averaged iterates of linear stochastic approximation (LSA). Their main results have n^{-1/4} convergence rates (up to polylogs and dimension-dependent terms) under reasonable assumptions. Intriguingly, the authors suggest that the magnitude of the error in their CLT might be optimal.

The paper was generally well-received by the reviewers, with scores 5, 6 and 7. The issues raised in the revision period do not seem significant, and can be resolved in the final version. In that connection, I would encourage the authors to add their remarks on weaker assumptions to the text (cf. the response to reviewer uNZU).

Overall, I believe the paper deserves to be accepted: the quantitative CLT is quite interesting, and the bootstrap results seem to be new.